# Beta cell-derived cholecystokinin drives obesity-associated pancreatic adenocarcinoma development

Cathy C. Garcia[1,2,3,4,18], Aarthi Venkat[5,6,18], Daniel C. McQuaid[1,2,7,18], Sherry S. Agabiti[1,2], Alexander Tong[8], Boby Mathew[9,10], Rebecca L. Cardone[11], Rebecca Starble [12], Christian F. Ruiz[1,2], Christy Zheng[1,2,3], Akin Sogunro[1,2,7], Jeremy B. Jacox [2,13,14], Ken H. Loh [9,10,15], Richard G. Kibbey[11,13,15,16], Smita Krishnaswamy [1,5,8,19] & Mandar Deepak Muzumdar [1,2,3,7,13,14,17,19]

Pancreatic endocrine-exocrine crosstalk plays a key role in normal physiology and disease and can be altered by host metabolic states, such as obesity. Classically, endocrine islet beta (β) cell secretion of insulin is thought to promote the development of obesity-associated pancreatic adenocarcinoma (PDAC), an exocrine cell-derived tumor. Here, we show that β cell expression of the peptide hormone cholecystokinin (CCK) is necessary and sufficient for obesity-associated PDAC progression in mice and that CCK expression – rather than insulin – correlates strongly with enhanced tumorigenesis. Single-cell RNA-sequencing, in silico latent-space archetypal and trajectory analysis, and experimental lineage tracing in vivo reveal that obesity induces the expansion of postnatal immature β cells, which adapt to express CCK via stress-responsive JNK/cJun signaling. Finally, obesity perturbs CCK-dependent peri-islet exocrine cell transcriptional states and enhances islet-proximal tumor formation. These results define endocrine-exocrine CCK signaling as a bona fide driver of obesity-associated PDAC development and uncover avenues to target the endocrine pancreas to subvert exocrine tumorigenesis.

The pancreas is comprised of two functionally distinct cellular compartments: 1) the *endocrine* pancreas, composed of islet cells that secrete hormones to maintain glucose homeostasis, and 2) the *exocrine* pancreas, encompassing acinar cells that produce digestive enzymes and the ducts through which these enzymes traverse. Recent studies have challenged the longstanding dogma that the endocrine and exocrine compartments function independently, demonstrating endocrine-exocrine crosstalk in normal physiology

[1]Department of Genetics, Yale University School of Medicine, New Haven, CT, USA. [2]Yale Cancer Biology Institute, Yale University, West Haven, CT, USA. [3]Molecular Cell Biology, Genetics, and Development Program, Yale University, New Haven, CT, USA. [4]Stanford Cancer Institute, Stanford University, Palo Alto, CA, USA. [5]Computational Biology and Bioinformatics Program, Yale University, New Haven, CT, USA. [6]Eric and Wendy Schmidt Center at the Broad Institute of MIT and Harvard, Cambridge, MA, USA. [7]M.D.-Ph.D. Program, Yale University, New Haven, CT, USA. [8]Department of Computer Science, Yale University, New Haven, CT, USA. [9]Department of Comparative Medicine, Yale University School of Medicine, New Haven, CT, USA. [10]Institute for Biomedical Design and Discovery, Yale University, West Haven, CT, USA. [11]Department of Internal Medicine, Section of Endocrinology, Yale University School of Medicine, New Haven, CT, USA. [12]Department of Pathology, Yale University School of Medicine, New Haven, CT, USA. [13]Yale Cancer Center, Smilow Cancer Hospital, New Haven, CT, USA. [14]Department of Internal Medicine, Section of Medical Oncology and Hematology, Yale School of Medicine, Yale University, New Haven, CT, USA. [15]Yale Center for Molecular and Systems Metabolism, Yale University, New Haven, CT, USA. [16]Department of Cellular & Molecular Physiology, Yale University School of Medicine, New Haven, CT, USA. [17]Program in Genetics, Genomics, and Epigenetics, Yale Cancer Center, Yale University, New Haven, CT, USA. [18]These authors contributed equally: Cathy C. Garcia, Aarthi Venkat, Daniel C. McQuaid. [19]These authors jointly supervised this work: Smita Krishnaswamy, Mandar Deepak Muzumdar. ✉e-mail: smita.krishnaswamy@yale.edu; mandar.muzumdar@yale.edu

and disease[1,2]. For example, maturity-onset diabetes of the young type 8 (MODY 8) is associated with mutations of *CEL*, a gene that encodes for carboxyl ester lipase produced and secreted by acinar cells[3]. Mutant CEL is acquired by islet beta (β) cells and aggregates intracellularly to induce endoplasmic reticulum (ER) stress, resulting in β cell dysfunction and diabetes, a disease of the endocrine pancreas[4]. Other exocrine diseases have similarly been associated with the development of diabetes, including chronic pancreatitis, Wolcott-Rallison syndrome, cystic fibrosis, and pancreatic ductal adenocarcinoma (PDAC)[2]. Conversely, obesity and diabetes−host metabolic states linked to β cell dysfunction−are associated with an increased risk of developing and dying of PDAC[5–10], a highly lethal tumor thought to primarily arise from acinar cells of the exocrine pancreas[11,12]. However, the cellular and molecular mechanisms that govern endocrine-exocrine interactions in tumorigenesis are not completely understood.

To date, few studies have directly examined the importance of β cells and β cell-secreted hormones in PDAC pathogenesis. Recent experiments have demonstrated a functional role of insulin signaling in pancreatic tumorigenesis. Partial knockout of the insulin genes (*Ins1* and *Ins2*) or acinar cell-specific knockout of the insulin receptor (*InsR*) reduced oncogenic *Kras*-driven tumor development in mice fed a high-fat diet (HFD)[13–15]. Mechanistically, insulin signaling in acinar cells increased digestive enzyme production and acinar-to-ductal metaplasia (ADM)[15], an early prerequisite step in PDAC development[12,16]. Our lab previously showed that obesity (due to loss of the appetite suppression hormone leptin (*Lep^{ob/ob}*)[17]) induces aberrant β cell expression of the peptide hormone cholecystokinin (CCK) in mice and that islet CCK expression is associated with increased body-mass index (BMI) in humans[18]. CCK stimulates acinar cell proliferation, digestive enzyme production, and ADM[18–20], arguing that CCK−like insulin−may drive PDAC development. Indeed, exogenous administration of the CCK analog cerulein promotes oncogenic *Kras*-driven tumorigenesis in mice[21]. We crossed *Lep^{ob/ob}* mice with an autochthonous mouse model that mimics the genetic and histologic features of human PDAC progression (*KC: Pdx1-Cre;Kras^{LSL-G12D}*)[22]. Resultant obese *KCO* (*KC;Lep^{ob/ob}*) mice exhibited enhanced preinvasive pancreatic intraepithelial neoplasia (PanIN) formation and progression to invasive PDAC, which was associated with increased CCK expression[18]. Furthermore, both weight loss and the antidiabetic dapagliflozin reduced β cell CCK expression and neoplasia[18], arguing that β cell-derived CCK is a marker of both a dysregulated metabolic state and enhanced exocrine tumorigenesis.

Here, we demonstrate that β cell CCK expression is necessary and sufficient for obesity-associated PDAC development in vivo, validating CCK as a bona fide driver of exocrine tumor formation. We leverage single-cell RNA sequencing (scRNA-seq) of multiple congenic obesity models and a suite of machine learning-based computational tools for in silico lineage tracing (TrajectoryNet)[23], archetypal analysis (AAnet)[24,25], batch integration across datasets (scMMGAN)[26], optimal-transport distance-preserving embedding of patient data (DiffusionEMD)[27], and a framework for uncovering cellular transitions and dynamic regulatory interactions (Cflows)[28]. Using these methods, we gain deep insights into β cell heterogeneity and the dynamic transcriptional changes that lead to β cell CCK expression in obesity (Supplementary Fig. 1). Through both in silico and experimental lineage tracing approaches and gene regulatory analysis, we further discover that CCK+ β cells emerge from stress-induced expansion and adaptation of a postnatal immature β cell population and that CCK expression is mediated by stress-responsive JNK/cJun signaling. Finally, we show that obesity is associated with CCK-dependent transcriptional alterations in peri-islet acinar cells and increased peri-islet tumorigenesis. Together, our results establish the critical importance of endocrine-exocrine signaling in PDAC pathogenesis and implicate

β cell stress pathways and hormones that could be targeted to intercept exocrine tumorigenesis.

## Results

### β cell CCK expression promotes PDAC progression

While obesity is associated with increased β cell CCK expression[18,29], the extent to which β cell CCK can independently drive tumorigenesis remains unclear. To determine this, we crossed a transgenic model of β cell-specific CCK (mouse gene *Cck*) overexpression (*Ins1-Cck*)[30] to our previously generated *KCO* model of obesity-associated PDAC (Fig. 1a). As anticipated, 3-month-old lean *KC;Lep^{ob/+};Ins1-Cck/+* and obese *KCO* mice exhibited increased pancreatic CCK expression relative to lean *KC;Lep^{ob/+}* littermates (Fig. 1b, c). Strikingly, *KC;Lep^{ob/+};Ins1-Cck/+* mice demonstrated a significant increase in disease burden (combined ADM, PanIN, and PDAC) and propensity to progress to PDAC compared to age-matched *KC;Lep^{ob/+}* littermates (Fig. 1d, e). These differences in tumorigenesis were independent of changes in weight, glucose, or endogenous insulin production (C-peptide) (Fig. 1f–h), and C-peptide levels were not associated with *Cck* expression nor disease burden in *KC;Lep^{ob/+};Ins1-Cck/+* mice (Fig. 1i, j). Critically, there was no significant difference in overall disease burden or progression to PDAC between *KC;Lep^{ob/+};Ins1-Cck/+* and *KCO* littermates (Fig. 1d, e), suggesting that β cell CCK overexpression is sufficient to phenocopy the pro-tumorigenic effects of obesity. Together, these experiments argue that β cell CCK is a potential insulin-independent driver of pancreatic tumor development in obesity.

### Pancreatic CCK drives obesity-associated PDAC development

To ascertain if CCK is necessary for obesity-associated PDAC development, we obtained a conditional model for *Cck* knockout (*Cck^{flox/flox}*) generated by the International Knockout Mouse Project[31]. We first validated pancreas-specific *Cck* knockout in obese non-tumor-bearing 4-month-old *Pdx1-Cre;Lep^{ob/ob};Cck^{flox/flox}* mice (Supplementary Fig. 2a, b), as we observed pancreatic *Cck* expression was abolished while duodenal expression was preserved (Supplementary Fig. 2c–e). Since CCK has been reported to be a β cell survival factor[29,30,32], we next evaluated the role of pancreatic *Cck* knockout on islet health. *Pdx1-Cre;Lep^{ob/ob};Cck^{flox/flox}* mice displayed comparable glucose levels and islet mass to *Cck^{flox/+}* and *Cck^{+/+}* controls (Supplementary Fig. 2f, g), corresponding to similar C-peptide levels under both fasted and fed conditions (Supplementary Fig. 2h). These data argue that β cell CCK expression is dispensable for maintaining islet homeostasis in obesity.

We next tested whether β cell *Cck* knockout altered tumor development by crossing *Cck^{flox/flox}* mice into the *KCO* model (Fig. 2a). As expected, 3-month-old *KCO;Cck^{flox/flox}* littermates demonstrated significantly decreased pancreatic CCK expression compared to age-matched littermate controls (Fig. 2b, c). Remarkably, *KCO;Cck^{flox/flox}* mice exhibited significantly reduced disease burden and progression to adenocarcinoma relative to *KCO;Cck^{+/+}* littermates (Fig. 2d, e) despite increased weight and C-peptide (Supplementary Fig. 2i–k). Partial reductions in disease burden and progression were also observed in *KCO; Cck^{flox/+}* mice (Fig. 2d, e), consistent with a gene dosage effect. Strikingly, we observed that pancreatic *Cck* expression was inversely correlated with insulin secretion (C-peptide) in multiple cohorts of *KCO* mice, including those subject to *Cck* knockout (Figs. 1k and 2f). In fact, *Cck* gene expression strongly correlated with disease burden, whereas C-peptide was negatively correlated (Figs. 1l, m and 2g, h). Combined, these data indicate that β cell CCK expression is necessary for obesity-associated PDAC progression and that CCK – rather than insulin – is the primary driver of pancreatic tumorigenesis in these obesity-mediated tumor models.

### CCK and insulin expression are inversely correlated in obesity

We next investigated the molecular mechanisms by which obesity reprograms β cells to express CCK and drive tumor progression.

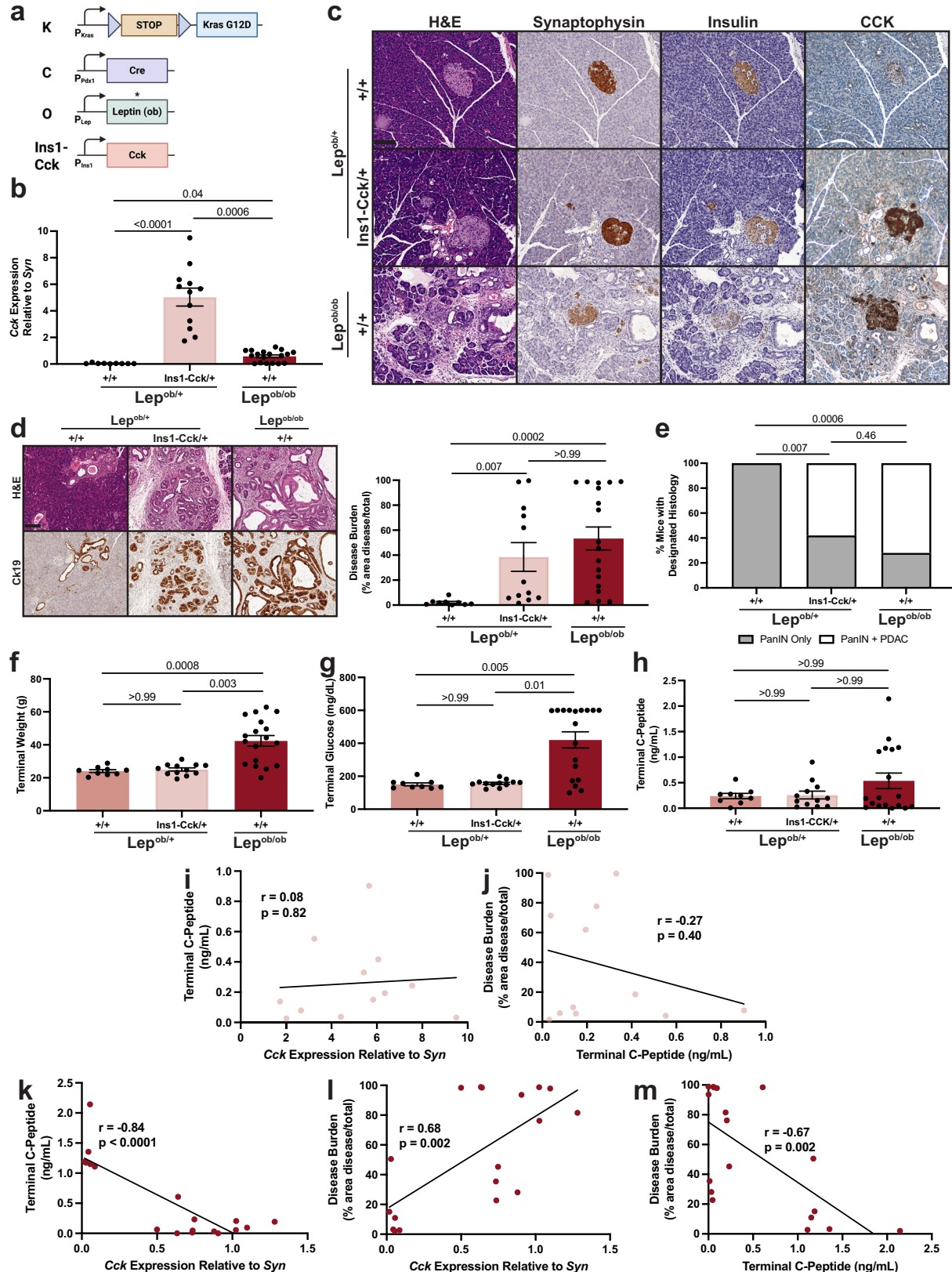

We performed scRNA-seq of isolated islet cells from age-matched male congenic (C57/B6) wild-type (WT), moderately obese HFD-fed, and severely obese *Lep*ob/ob mice that displayed increasing weight and glucose levels (Supplementary Fig. 3a–c). Consistent with these phenotypic observations, primary islets isolated from HFD-fed and *Lep*ob/ob mice exhibited decreased glucose-stimulated insulin secretion (GSIS), a marker of β cell dysfunction (Supplementary Fig. 3d, e). After preprocessing of scRNA-seq data (see Methods), we visualized and clustered 23,469 single cells from all samples (Supplementary Fig. 3f). We annotated these clusters based on published endocrine and exocrine

**Fig. 1 | β cell CCK expression promotes PDAC development. a** Schematic of alleles used to generate *KC;Lep^ob/+^*, *KC;Lep^ob/+^;Ins1-Cck/+*, and *KCO* mice. Created in BioRender. McQuaid, D. (2026) https://BioRender.com/5e86buu. **b** Pancreatic *Cck* expression (qRT-PCR, mean ± SEM) normalized to islet marker synaptophysin (*Syn*) of mice of designated genotypes. *p*-values of Kruskal-Wallis with Dunn's post-hoc test are shown. **c** Images of CCK IHC on pancreata of 3-month old mice of each designated genotype. Images are representative of *n* = 3 mice per group. Scale bar, 100 μm. **d** IHC images of ductal tumorigenesis (Ck19) and quantification of disease burden (mean ± SEM) of 3 month-old mice of each designated genotype. Images are representative of *n* = 3 mice per group. *p*-values of Kruskal-Wallis with Dunn's post-hoc test are shown. Scale bar, 100 μm. **e** Percentage of mice of designated genotypes harboring PanINs and/or PDAC. *p*-values of two-sided Fisher's exact tests are shown. **f–h** Terminal weight (**f**), random glucose (**g**), and C-peptide (**h**) (mean ± SEM) of mice of designated genotypes. *p*-values of Kruskal-

Wallis with Dunn's post-hoc test are shown. For quantitative measures in (**b–h**), 3-month-old *KC;Lep^ob/+^* (*n* = 9 mice (4 male, 5 female)), *KC;Lep^ob/+^;Ins1-Cck/+* (*n* = 12 mice (6 male, 6 female)), and *KCO* (*n* = 18 mice (8 male, 10 female)) littermates were analyzed. **i, j** No association between terminal C-peptide and pancreatic *Cck* expression (**i**) (Pearson correlation analysis, *r* = 0.08, *p* = 0.82) nor disease burden (**j**) (Pearson correlation analysis, *r* = 0.27, *p* = 0.40). Each point represents one 3-month-old *KC;Lep^ob/+^;Ins1-Cck/+* mouse (*n* = 12). **k** Pancreatic *Cck* expression is inversely correlated (Pearson correlation analysis, *r* = 0.84, *p* < 0.0001) with terminal C-peptide in obese mice. **l** Pancreatic *Cck* expression is positively correlated (Pearson correlation analysis, *r* = 0.68, *p* = 0.002) with disease burden in obese mice. **m** Terminal C-peptide levels are inversely correlated (Pearson correlation analysis, *r* = −0.67, *p* = 0.002) with disease burden in obese mice. For (**k–m**), each point represents one 3-month-old *KCO* mouse (*n* = 18). Source data are provided as a Source Data file.

markers from human and mouse islet maps[33] (Supplementary Fig. 3g) and subsetted the data to 20,294 endocrine cells for downstream analysis. We re-embedded the endocrine cells and annotated clusters corresponding to alpha (α), β, delta (δ), and pancreatic polypeptide (PP) cells based on high expression of glucagon (*Gcg*), insulin (*Ins1*, *Ins2*), somatostatin (*Sst*), and pancreatic polypeptide (*Ppy*), respectively (Supplementary Fig. 4a). Similar to prior studies[34,35], a small population of polyhormonal cells (expressing insulin and a second hormone) was also observed. *Cck* was primarily expressed in β cells, which showed the greatest variation in transcriptional states across conditions (Supplementary Fig. 4a) and were the most abundant islet cell type (Supplementary Fig. 4b).

We subsequently re-embedded β cells from all samples (Supplementary Fig. 4c, d) and continuously inferred individual and heterogeneous cellular trajectories over the progression of obesity using an ODE-based dynamic optimal transport neural network termed TrajectoryNet and the *Cflows* framework[23,28]. TrajectoryNet is designed to interpolate continuous dynamics for every single cell across distinct timepoints, where the dynamics learned are biologically plausible through ensuring energy efficiency and modeling cellular proliferation. By learning individual cellular trajectories and leveraging real timepoint dynamics to guide trajectories, this approach represents an advancement over traditional trajectory-based inference methods, which infer a single pseudotemporal ordering over the entire population of cells based on expression similarity, which may not correspond to true latent dynamics[36]. Here, we learned individual cellular trajectories from the combined WT (timepoint 1) to HFD (timepoint 2) to *Lep^ob/ob^* (timepoint 3) conditions, and analyzed transcriptional changes along these trajectories by decoding them back into the gene expression space. This analysis revealed a clear trajectory corresponding to increasing obesity (Supplementary Fig. 5a, b). Along this trajectory, insulin (*Ins1*, *Ins2*) showed a reciprocal relationship with CCK, wherein the insulin genes decreased while *Cck* increased in expression (Supplementary Fig. 5c). These findings were concordant with the inverse relationship between pancreatic *Cck* expression and C-peptide in tumor-bearing *KCO* mice (Figs. 1k and 2f). Gene trends learned by TrajectoryNet showed a decrease in β cell identity, maturation, and insulin secretion markers and increased protein processing, dedifferentiation, and ER stress markers as obesity progresses (Supplementary Fig. 5d). These data argue that β cells retain secretory capacity but upregulate *Cck* at the expense of insulin. We took advantage of the Min6 murine insulinoma cell line to directly detect whether CCK is sorted into granules for secretion. This frequently employed β cell model expresses insulin and CCK at comparable levels to β cells of *KCO* mice (Supplementary Fig. 5e). Co-immunoelectron microscopy revealed that CCK was present in the same secretory granules as insulin, including those at the plasma membrane (Supplementary Fig. 5f). These findings indicate that β cell CCK may be secreted to drive tumor progression.

## Archetypal analysis reveals obesity-dependent alterations in β-cell heterogeneity

Recent studies have suggested that adult β cells are heterogeneous and that host metabolic conditions can modulate this heterogeneity[34,37,38]. To characterize obesity-associated β cell heterogeneity, we separately re-embedded the cells for each condition (WT, HFD, and *Lep^ob/ob^*) and applied the archetypal analysis approach AAnet. AAnet is an autoencoder that embeds cells into a simplicial latent representation and defines them with respect to cellular "archetypes," or extreme states[24,25], termed "latent-space" archetypal analysis. Unlike traditional archetypal analysis, AAnet does not assume that cells are archetypal in gene expression space but instead finds a lower-dimensional latent space that is amenable to such factor analysis. This enables the characterization of β cell states while preserving continuous variation between highly plastic ones. AAnet identified seven archetypes (AT) within cells from WT mice, four ATs from HFD-fed mice, and three ATs from *Lep^ob/ob^* mice, where the number of ATs was determined based on the elbow point of reconstruction error[20] (Fig. 3a and Supplementary Fig. 6a). ATs showed varying abundance across the three conditions (Supplementary Fig. 6b). Similarity scores calculated between all ATs identified three groups with shared signatures (Groups 1, 2, and 3) and differing proportions across conditions (Supplementary Fig. 6c, d).

We characterized these β cell ATs using marker genes defined by previous research on β cell heterogeneity[37,39,40] (Fig. 3b), visualizing each marker gene and its variability (Supplementary Fig. 6e–g and Supplementary Fig. 7-9) and determining differential expression across all genes per condition for each AT (Supplementary Data 1). Group 1 ATs (WT AT 5 and 7; HFD AT 1; *Lep^ob/ob^* AT 1) were significantly (*q* < 0.05) enriched for expression of insulin, insulin secretion markers, and β cell maturation markers, consistent with a mature β cell phenotype (Fig. 3b and Supplementary Data 1). Group 3 ATs (WT AT 1, 2, 3, and 6; HFD AT 2 and 3) were characterized by significantly (*q* < 0.05) higher mitochondrial expression and – compared to Group 1 – reduced insulin expression (Fig. 3b and Supplementary Data 1). For WT AT 1 and HFD AT 3, maturation and insulin secretion markers *Ucn3*, *Slc2a2*, *Slc30a8* were significantly enriched. Group 2 ATs (WT AT 4; HFD AT 4; *Lep^ob/ob^* AT 2 and 3) displayed significantly lower expression of insulin (*Ins1*) and maturation markers and higher expression of dedifferentiation/immaturity[33,41], hormone processing, and secretion markers (Fig. 3b and Supplementary Data 1). Together, these findings capture β cell diversity and complexity and reveal archetypal clusters of immature (Group 2), intermediate (Group 3), and mature (Group 1) β cells across WT, HFD, and *Lep^ob/ob^* mice.

## Identification of the cell-of-origin for CCK-expressing β cells in obesity

We next sought to decipher how *Cck*+ β cells emerge in response to obesity. Obesity increases insulin demand, augmenting β cell mass in mice and humans[42,43]. Lineage tracing studies in mice have shown that

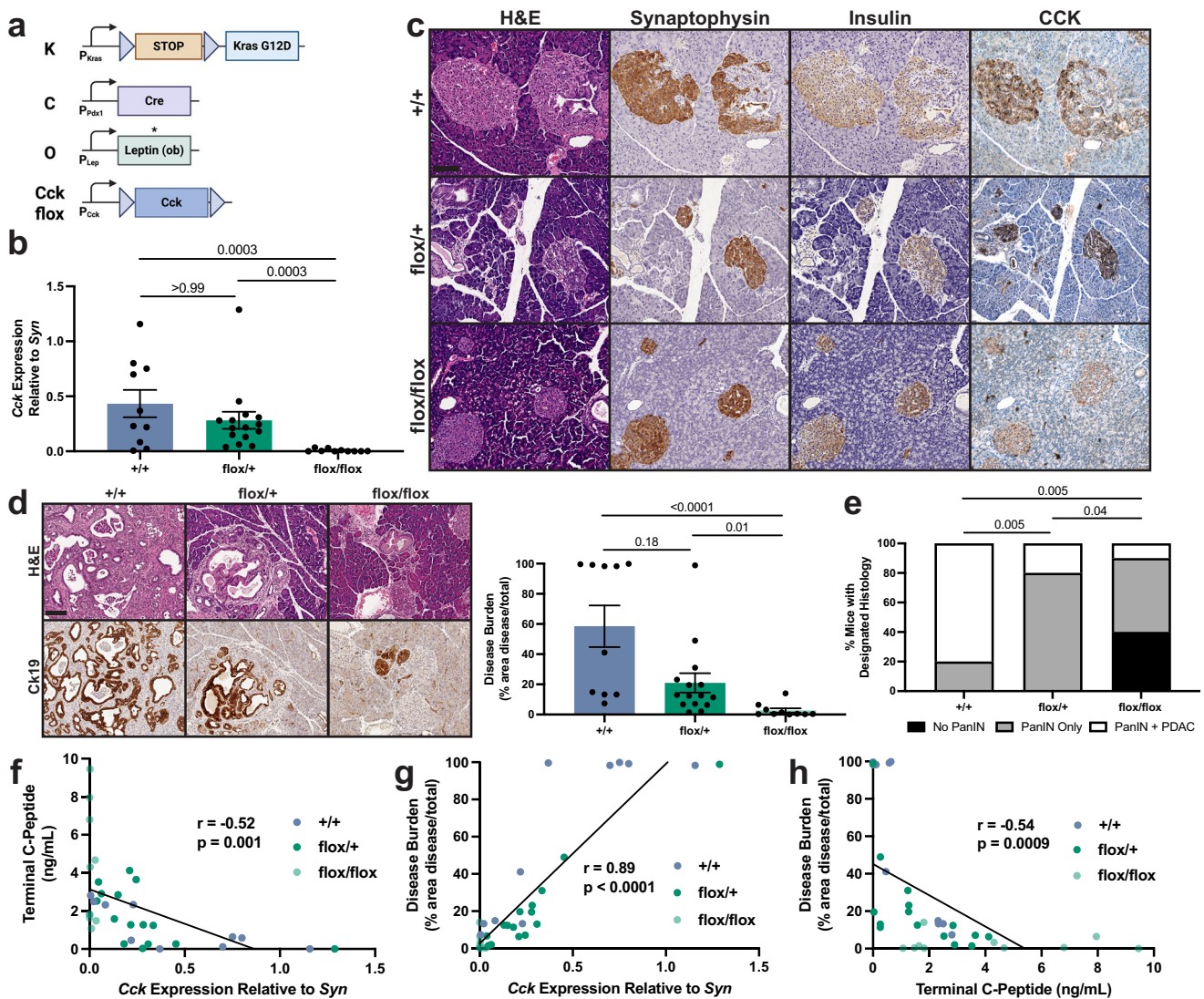

**Fig. 2 | Pancreatic CCK is required for obesity-associated PDAC progression.**
**a** Schematic of alleles used to generate *KCO*, *KCO;Cck^flox/+^*, and *KCO;Cck^flox/flox^* mice.
Created in BioRender. McQuaid, D. (2026) https://BioRender.com/b26g9z9.
**b** Pancreatic *Cck* expression (qRT-PCR, mean ± SEM) normalized to islet marker
synaptophysin (*Syn*) of *KCO* mice of designated *Cck* genotypes. *p*-values of Kruskal-
Wallis with Dunn's post-hoc test are shown. **c** Images of CCK IHC on pancreata of 3-
month-old *KCO* mice of each designated *Cck* genotype. Images are representative
of *n* = 3 mice per group. Scale bar, 100 μm. **d** IHC images of ductal tumorigenesis
(Ck19) and quantification of disease burden (mean ± SEM) of 3-month-old *KCO*
mice of each designated *Cck* genotype. Images are representative of *n* = 3 mice per
group. *p*-values of Kruskal-Wallis with Dunn's post-hoc test are shown. Scale bar,
100 μm. **e** Percentage of *KCO* mice of designated *Cck* genotypes harboring PanINs

and/or PDAC. *p*-values of two-sided Fisher's exact test with Freeman-Haltman
extension are shown. **f** Pancreatic *Cck* expression is inversely correlated (Pearson
correlation analysis, *r* = −0.52, *p* = 0.001) with terminal C-peptide of *KCO* mice of
designated *Cck* genotypes (*n* = 35 total mice). **g** Pancreatic *Cck* expression is
positively correlated (Pearson correlation analysis, *r* = 0.89, *p* < 0.0001) with dis-
ease burden of *KCO* mice of designated *Cck* genotypes (*n* = 35 total mice).
**h** Terminal C-peptide levels are inversely correlated (Pearson correlation analysis,
*r* = −0.54, *p* = 0.0009) with disease burden of *KCO* mice of designated *Cck* geno-
types (*n* = 35 total mice). For quantitative measures in (**b**–**h**), 3-month-old
*KCO;Cck^flox/flox^* mice (*n* = 10 mice (9 male, 1 female)) relative to *KCO* (*n* = 10 mice (7
male, 3 female)) and *KCO;Cck^flox/+^* littermates (*n* = 15 (8 male, 7 female)) were ana-
lyzed. Source data are provided as a Source Data file.

new β cells primarily arise from self-duplication rather than transdif-
ferentiation from other endocrine or exocrine cell types under a
variety of physiologic scenarios, including homeostasis, pregnancy,
and pancreatic injury[44]. In contrast, upon complete β cell ablation, α
(*Gcg*+) and δ (*Sst*+) cells may acquire insulin (*Ins1*+/*Ins2*+)
expression[45,46], supporting islet cell plasticity in β cell regeneration.
Additional studies have demonstrated that exocrine duct cells can
transdifferentiate into β cells upon pancreatic injury[47,48]. To distinguish
whether *Cck*+ cells arise from β cell duplication versus transdiffer-
entiation in obesity, we leveraged TrajectoryNet for in silico lineage
tracing. Since TrajectoryNet outputs a trajectory for each cellular state,
we can trace the trajectories of pathogenic states backward to find
their cell of origin. We traced back the most likely paths of *Cck*+ *Lep^ob/ob^*

cells, allowing us to identify "cell-of-origin" cells, defined as the cells at
the first timepoint of the learned trajectories (Fig. 3c). We then com-
pared the proportion of each cell type within the cells-of-origin versus
the distribution in measured WT cells, terming the ratio the cell type
"enrichment score" (see Methods). This analysis revealed that *Cck*+
*Lep^ob/ob^* β cells derive from β cells in the WT condition, and *Cck*+ *Lep^ob/ob^*
polyhormonal cells derive from polyhormonal cells in the WT condi-
tion (Fig. 3d). These data raise the possibility that *Cck*+ cells arise from
pre-existing β cells rather than by transdifferentiation of other
cell types.

We next used learned cellular dynamics from TrajectoryNet to
decipher which β cell ATs are most likely to be the cell-of-origin for the
highest *Cck*-expressing (*Cck*-hi) *Lep^ob/ob^* cells (*Lep^ob/ob^* AT 2; Fig. 3b).

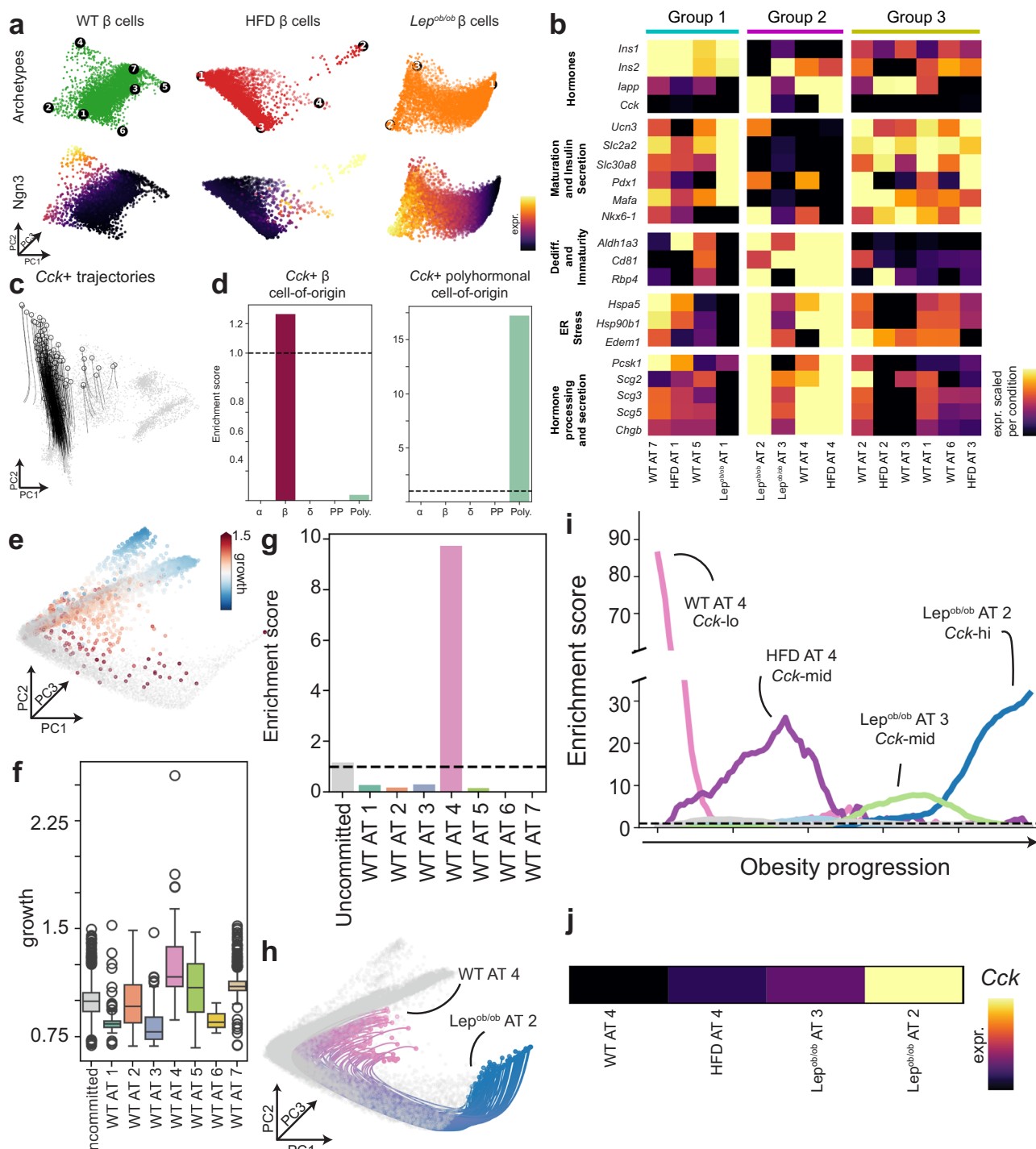

**Fig. 3 | In silico identification of the cell-of-origin for *Cck*-expressing β cells in obesity. a** AAnet-learned archetypes for WT (*n* = 5,847 cells), HFD (*n* = 3,887 cells), and *Lep^ob/ob^* (*n* = 7602 cells) samples (top row). Color embedding of denoised expression (color scale represents min to max of normalized UMI) of Neurogenin 3 (*Ngn3*) across different conditions (bottom row). **b** Heatmap for marker gene expression (color scale represents min to max of mean normalized UMI, scaled per gene within condition) for each AT in (**a**). **c** TrajectoryNet-learned trajectories tracing paths from cells from WT mice to *Cck*+ *Lep^ob/ob^* cells (*n* = 300 randomly sampled trajectories). Gray cells are all endocrine cells (*n* = 20,294 cells). **d** Enrichment score of each islet cell type for cell-of-origin of *Cck*+ *Lep^ob/ob^* β cells (left) and *Cck*+ *Lep^ob/ob^* polyhormonal cells (right), calculated over two TrajectoryNet runs that yielded identical results. **e** TrajectoryNet-inferred proliferation for cells from WT condition (*n* = 5847 cells), where proliferation rate > 1 implies growth. Gray cells are all β cells (*n* = 17,336 cells). **f** Cells committed to WT AT 4

(*n* = 168 cells) have a higher growth rate than other archetypes (WT AT 1 *n* = 703 cells, WT AT 2 *n* = 127 cells, WT AT 3 *n* = 263 cells, WT AT 5 *n* = 136 cells, WT AT 6 *n* = 117 cells, WT AT 7 *n* = 983 cells) or uncommitted cells (*n* = 3350 cells). Box plots display 25, 50, and 75th percentiles ± 1.5 interquartile range (IQR). **g** Enrichment scores (mean over *n* = 2 TrajectoryNet runs) for each WT archetype within cells-of-origin for *Cck*-hi *Lep^ob/ob^* AT 2 shows high enrichment of WT AT 4. **h** TrajectoryNet-learned trajectories tracing paths from cells from WT AT 4 to *Lep^ob/ob^* AT 2 (*n* = 133 trajectories). Gray cells are all β cells (*n* = 17,336 cells). **i** Enrichment scores (mean over *n* = 2 TrajectoryNet runs) for each archetype within cells on the trajectory from WT AT 4 to *Lep^ob/ob^* AT 2, demonstrating archetypes (HFD AT 4 and *Lep^ob/ob^* AT 3) that represent potential intermediate states. **j** Heatmap of *Cck* expression (color scale represents min to max of mean normalized UMI per AT) for WT AT 4, HFD AT 4, *Lep^ob/ob^* AT 3, and *Lep^ob/ob^* AT 2. Source data are provided as a Source Data file.

TrajectoryNet utilizes an auxiliary network to learn the relative proliferation rates of cells from the WT condition, where WT cells predicted to be proliferating have a relative growth rate of >1. Across all WT ATs, cells committed to WT AT 4 had a higher proliferation rate than all other ATs (Fig. 3e, f). This argues that immature β cells represent a regenerative subpopulation in adult mice, consistent with prior data[49,50]. We retrieved the cells-of-origin for the *Cck*-hi *Lep^(ob/ob)* AT 2 cells and determined the enrichment for each WT AT within this population. Overwhelmingly, WT AT 4 was the most likely source for *Cck*-hi *Lep^(ob/ob)* AT 2 cells over all other ATs (Fig. 3g, h). We next leveraged TrajectoryNet to compute the enrichment of potential intermediate state ATs along the trajectory of cells originating from WT AT 4 and terminating at *Lep^(ob/ob)* AT 2 (see Methods). Interestingly, immature HFD AT 4 was enriched in the middle of the trajectory, followed by *Lep^(ob/ob)* AT 3 (Fig. 3i). Strikingly, the four ATs within this trajectory displayed increasing *Cck* expression: WT AT 4 was *Cck*-neg, HFD AT 4 and *Lep^(ob/ob)* AT 3 were *Cck*-mid, and *Lep^(ob/ob)* AT 2 was *Cck*-hi (Fig. 3j), consistent with *Cck* being a key marker of β cell adaptation to obesity. Collectively, these findings suggest that a subpopulation of immature adult β cells with proliferative potential (WT AT 4) may give rise to *Cck*-expressing β cells in obesity.

### Experimental lineage tracing confirms that CCK+ cells originate from immature β cells

To validate our in silico findings, we performed in vivo lineage tracing by crossing tamoxifen-inducible *Gcg-Cre^(ERT2)*, *Ins1-Cre^(ERT)*, and *Ngn3-Cre^(ERT2)* mouse lines[51–53] with *Lep^(ob/ob)* mice harboring a Cre-inducible fluorescent reporter (*Rosa26^(LSL-TdTomato)*)[54] to label α, mature β, and immature β cells, respectively (Fig. 4a). Neurogenin 3 (Ngn3) drives β cell maturation by binding to the promoter of β cell-specific transcription factors (Pax4 and NeuroD), and its expression is lost once β cells differentiate to a mature phenotype[55]. Notably, Ngn3 was enriched in WT AT 4 (Fig. 3a). We first confirmed faithful labeling of α and β cell populations by treating mice with tamoxifen at ~4 weeks of age (before significant obesity onset), isolating pancreata one week later (5 weeks of age), and performing immunofluorescence (IF) for glucagon or insulin, respectively, on tissue sections to determine overlap with TdTomato+ cells (Supplementary Fig. 10a, b). As expected, *Gcg-Cre^(ERT2)* did not label β cells but demonstrated high efficiency labeling of α cells (Supplementary Fig. 10a). *Ins1-Cre^(ERT)* and *Ngn3-Cre^(ERT2)* both labeled β cells, with *Ins1-Cre^(ERT)* resulting in much greater average labeling (Fig. 4b and Supplementary Fig. 10b), consistent with Ngn3+ cells representing a subpopulation of β cells at this time point.

We next evaluated the effects of obesity by treating *Lep^(ob/ob)*; *Rosa26^(LSL-TdTomato)*;*Cre^(ERT)* mice with tamoxifen at 4 weeks of age and performing co-IF for CCK and insulin to determine overlap with TdTomato 12 weeks post-tamoxifen (16 weeks of age). Importantly, all end glucose levels and weights were comparable across groups (Fig. 4c, d). *Gcg-Cre^(ERT2)* mice showed CCK expression in most of the islet. However, only ~3% of CCK+ cells expressed TdTomato (Fig. 4e, f), indicating that CCK+ cells were not primarily arising from α to β cell transdifferentiation. In contrast, most CCK+ cells were labeled with TdTomato in *Ins1-Cre^(ERT)* (Fig. 4e, f). The incomplete labeling of CCK+ cells in *Ins1-Cre^(ERT)* mice was likely due to inefficient labeling of β cells with this *Cre^(ERT)* line in obese mice (Supplementary Fig. 10b), which has been previously observed in lean mice[52]. In support of this hypothesis, we discovered a positive linear correlation between insulin+/TdTomato+ cells (a marker of labeling efficiency) and CCK+/TdTomato+ cells (Supplementary Fig. 10c). Like *Ins1-Cre^(ERT)*, the majority of CCK+ cells were TdTomato+ in obese *Ngn3-Cre^(ERT2)* mice (Fig. 4c, d). Strikingly, this frequency was comparable to *Ins1-Cre^(ERT)* despite reduced initial β cell labeling (Fig. 4b and Supplementary Fig. 10b). This indicates that Ngn3-labeled β cells expanded, consistent with the proliferative potential of immature Ngn3+ adult β cells, as we observed in silico (WT AT 4; Fig. 3e, f). Indeed, the proportion of insulin+/

TdTomato+ cells comparing 12 weeks vs. 1-week post-tamoxifen increased significantly in obese (*Lep^(ob/ob)*) compared to lean (*Lep^(ob/+)*) *Ngn3-Cre^(ERT2)* mice (Fig. 4b and Supplementary Fig. 10d). Conversely, the fraction of insulin+/TdTomato+ cells significantly decreased in obese *Ins1-Cre^(ERT)* mice (Fig. 4b), showing that mature β cells present prior to obesity onset were likely replaced. Neither TdTomato+ nor CCK+ β cells in the *Ngn3-Cre^(ERT2)* or *Ins1-Cre^(ERT)* lines at 16 weeks of age were Ki67+ (Supplementary Fig. 10e), suggesting that islet cells were largely post-mitotic at this time point. Together, these experimental data confirm the in silico analyses demonstrating that an immature subpopulation of postnatal β cells expand and adopt a CCK+ state.

### Obesity is associated with transcriptional signatures of β-cell stress

To characterize transcriptional dynamics along the axis of *Cck*-hi *Lep^(ob/ob)* AT 2 development (Fig. 3h), we clustered all highly variable gene trends to *Lep^(ob/ob)* AT 2 into genes that are decreasing (blue) or increasing (red) in expression over the obesity progression (Fig. 5a). We then performed gene set enrichment analysis (GSEA) for each gene cluster[56,57]. As expected, genes decreasing with obesity were significantly ($q < 0.05$) associated with gene expression regulation in β cells and regulation of insulin secretion (Fig. 5b and Supplementary Data 2a). Upregulated genes over the obesity trajectory were significantly ($q < 0.05$) associated with diabetes and cellular stress responses, including oxidative (electron transport chain, oxidative phosphorylation) and ER stress (Fig. 5b and Supplementary Data 2b). Several additional gene sets related to the unfolded protein response and protein processing in the ER were also significantly enriched for genes increasing over the obesity trajectory (Supplementary Data 2b). Together, these results argue that obesity promotes transcriptional signatures of β cell stress and that oxidative and ER stress are hallmarks of the obesity-induced stress response.

To confirm these findings in orthologous systems, we compared our data to recently published transcriptional signatures derived from primary human islets treated with various β cell stress agents[58] including drugs that induce ER stress (brefeldin A (BFA) and thapsigargin (TG)) and inflammation stress-inducing cytokines (IFNγ, TNFα, and/or IL1β). Enriched gene sets in the BFA- and TG-treated (vs. control) islets—including those associated with ER stress—highly and significantly overlapped with gene sets enriched over the obesity progression ($p = 1.21 \times 10^{-48}$, hypergeometric test) (Fig. 5c and Supplementary Data 3). While human islets treated with cytokines showed significant overlap with the obesity signatures ($p = 7.88 \times 10^{-5}$, hypergeometric test), the overlapping gene sets largely represented a general stress response (Fig. 5d and Supplementary Data 3). These data suggest that obesity induces transcriptional signatures of ER stress in β cells in mice.

We further analyzed the effects of pharmacologic β cell stress induction in vivo in mice subject to multiple low doses of streptozotocin (STZ)[37]. STZ is taken up specifically in β cells via the *Glut2* transporter and induces metabolic (NAD+ depletion) and ER stress[59–62]. Like obesity, STZ treatment led to hyperglycemia, decreased pancreatic *Ins1* expression, and increased β cell *Cck* expression in mice (Fig. 5e–g). To determine the molecular mechanisms of STZ-induced *Cck* expression, we reanalyzed a previously published scRNA-seq dataset of β cells from mice treated with STZ vs. vehicle control[37]. Visualization of these data showed distinct β cell clusters corresponding to treatment groups (β-mSTZ for STZ-treated mice) and two refined subclusters (β1 and β2) within the vehicle-treated cluster, as previously described[37]. We characterized β cell marker gene expression and variability of these subclusters (Supplementary Fig. 11). Concordant with obesity, *Ins1* and *Ins2* expression decreased and *Cck* expression increased with STZ treatment (Fig. 5h). β1 cells displayed high expression of maturation markers and β cell identity transcription

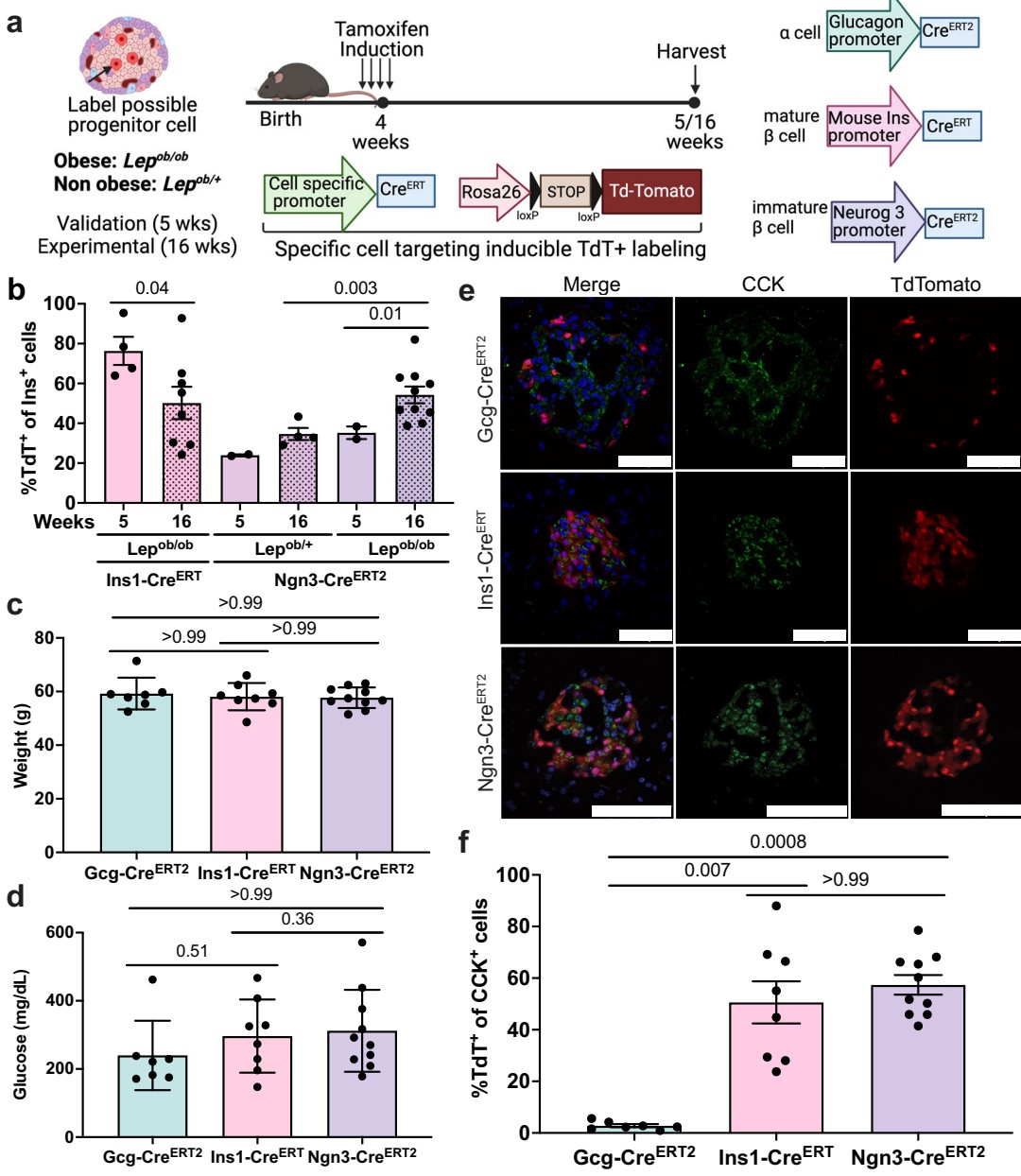

**Fig. 4 | Ngn3+ immature β cells expand to give rise to CCK-expressing β cells.**
**a** Schematic of experimental setup and timeline of lineage tracing experiments. Created in BioRender. Garcia, C. (2026) https://BioRender.com/housa2k. Mice were given tamoxifen at ~4 weeks of age (prior to significant obesity) and analyzed at ~5 weeks of age (one-week post-tamoxifen) to ensure fidelity of labeling or ~16 weeks of age (when islet CCK is present) for lineage tracing analysis. **b** Average percentage (mean ± SEM) of insulin+ cells labeled with TdTomato in obese *Ins1-Cre^ERT;Lep^ob/ob;Rosa26^LSL-TdTomato* (*n* = 4 mice (2 male, 2 female), 5 weeks; *n* = 8 mice (4 male, 4 female), 16 weeks), obese *Ngn3-Cre^ERT2;Lep^ob/ob;Rosa26^LSL-TdTomato* (*n* = 2 mice (1 male, 1 female), 5 weeks; *n* = 10 mice (5 male, 5 female), 16 weeks), and lean *Ngn3-Cre^ERT2;Lep^ob/+;Rosa26^LSL-TdTomato* mice (*n* = 2 mice (1 male, 1 female), 5 weeks; *n* = 4 mice (2 male, 2 female), 16 weeks) administered tamoxifen at 4 weeks of age. *p*-values of two-sided Welch's *t*-test are shown. **c, d** Terminal weight (**c**) and random

glucose (**d**) (mean ± SEM) of *Lep^ob/ob;Rosa26^LSL-TdTomato* mice harboring designated *Cre^ERT* transgenes. *p*-values of Kruskal-Wallis with Dunn's post-hoc test are shown. **e** Images of CCK immunofluorescence and lineage-traced TdTomato-labeling of *Lep^ob/ob;Rosa26^LSL-TdTomato* mice harboring designated *Cre^ERT* transgene. DAPI labels nuclei blue. Scale bar, 100 μm. **f** Average percentage (mean ± SEM) of CCK+ cells labeled with TdTomato in *Lep^ob/ob;Rosa26^LSL-TdTomato* mice harboring designated *Cre^ER* transgenes. *p*-values of Kruskal-Wallis with Dunn's post-hoc test are shown. For (**c–f**), 16 week-old *Lep^ob/ob Gcg-Cre^ERT2;Lep^ob/ob;Rosa26^LSL-TdTomato* (*n* = 7 mice (4 male, 3 female)), *Ins-Cre^ERT;Lep^ob/ob;Rosa26^LSL-TdTomato* (*n* = 8 mice (4 male, 4 female)), and *Ngn3-Cre^ERT2;Lep^ob/ob;Rosa26^LSL-TdTomato* (*n* = 10 mice (5 male,5 female)) mice administered tamoxifen at 4 weeks of age were analyzed. Source data are provided as a Source Data file.

factors (TFs) (Fig. 5i). In contrast, β2 cells reflected an immature state in vehicle-treated mice, with decreased expression of insulin secretion and maturation genes. β-mSTZ cells displayed features of *Lep^ob/ob* AT 2 cells, exhibiting loss of β cell maturation markers but increased expression of *Cck* and dedifferentiation/immaturity, ER stress, hormone processing, and secretion markers (Fig. 5i).

Given this similarity, we sought to map β1, β2, and β-mSTZ cells directly to β cells in our obesity dataset for comparison across all transcriptional measurements. However, due to systematic technical variation from differences in cell handling in distinct batches (batch effects), the datasets were not directly comparable. To overcome this, we employed scMMGAN[26], a data integration approach that uses

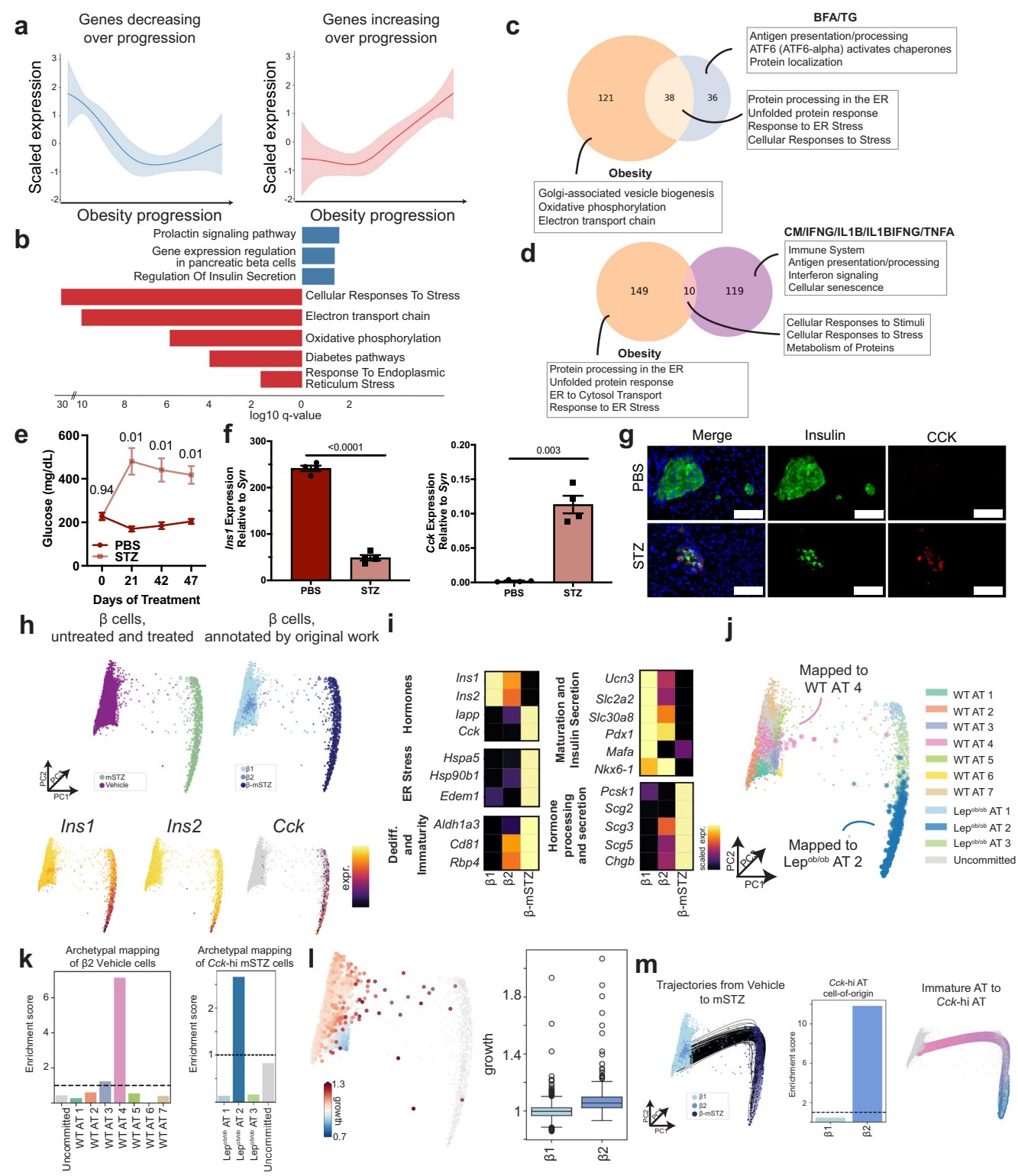

adversarial learning to align cells from different batches based on their data geometry. scMMGAN enables batch integration and preserves biological variation for direct comparison across diverse experimental setups and stressors. We used scMMGAN to learn the best mapping of vehicle-treated β cells (β1, β2) to WT β cell ATs and β-mSTZ cells to *Lep^ob/ob* β cell ATs using our trained AAnet models (Fig. 5j and Supplementary Data 4). The archetypal mapping of the vehicle-treated β cells revealed representation of all WT ATs, where the immature β2 state was enriched for WT AT 4 (Fig. 5j, k). β-mSTZ cells also

displayed representation of all *Lep^ob/ob* ATs, where *Cck*-hi (>95th percentile expression) STZ β cells were enriched for *Lep^ob/ob* AT 2 (Fig. 5j, k). Based on results in our obesity models (Fig. 3g, h), we hypothesized that *Cck*-hi β-mSTZ cells originate from immature β2 cells. To test this, we ran TrajectoryNet to trace back the most likely paths from β-mSTZ cells assigned to *Lep^ob/ob* AT 2 to the vehicle-treated β cells. As expected, TrajectoryNet-learned relative proliferation rates determined that β2 cells have a higher growth rate than β1 cells (Fig. 5l). TrajectoryNet further demonstrated that β2 cells – rather than β1 – were far more

**Fig. 5 | Physiologic (obesity) and pharmacologic stressors induce β cell CCK expression. a** Gene clusters derived from gene trends (mean scaled expression ± SD) to *Lep^ob/ob* AT 2, grouped into highly variable genes decreasing (*n* = 1137 genes) or increasing (*n* = 1141 genes). **b** Gene set enrichment analysis shows significantly (*q* < 0.05) enriched gene sets for each gene cluster. Red is upregulated. Blue is downregulated. **c** Gene set overlap with primary human islet cells treated with ER stress inducers (*p* = 1.21 × 10⁻⁴⁸, hypergeometric test). **d** Gene set overlap with primary human islet cells exposed to inflammatory cytokines (*p* = 7.88 × 10⁻⁵, hypergeometric test). **e** Serial random glucose (mean ± SEM, *n* = 4 male mice per group) of C57/B6 male mice treated with STZ or PBS (vehicle). *p*-values of two-sided repeated measures ANOVA with Tukey's post-hoc test are shown. **f** Final pancreatic *Ins1* and *Cck* expression (qRT-PCR, mean ± SEM, *n* = 4 male mice per group) normalized to islet marker synaptophysin (*Syn*) of mice in (**e**). *p*-values of two-sided Welch's *t*-test are shown. **g** Immunofluorescence of islet insulin and CCK expression of mice in (**e**). Images are representative of *n* = 4 mice per group. **h** Embedding of β cells (*n* = 6760 cells) from vehicle and STZ-treated conditions, colored by treatment, cluster annotation from original work[37], and marker gene expression (color

scale represents min to max of normalized UMI for *Ins1* and *Ins2* and 1-5.5 for *Cck*; gray denotes <1 normalized UMI). **i** Heatmap of marker gene expression (color scale represents min to max of mean normalized UMI, scaled per gene) for three β cell subclusters in (**h**). **j** Visualization of β cells (*n* = 6760 cells) in (**h**) mapped to all archetypes. **k** Enrichment scores (mean over *n* = 2 scMMGAN runs) for each WT archetype for β2 cells and each *Lep^ob/ob* archetype for *Cck*-hi mSTZ β cells. **l** TrajectoryNet-inferred proliferation for cells from vehicle condition (*n* = 5600 cells), where proliferation rate > 1 implies growth. Gray cells are all β cells (*n* = 6760 cells). TrajectoryNet-inferred proliferation rates calculated for β1 (*n* = 5531 cells) and β2 (*n* = 263 cells) cells in vehicle-treated mice. Box plots display 25, 50, and 75th percentiles ± 1.5 interquartile range (IQR). **m** Left: Visualization of trajectories from vehicle to β-mSTZ cells mapped to *Lep^ob/ob* AT 2 (*n* = 200 randomly sampled trajectories). All β cells visualized (*n* = 6760 cells), colored by annotation assignment. Middle: The cell-of-origin enrichment score (mean over *n* = 2 TrajectoryNet runs) is higher for β2 than β1. Right: TrajectoryNet-learned trajectories tracing paths from cells mapped to WT AT 4 to cells mapped to *Lep^ob/ob* AT 2 (*n* = 195 trajectories). Gray cells are all β cells (*n* = 6760 cells). Source data are provided as a Source Data file.

---

enriched within the cells-of-origin for *Cck*-hi β-mSTZ cells (Fig. 5m). These data argue that both physiologic (obesity) and pharmacologic (STZ) stressors induce immature β cell expansion and adaptation to a CCK-expressing state in vivo.

## The murine obesity trajectory correlates with type 2 diabetes in humans and mice

We next evaluated whether obesity mirrored other physiologic perturbations of β cell function, including age and type 2 diabetes (T2D). Specifically, we used scMMGAN to map β cells from previously published non-diabetic (ND) and T2D mouse scRNA-seq datasets[33] totaling 85,129 cells from 31 datasets integrated across conditions and developmental stages (Supplementary Fig. 12a and Supplementary Data 4). Along the obesity progression axis, *Ins1* and *Ins2* expression from the mouse reference cells decreased, and *Cck* expression increased (Supplementary Fig. 12b). Predominantly, cells from diabetes models (*db/db* and mSTZ) mapped to the latter end of the obesity progression, and cells from normal mice – including aged (≥20 months) mice – mapped to the WT region of the obesity progression (Supplementary Fig. 12c). These data suggest that diabetes induction, and not age, aligns to β cell adaptation in obesity.

To further refine this β cell mapping using our trained AAnet models, we annotated the archetypal assignment of each cell and calculated the proportion of each AT for all datasets (Supplementary Fig. 12d). All ATs were represented in at least one ND dataset and at least one T2D dataset, and most ATs were represented in multiple datasets. Additionally, datasets from the same condition showed highly similar archetypal proportions. T2D datasets were predominantly enriched for *Lep^ob/ob* ATs (Supplementary Fig. 12d). ND mice from different developmental stages showed differences in archetypal commitment. Embryonic ND samples had a significantly higher proportion of *Lep^ob/ob* AT 2 versus other samples (Supplementary Fig. 12e–g), suggesting *Cck*-hi cells result from dedifferentiation towards an embryonic state. In contrast, postnatal P16 mice (when β cells begin to mature) had a significantly higher proportion of WT AT 4 (Supplementary Fig. 12e–g), an immature state. 2-month-old mice were enriched for archetypal states representing an intermediate maturation phenotype (WT AT 3 and HFD AT 2); 4-6-month-old mice were enriched for a mature phenotype (WT AT 1,5,7 and HFD AT 1); and aged mice were enriched for intermediate maturation ATs (WT AT 2 and 6; Supplementary Fig. 12g). These data argue that aging alone does not mimic the effects of obesity.

To determine the relevance of our findings to human biology, we used scMMGAN to map five published human scRNA-seq datasets consisting of 15 ND and 9 T2D donors to all cells from WT, HFD-fed, and *Lep^ob/ob* mice (Supplementary Data 4). The mapped insulin (*INS*)

expression for human donors decreased along the obesity trajectory, recapitulating β cell changes along the mouse obesity progression (Fig. 6a). We annotated the archetypal assignment of each cell for each donor and calculated the proportion of each AT (Fig. 6b). All ATs were represented in at least one ND sample, and all ATs (except WT AT 2) were represented in at least one T2D sample. All ATs were found in both male and female donors, with most present in multiple ND and T2D donors (Supplementary Fig. 13). When comparing archetypal proportions between ND and T2D donors, mature β cells (WT AT 1) were notably more abundant in ND donors. In contrast, *Lep^ob/ob* AT 2 and *Lep^ob/ob* AT 3 cells were present in significantly greater proportions among T2D donors (Fig. 6c). To further assess the relationship between measured clinical variables in human donors (age, sex, BMI, and T2D status) and the obesity trajectory in our dataset, we leveraged the patient embedding approach DiffusionEMD[27]. As clinical variables are determined on the level of each donor, rather than each individual cell, it is not sufficient to compare the relationship between each variable and the cellular progression axis. DiffusionEMD overcomes this problem by using optimal transport to map an embedding of donors based on the distribution of each donor's cells on the obesity trajectory. That is, donors with similar cell distributions are embedded close together, and donors with very different cell distributions (*e.g.*, opposite sides of the obesity progression axis) are embedded far apart. The resulting DiffusionEMD embedding revealed a linear trajectory across donors, where donors on the left are mapped early on the obesity trajectory (high *INS*), and donors on the right are mapped later along the trajectory (low *INS*) (*r* = −0.93) (Fig. 6d). Visualizing clinical variables showed a low positive to negligible correlation between age (*r* = 0.28), BMI (*r* = −0.18), and sex (*r* = −0.17) (Fig. 6e) with the obesity progression axis. Instead, we observed much stronger correlations between obesity progression and HbA1c (*r* = 0.50) and T2D status (*r* = 0.49) (Fig. 6e). These results suggest that the β cell adaptations associated with obesity progression in mice are concordant with T2D across species, donors, and scRNA-seq datasets.

## JNK/cJun signaling modulates CCK expression in β cells

We next sought to determine the regulatory mechanisms that govern β cell *Cck* expression in obese mice. TFs previously associated with *Cck* regulation predominantly decreased in expression along the obesity trajectory except for *Jun*, *Fos*, and *Crem*, which increased (Fig. 7a). To predict which TFs regulate *Cck* expression, we leveraged *Cflows*, a framework that combines neural ODE networks with Granger causality to infer gene regulatory interactions from inferred continuous trajectories[28]. We first computed Granger causality scores between highly variable TFs and genes and subsequently pruned to TF-target interactions with high Granger causality and prior evidence for

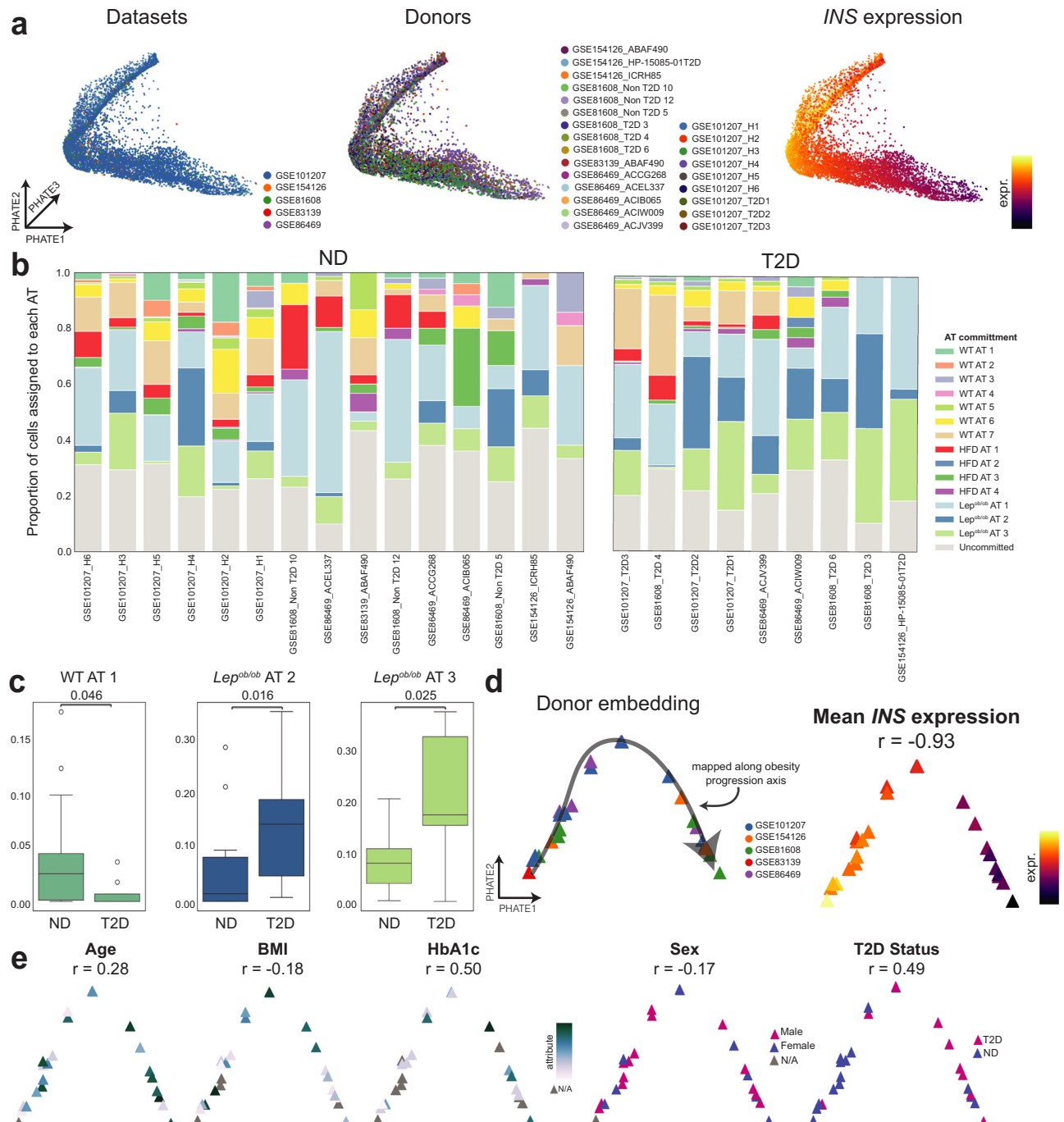

**Fig. 6 | Obesity progression in mouse latent-space archetypal analysis corre-lates with human type II diabetes. a** Embedding of human β cells ($n = 10{,}663$ cells) from non-diabetic (ND) and type II diabetic (T2D) donors, mapped onto the obesity progression with scMMGAN. Embedding colored by sample datasets, individual donors, and mapped *INS* expression (min to 99th percentile of nor-malized UMI). **b** Proportion of each archetype for each ND and T2D dataset. **c** Archetypal proportion is significantly different between ND ($n = 15$ samples) and T2D ($n = 9$ samples) datasets in (**b**) for WT AT 1, *Lep^{ob/ob}* AT 2, and *Lep^{ob/ob}* AT 3. Box plots display 25, 50, and 75th percentiles $\pm$ 1.5 interquartile range (IQR). *p*-values of two-sided Wilcoxon rank sum tests are shown. **d** Embedding of mapped patient

samples ($n = 24$ samples) by DiffusionEMD, where each triangle corresponds to an individual donor. Axis direction determined by mapping along obesity progression axis. Major axis of variation (quantified by Fiedler vector) negatively correlated (Pearson correlation analysis, $r = -0.93$) with mapped *INS* expression per donor (color scale denotes min to max of mean normalized UMI). **e** Patient sample embedding ($n = 24$ samples) colored by age, BMI, HbA1c, sex, and T2D status of donors. Gray corresponds to data not available. Pearson correlation analysis *r* values with Fiedler vector are shown, capturing association between clinical vari-ables and murine obesity progression. Source data are provided as a Source Data file.

transcriptional regulatory interactions in a reference database[63] (see Methods). We then subsetted to the top 100 regulatory TFs (based on Granger causality scores over all targets) and their targets with high Granger scores (Supplementary Data 5). For visualization, we further subsetted to only TF-target interactions where the TF and the target

both increased in expression along the obesity trajectory. This resulted in a subnetwork of 39 interactions across 36 TFs and targets, including *Cck* (Fig. 7b). The subnetwork was enriched ($q < 0.05$) for the AP-1 transcription factor network, including *Jun* and *Fos* directly connected to *Cck*, as well as *Egr1*, *Act1*, *Ccnd1*, and *Cdk1* within the subnetwork

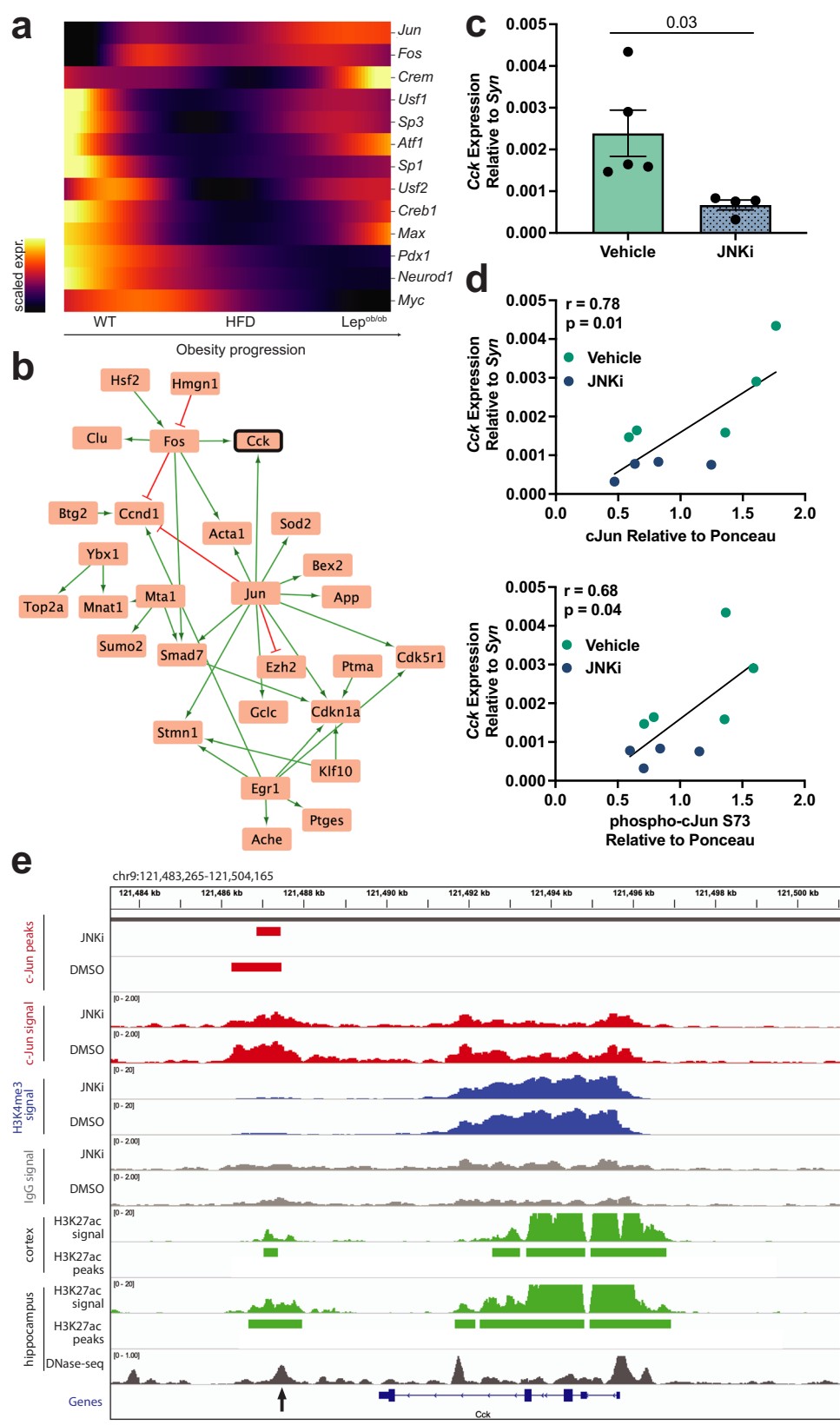

(BioPlanet). CCKR signaling map (Panther P06959, *e.g.*, *Cck*, *Jun*, *Fos*, *Clu*, *Itgb1*, and *Ccnd1*) was also significantly enriched (Supplementary Data 6).

*Jun* encodes the TF cJun, which is activated and stabilized by phosphorylation by cJun N-terminal kinases (JNKs). JNK/cJun signaling is a stress-responsive mitogen-activated protein kinase (MAPK)

pathway that can be induced by oxidative and ER stress[64,65], raising the possibility that β cell *Cck* transcription may be regulated by JNK/cJun signaling in vivo. Consistent with this hypothesis, JNK inhibition (JNKi) decreased pancreatic *Cck* expression in *Lep^{ob/ob}* mice proportional to the levels of total and phosphorylated cJun (Fig. 7c, d and Supplementary Fig. 14a). Similarly, we observed that JNKi can reduce *Cck*

**Fig. 7 | JNK/cJun signaling governs β cell CCK expression. a** Gene expression (color scale represents min to max of normalized UMI, scaled per gene) trends to *Cck+ Lep^ob/ob* cells for putative CCK transcription factors shows a near continuous increase in cJun expression. **b** In silico gene regulatory network derived by Granger causality analysis of the TrajectoryNet trajectories. Green edges correspond to activation, and red edges correspond to inhibition. **c** Pancreatic *Cck* expression (qRT-PCR, mean ± SEM) relative to islet marker synaptophysin (*Syn*) in 16-week-old *Lep^ob/ob* mice treated with JNKi (*n* = 4 mice (1 male, 3 female); 20 mg/kg SP-600125 daily) or vehicle control (*n* = 5 mice (2 male, 3 female)) for five days. *p*-value of two-sided Welch's *t*-test is shown. **d** Correlation of pancreatic *Cck* expression with cJun

and phospho-cJun S73 levels (relative to total protein (Ponceau); Pearson correlation analysis, *r* = 0.78, *p* = 0.01 for cJun, *r* = 0.68, *p* = 0.04 for phospho-cJun) in 16-week-old *Lep^ob/ob* mice treated with JNKi (*n* = 4 mice (1 male, 3 female)) or vehicle control (*n* = 5 mice (2 male, 3 female)) in (**c**). **e** CUT&RUN tracks (average of *n* = 2 biological replicates for each track) showing decreased cJun binding downstream of the *Cck* gene with JNKi (arrow). Significant peaks are shown by boxes. This cJun peak aligns with a putative mouse enhancer region (DNA hypersensitivity and H3K27Ac ChIP-seq peaks). Tracks were visualized with the Integrated Genomics Viewer (IGV). cJun binding at the *Cck* promoter is not significantly altered with JNKi. Source data are provided as a Source Data file.

expression in Min6 insulinoma cells in vitro (Supplementary Fig. 14b, c). To determine whether cJun directly regulates *Cck* expression, we performed cJun CUT&RUN analysis in Min6 cells, which revealed a reduction in overall peak number and intensity with JNKi (Supplementary Fig. 14d), consistent with reduced cJun levels (Supplementary Fig. 14b). As expected, cJun principally bound promoters or distal intergenic regions (Supplementary Fig. 14e), and the most frequent binding motif (27.2%, $p = 5.4 \times 10^{-18}$, $E = 3.8 \times 10^{-16}$, binomial test) matched the canonical AP-1 binding site (5′-TCA[GC]TCA-3′). cJun was enriched at a region ~3 kb downstream of the 3′ end of the *Cck* gene, the peak signal intensity of which was selectively reduced with JNKi (Fig. 7e). Data from ENCODE[66,67] derived from *Cck*-expressing tissues (mouse cortex and hippocampus) suggested that this region may be an enhancer marked by H3K27Ac and DNA hypersensitivity (Fig. 7e). Furthermore, analyses of human islet ATAC-seq[68], H3K27Ac ChIP-seq[68], and Hi-C[69] data confirmed that this putative enhancer region is conserved in humans and resides at the boundary of a chromatin loop (Supplementary Fig. 14f). This higher-order chromatin structure may provide an additional level of transcriptional regulation of *Cck*. These data indicate that *Cck* expression may be mediated by stress-responsive JNK-cJun signaling via a conserved 3′ distal enhancer.

### Functional β cell depletion suppresses pancreatic exocrine tumorigenesis in lean mice

Given the importance of β cells in obesity-associated PDAC development, we further tested whether functional β cell depletion could alter exocrine tumorigenesis even in lean mice. We crossed the rapidly progressive *KPC* (*Pdx1-Cre; Kras^LSL-G12D; Trp53^LSL-R172H/+*) PDAC model with *Akita* mice (Fig. 8a), which harbor a mutation in the *Ins2* gene that causes protein misfolding, ER stress, and, consequently, non-inflammatory β cell loss[70]. We confirmed decreased β cell mass and induction of ER stress in islets of *KPC-Akita* mice without changes in islet immune cell infiltration (Fig. 8b, c). *KPC-Akita* mice exhibited significantly reduced circulating insulin and C-peptide levels and early-onset hyperglycemia compared to *KPC* controls (Fig. 8d, e), consistent with functional β cell depletion. All *KPC* and *KPC-Akita* mice developed PDAC without evidence of liver metastatic disease at 12 weeks of age. Importantly, there were no stage-specific differences in proliferation, immune cell infiltration, or fibrosis comparing PanIN and PDAC lesions between these models (Fig. 8f). Nonetheless, *KPC-Akita* mice exhibited significantly reduced overall disease burden compared to age-matched *KPC* littermates (Fig. 8g), arguing that tumor initiation rather than progression was partially compromised. These data showcase a basal role for functional β cells in PDAC development, even in lean mice.

### Peri-islet acinar cell adaptation and enhanced tumorigenesis in obese mice

Finally, we sought to determine how β cell adaptations to obesity influence exocrine cells to drive tumor development by analyzing transcriptional changes in acinar cells, the putative cell-of-origin for PDAC[11,12,16]. We re-embedded scRNA-seq data for *Cpa1+/Prss2*+ cells and recovered 260 acinar cells that passed quality control metrics

from all samples (Supplementary Fig. 15a). We suspected that these cells were enriched in those found in the peri-islet area, as they were obtained from highly pure islet isolations and represented ~1% of sequenced cells. Consistent with this hypothesis, acinar cells from obese (HFD-fed and *Lep^ob/ob*) versus lean (WT) mice were enriched in gene signatures previously observed in peri-islet acinar cells of the leptin receptor-deficient (*Lepr^db/db*) obesity model[71,72] (Supplementary Fig. 15b). Differential expression analysis revealed increased expression of multiple proteases in acinar cells of HFD-fed and *Lep^ob/ob* mice (Supplementary Fig. 15c), including those (*Try4, Try5, Try10*) confirmed to be upregulated in peri-islet acinar cells in obese *Lepr^db/db* mice by single molecule fluorescence in situ hybridization (smFISH)[72]. Furthermore, a subset of these proteases (*Cela3b, Cpb1, Ctrb1*) were previously shown to be induced by CCK signaling in acinar cell models[72], raising the possibility that this may be a consequence of β cell-derived CCK.

We next clustered acinar cells into three subclusters, the proportions of which were altered by obesity (Fig. 9a, b). Obesity was associated with a decrease in *Tff2*+ acinar cells (cluster 1) (Fig. 9c, d and Supplementary Fig. 15d), which have been shown to be resistant to oncogenic *Kras*-induced transformation[73]. Conversely, obesity was associated with an increase in the proportion of acinar cells expressing higher levels of proteases (clusters 2 and 3) (Fig. 9c and Supplementary Fig. 15d). Strikingly, cluster 3 was uniquely present in obesity models (Fig. 9b), and cells within this cluster exhibited high expression levels of the Regenerating (*Reg*) gene family (Fig. 9c, e and Supplementary Fig. 15d). *Reg* expression was also observed in a published analysis of peri-islet acinar cells in human pancreas samples[74], the gene signature of which was enriched in cluster 3 versus cluster 1 (Fig. 9f), arguing that acinar cells from cluster 3 harbor features of human peri-islet acinar cells. *Reg* expression is associated with acinar cell stress and inflammation[75], and Reg proteins can promote ADM[76,77], making them both markers and drivers of a tumor-permissive acinar cell state. In concordance with our scRNA-seq data, bulk pancreata of obese *Lep^ob/ob* mice exhibited upregulation of *Reg* genes compared to lean congenic WT mice (Fig. 9g). Importantly, *Reg* expression was abolished by chronic CCK receptor antagonist treatment of *Lep^ob/ob* mice (Fig. 9g), indicating that cluster 3 cells may be induced by paracrine signaling from β cell-derived CCK in mice. Consistent with this hypothesis, we observed increased peri-islet acinar cell Reg2 protein expression in *KC;Lep^ob/ob;Ins1-Cck/+* and *KCO* mice compared to *KC;Lep^ob/+* controls (Fig. 9h). Together, these data argue for the potential of β cell-derived CCK to promote a tumor-permissive peri-islet acinar cell state.

We reasoned that these CCK-dependent alterations in peri-islet transcriptional states may enhance *Kras*-driven tumorigenesis in proximity to islets in obesity. In two-dimensional analyses of tissue sections, we found that greater proportions of PanINs in obese *KCO* mice emerged in proximity to islets than in lean *KC;Lep^ob/+* mice (Fig. 10a). To confirm whether the spatial relationships between islets and neoplasia were preserved in three dimensions (3D), we employed tissue clearing and immunofluorescence, coupled with 3D light-sheet microscopy imaging of intact pancreata of obese *KCO* mice and lean

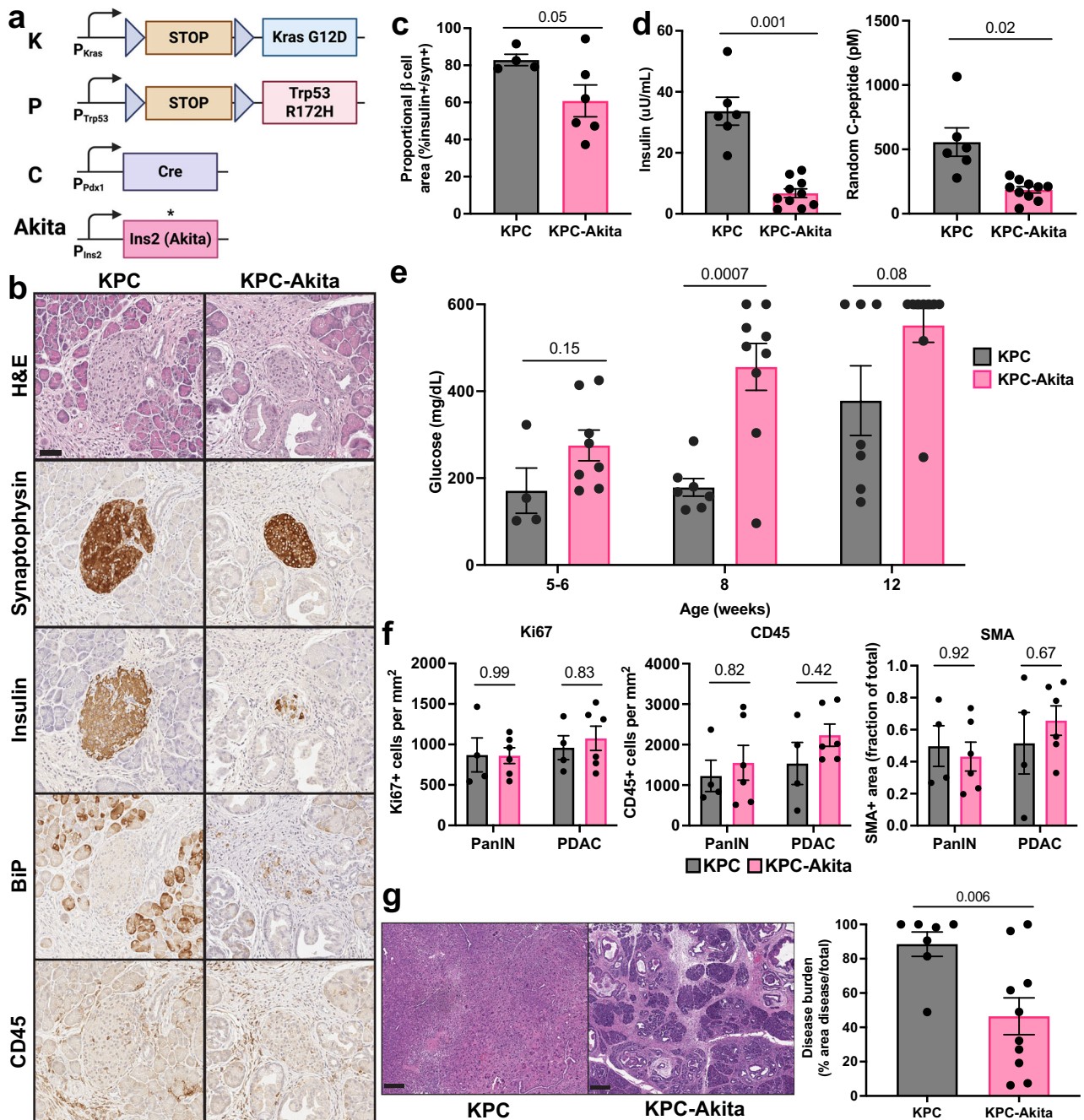

**Fig. 8 | Functional depletion of β cells suppresses pancreatic tumorigenesis.**
**a** Schematic of the alleles used to generate *KPC* and *KPC-Akita* mice. Created in BioRender. McQuaid, D. (2026) https://BioRender.com/m38ppei. **b** H&E and IHC stains show a reduction in the proportion of insulin-expressing islet cells and increased ER stress (BiP) without alterations in CD45+ immune cell infiltration of islets (labeled by pan-neuroendocrine marker synaptophysin) of *KPC-Akita* mice. Images are representative of *n* = 4 *KPC* and *n* = 6 *KPC-Akita* 12-week-old mice. Scale bar is 50 μm. **c** Quantification of proportional β cell area (insulin+) of total islet area (synaptophysin+) (mean ± SEM) in 12-week-old *KPC* (*n* = 4 (2 male, 2 female)) and *KPC-Akita* (*n* = 6 (3 male, 3 female)) mice. *p*-value of two-sided Welch's t-test is shown. **d** Terminal non-fasting insulin and C-peptide levels (mean ± SEM) for 12-week-old *KPC* (*n* = 6 (3 male, 3 female)) and *KPC-Akita* (*n* = 10 (5 male, 5 female)) littermates. *p*-values of two-sided Welch's *t*-tests are shown. **e** Serial non-fasting blood glucose measurements (mean ± SEM) of 12-week-old *KPC* (*n* = 4 (2 male,

2 female), 5-6 weeks; *n* = 7 (3 male, 4 female), 8 weeks; *n* = 7 (4 male, 3 female), 12 weeks) and *KPC-Akita* (*n* = 8 (4 male, 4 female), 5-6 weeks; *n* = 9 (5 male, 4 female), 8 weeks; *n* = 9 (4 male, 5 female), 12 weeks) littermates. Maximum glucometer measurement is 600 mg/dL. *p*-values of mixed-effects model with Geisser-Greenhouse correction and Tukey's post-hoc test are shown. **f** Quantification of Ki67+ tumor cells, CD45+ immune cells, and smooth-muscle actin (SMA)+ fibroblast area (mean ± SEM) in fields of view containing PanIN and PDAC lesions of 12-week-old *KPC* (*n* = 4 (2 male, 2 female)) and *KPC-Akita* (*n* = 6 (3 male, 3 female)) mice. *p*-values of two-way ANOVA with Šidák's post-hoc test are shown for each marker. **g** Representative H&E images of *KPC* and *KPC-Akita* mice and quantification of tumor burden (mean ± SEM) in 12-week-old *KPC* (*n* = 7 (4 male, 3 female)) and *KPC-Akita* (*n* = 10 (5 male, 5 female)) littermates. *p*-value of two-sided Welch's *t*-test is shown. Scale bar is 250 μm. Source data are provided as a Source Data file.

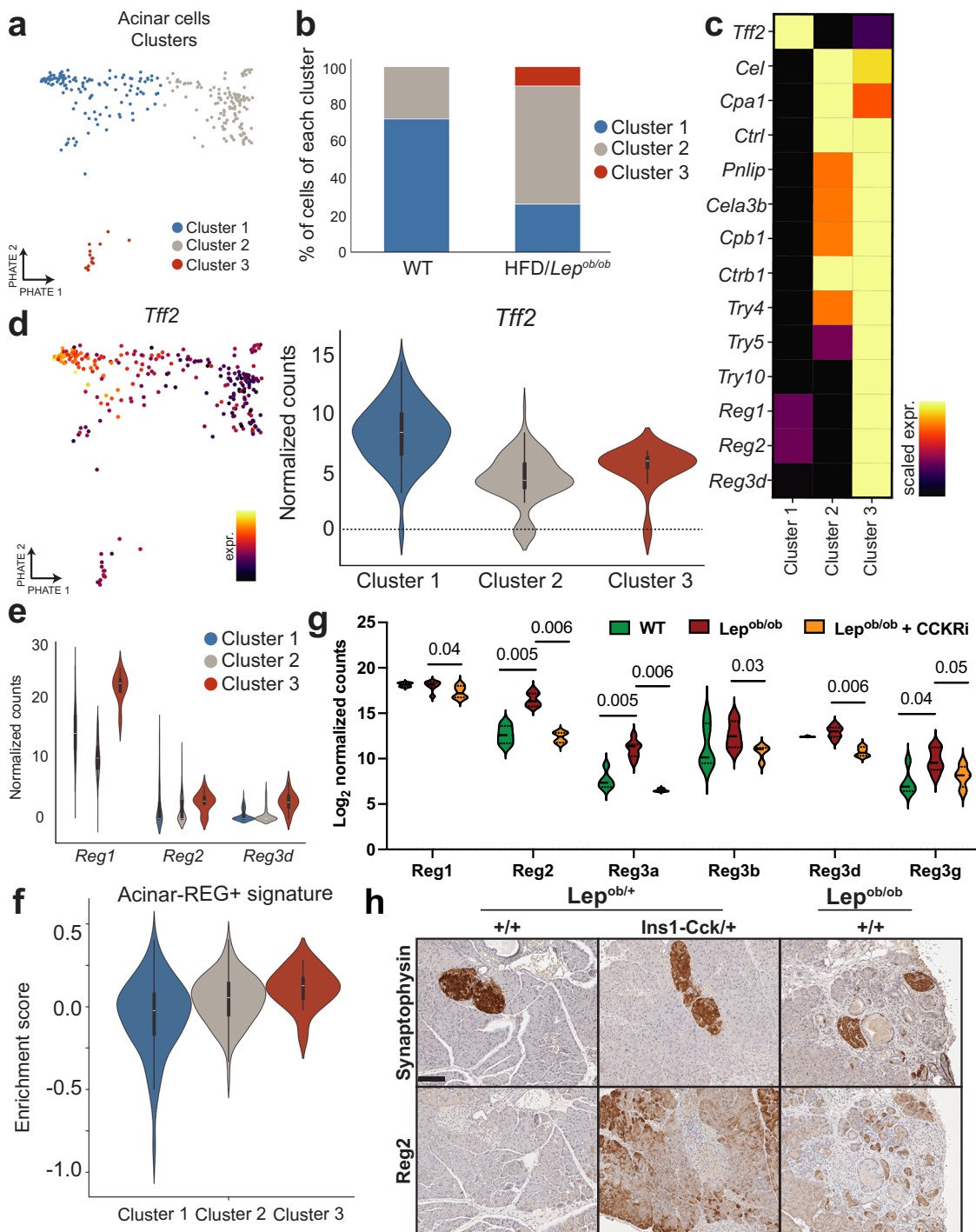

**Fig. 9 | CCK-dependent peri-islet acinar cell transcriptional alterations in obesity. a** PHATE visualization of **k**-means clustering of acinar cell clusters (n = 260 cells). **b** Differential proportions of acinar cell clusters in (**a**) in lean (WT) vs. obese (HFD/*Lep^ob/ob*) conditions. *p* < 0.0001, chi-square test. **c** Heatmap of *Tff2*, protease, and *Reg* gene expression (color scale represents min to max of mean normalized UMI, scaled per gene) for each cluster in (**a**). **d** PHATE visualization (n = 260 cells) of *Tff2* gene expression (color scale represents min to max of normalized UMI) and distribution (violin plots showing min/max with 25th, 50th, and 75th percentiles delineated by lines) in acinar cell clusters (cluster 1 *n* = 121 cells, cluster 2 *n* = 124 cells, cluster 3 *n* = 15 cells). **e** Violin plots of *Reg* gene expression distribution (min/max with 25th, 50th, and 75th percentiles delineated by lines) in acinar cell clusters. Cell numbers same as (**d**). **f** Violin plot of enrichment score distribution (min/max with 25th, 50th, and 75th percentiles delineated by lines) of the human peri-islet

acinar-REG+ differential gene expression signature derived from Tosti et al.[74] for each murine acinar cell cluster. Cell numbers same as (**d**). **g** Violin plots of pancreatic *Reg* gene expression (RNA-seq) distribution (min/max with 25th, 50th, and 75th percentiles delineated by lines) of 12-week-old WT (*n* = 5 mice), *Lep^ob/ob*) (*n* = 7 mice), and *Lep^ob/ob* mice treated with CCK receptor antagonists (*n* = 4 mice (*n* = 2 each with proglumide or lorglumide)) for 6 weeks. *q*-values of Wilcoxon rank sum test with two-stage Benjamini, Krieger, Yekutieli step-up are shown. **h** IHC images demonstrate increased peri-islet (synaptophysin-positive) acinar cell expression of Reg2 in 3-month-old β cell CCK-expressing tumor models (*KC;Lep^ob/+;Ins1-Cck/+* and *KCO*) compared to controls (*KC;Lep^ob/+*). Images are representative of *n* = 10 islets analyzed for *n* = 2 mice per genotype. Scale bar, 100 μm. Source data are provided as a Source Data file.

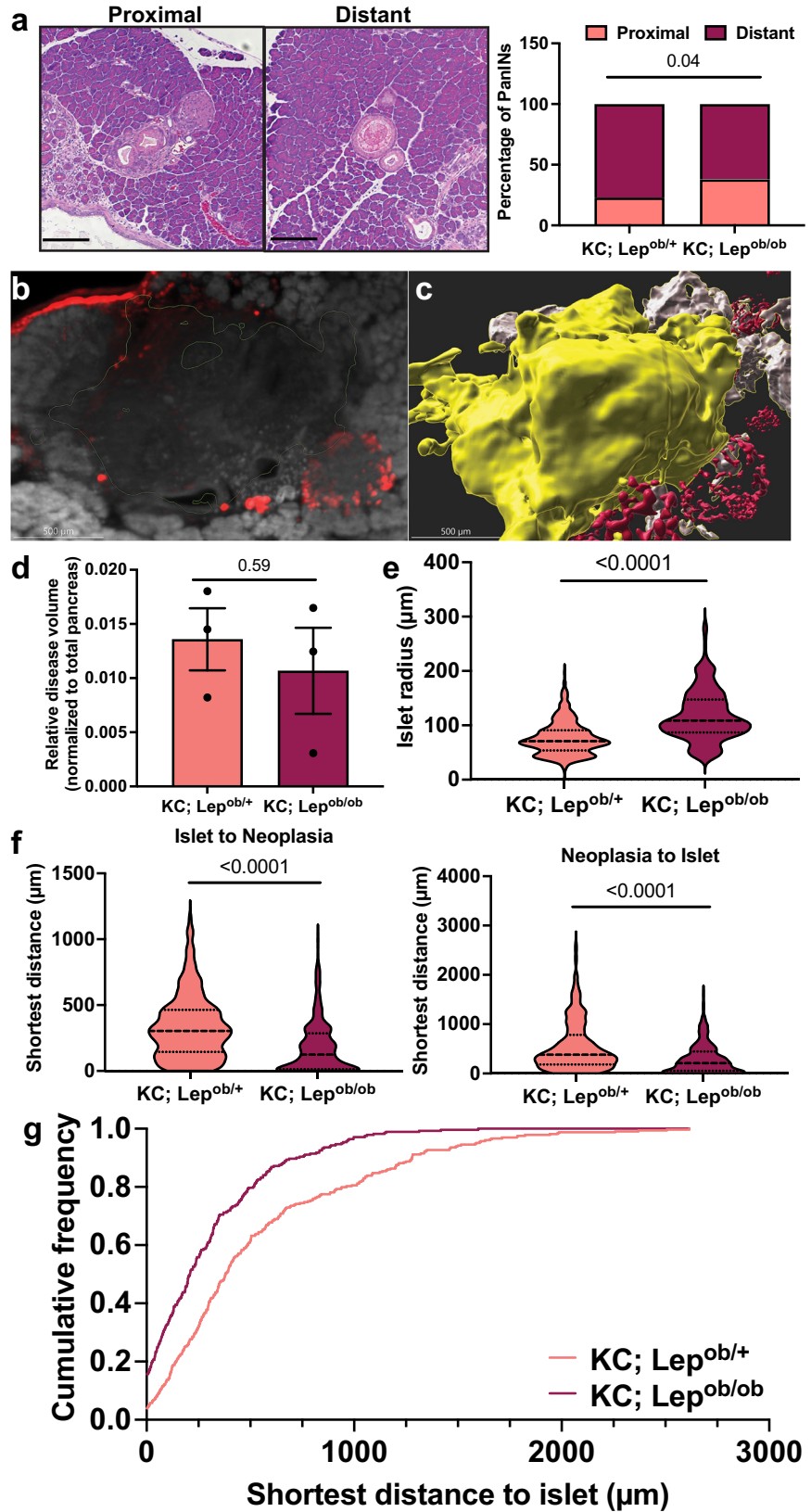

*KC;Lep^{ob/+}* controls at timepoints with comparable disease burden (8- and 12-weeks-old, respectively; Fig. 10b and Supplementary Movies 1, 2). 3D reconstructions of the tissue (Fig. 10c) identified 329 tumors in lean *KC;Lep^{ob/+}* mice and 281 in obese *KCO* mice with no significant difference in relative disease volume (Fig. 10d). Although islet radius was significantly larger in *KCO* mice (Fig. 10e), as previously reported

for obese mice[78], fewer total islets were identified (271 vs. 444 in *KC;Lep^{ob/+}* mice). In support of islet proximity dependent tumorigenesis, both the neoplasia-to-islet and islet-to-neoplasia distances were significantly shorter in obese *KCO* mice (Fig. 10f). Cumulative frequency analysis confirmed a higher proportion of islets located closer to neoplastic lesions in obese mice (Fig. 10g), supporting endocrine-

**Fig. 10 | Enhanced islet proximal neoplasia in obese mice. a** Representative images and percentage of islet proximal (<400 μm) or distant (>400 μm) PanINs in lean 13-week-old *KC; Lep^{ob/+}* (*n* = 146 islets from *n* = 5 mice) and obese 7-week-old *KC; Lep^{ob/ob}* (*KCO*; *n* = 58 islets from *n* = 4 mice). *p*-value of two-sided Fisher's exact test is shown. Scale bars are 100 μm. **b** Light sheet microscopy 2D projection of a pancreatic neoplastic lesion (outlined in green) in an 8-week-old *KCO* mouse proximal to an islet (red). **c** 3D reconstruction of image neoplastic lesion and islet in (**b**). **d** Relative neoplastic disease volume normalized to total pancreas volume (mean ± SEM) of lean 12-week-old *KC; Lep^{ob/+}* (*n* = 3 mice) and obese 8-week-old *KC; Lep^{ob/ob}* (*KCO*; *n* = 3 mice) mice. *p*-value of two-sided Welch's *t*-test is shown. **e** Violin distribution plots (min/max with 25^{th}, 50^{th}, and 75^{th} percentiles delineated by lines) of the radius of islets (*n* = 444 islets from *n* = 3 *KC; Lep^{ob/+}* mice and *n* = 271 islets from *n* = 3 *KCO* mice). *p*-value of two-sided Wilcoxon rank sum test is shown. **f** Violin distribution plots (min/max with 25^{th}, 50^{th}, and 75^{th} percentiles delineated by lines) of the shortest islet-to-neoplasia (*n* = 444 islets from *n* = 3 *KC; Lep^{ob/+}* mice and *n* = 271 islets from *n* = 3 *KCO* mice) and neoplasia-to-islet (*n* = 329 neoplastic lesions from *n* = 3 *KC; Lep^{ob/+}* mice and *n* = 271 neoplastic lesions from *n* = 3 *KCO* mice) distances. *p*-values of two-sided Wilcoxon rank sum test are shown. **g** Cumulative distribution frequencies of the shortest neoplasia-to-islet distances of lesions in (**f**). The cumulative distributions are significantly different between groups (*p* < 0.0001, Kolmogorov-Smirnov test). Source data are provided as a Source Data file.

exocrine CCK signaling in acinar cell adaptation and obesity-associated pancreatic tumorigenesis.

## Discussion

Although other cellular components of the pro-tumorigenic PDAC microenvironment – immune, fibroblast, endothelial, and nerve cells – have received greater attention[79], β cells are emerging as essential drivers of PDAC progression. Here, we show that functional depletion of β cells (*Akita*) slows tumorigenesis even in the rapidly progressive *KPC* PDAC model (Fig. 8). Our results – based on more faithful genetic PDAC models – mirror findings in Syrian hamsters in which pharmacologic β cell ablation limits carcinogen-induced (N-nitroso-bis (2-oxopropyl) amine (BOP)) PDAC development[80]. They further confirm a conserved basal role of β cells in exocrine tumorigenesis, even in lean mice. Our findings are consistent with recent studies demonstrating that reducing insulin (via partial *Ins1/2* knockout) and endocrine-exocrine insulin signaling (via acinar-specific *InsR* knockout) impedes oncogenic *Kras*-driven tumorigenesis[13–15]. Moreover, we disentangle the complex interplay between hyperglycemia, β cell function, and cancer. Human PDAC tumors are largely FDG-PET avid, consistent with efficient glucose uptake[81], and glucose metabolism is thought to be important for PDAC progression[82,83]. However, we found that *KPC-Akita* mice harbored less disease burden despite early hyperglycemia. Our data argue that, without functionally replete β cells, glucose is not the primary driver of PDAC development, lending support to the significance of β cells in integrating host metabolic features to drive pancreatic tumorigenesis. Despite the importance of basal insulin secretion in pancreatic tumorigenesis, β cell CCK appears to be a more significant driver of obesity-associated PDAC development. In contrast to insulin, CCK expression strongly correlated with tumor development in obese mice, and β cell-specific CCK overexpression was sufficient to mimic the impact of obesity on tumorigenesis (Fig. 1). Conversely, β cell *Cck* knockout completely abrogated the effect of obesity on PDAC progression (Fig. 2). Importantly, genetic *Cck* manipulation did not significantly alter insulin secretion nor glucose homeostasis. Together, these findings argue that β cell CCK is the primary driver of obesity-associated PDAC progression, countering conventional wisdom on the essential roles of glucose and insulin in obesity-driven cancer.

Our study also provides insights into the cellular and molecular mechanisms that govern β cell adaptation to obesity, which leads to a pro-tumorigenic CCK-expressing state. We leveraged a suite of computational methods (TrajectoryNet, AAnet, scMMGAN, DiffusionEMD) for scRNA-seq data analysis to characterize β cell heterogeneity and transcriptional dynamics with increasing obesity (Supplementary Fig. 1). Additionally, we showed the utility of the *Cflows* framework for uncovering gene regulatory networks from inferred cellular trajectories. These approaches, used in conjunction, represent a significant advance in single-cell analysis to attain unique biological insights, especially in the context of β cell biology. Standard practices in single-cell analysis characterize cellular states based on clustering and pseudotime inference[36]. Indeed, the majority of published single-cell analyses of mouse and human pancreatic islets in normal physiology

and diabetes involved unsupervised clustering to identify endocrine and exocrine cell types, then subclustering or identifying factors of variation within cell types of interest, including β cells[84–88]. However, these analyses are limited in their ability to infer cells-of-origin, gene regulatory relationships, and transcriptional dynamics within and across cell types, analyses enabled by TrajectoryNet through calculation of cell-specific trajectories. Furthermore, unlike subclustering, latent-space archetypal analysis (AAnet) avoids arbitrarily discretizing the continuous state space and losing gene signals and behavior that result from high plasticity and transformation between states[25]. By mapping published datasets directly to our obesity progression through batch integration, scMMGAN established that β cell ATs in the context of obesity are highly related to β cell states emerging in the context of diverse stressors across the entire transcriptome, beyond marker genes or β cell-specific signatures. The high confidence of scMMGAN mappings across distinct conditions, developmental stages, and species provides in silico validation of our identified ATs. Finally, while prior work has shown the relationship between marker gene expression and measured clinical variables[84], DiffusionEMD instead characterized broad similarity between patients based on all β cell transcriptomic measurements, enabling more accurate correlations of the murine obesity progression with clinical variables. Our single-cell analytic pipeline therefore enables avenues of β cell characterization not possible with existing methodologies.

Beyond its well-described roles in satiety and digestion, CCK has also been shown to be a survival factor for β cells against diverse stresses (obesity, STZ, cytokines)[29,30,32,89]. Thus, it is not surprising that CCK induction occurs in response to both physiologic (obesity) and pharmacologic (STZ) β cell stressors (Fig. 5). Previous work has shown that CCK is expressed during embryonic β cell development and implicated JNK3/cJun signaling in β cell proliferation during normal early postnatal development[90]. Leveraging the unique capabilities of AAnet to discover rare subpopulations within heterogenous β cells, we defined the cell-of-origin of *Cck* + β cells tracing them back to a single rare AT (WT AT 4) that resembles a recently described regenerative Ucn3-/Ngn3+ immature β cell type that persists into mouse adulthood[91,92]. Our data argue that obesity reverts β cells to a dedifferentiated immature state that aligns with human and mouse T2D datasets (Fig. 6) and that JNK/cJun signaling in β cells may also be co-opted to respond to obesity-associated stress. These observations suggest that obesity engages developmental transcriptional and signaling programs in β cells, which may contribute to pathogenic disease states, including cancer.

Finally, increased CCK and the concurrent reduction of insulin in obesity may be protective mechanisms analogous to what is observed with acinar cells under stress. Despite distinct physiologic functions, the endocrine and exocrine compartments of the pancreas adopt convergent paths in handling cellular stresses by downregulating their major protein products (insulin for β cells and proteases for acinar cells) and converting cellular states (β cell dedifferentiation and acinar-to-ductal metaplasia). Importantly, both of these cellular transitions serve to augment PDAC risk through dysregulated pro-tumorigenic

hormone secretion (*e.g.*, CCK) or by generating a permissive state (ADM) for oncogenic *Kras*-driven transformation[16,18,21,93,94]. Thus, self-preservation of the pancreas may pose an unanticipated threat to the organism at large and speaks to the unique interconnected nature of the endocrine and exocrine compartments and their role in cancer pathogenesis. Importantly, unlike whole animal *Cck* knockout[29], pancreas-specific CCK loss did not compromise insulin secretion nor glucose homeostasis in *Lep^{ob/ob}* mice. This difference speaks to the capacity of other sources of CCK (*e.g.*, small intestines) to maintain β cell function and unlocks a potential therapeutic window for β cell CCK targeting as a strategy to intercept PDAC progression without compromising islet function. Taken together, our research lays the groundwork for understanding the stress and signaling pathways that govern pro-tumorigenic β cell CCK expression and argue that targeting these endocrine β cell pathways may be an effective means for the prevention or interception of exocrine tumorigenesis.

There are several important limitations of our study. First, most of the functional and molecular analyses were performed in mouse models of obesity and PDAC. We previously observed an association between BMI and *CCK* expression in human islets[18] and show parallels between murine β cell states and human β cell heterogeneity (Fig. 6). Though these correlative analyses would argue that similar obesity-driven mechanisms of β cell adaptation and tumorigenesis may apply in humans, they are not definitive. Second, experimental lineage tracing studies may not be conclusive due to the lack of specific cell type markers and the potential for unforeseen leakiness of *Cre^{ERT}* lines. For example, to label WT AT 4 cells, we performed lineage tracing with the most readily available line with the greatest specificity for this AT, *Ngn3-Cre^{ERT2}*. Prior studies in older lean adult mice suggest that Ngn3+ β cells are rare and do not contribute significantly to β cell neogenesis during homeostasis[95]. In contrast, we observed that obesity is associated with expansion and adaptation of Ngn3+ β cells to express CCK (Fig. 4). However, we cannot exclude the possibility that we are labeling a larger islet population than WT AT 4 with this *Cre^{ERT}* line. Nonetheless, our findings are supported by orthogonal in silico analyses (Fig. 3) and are concordant with a recent lineage tracing study in mice fed a high-fat fast mimicking diet which showed expansion of Ngn3+ cells to comparable levels as we observed in *Lep^{ob/ob}* mice at similar timepoints[49]. These data argue that obesity results in the expansion of Ngn3+ cells independent of model or diet. Finally, how β cell CCK promotes exocrine tumorigenesis remains unclear. Murine acinar cells express CCK receptors and can release digestive enzymes in direct response to physiologic CCK stimulation, whereas excess stimulation can promote ADM[19,21]. However, whether human acinar cells express CCK receptors and respond directly to CCK stimulation remains controversial[96,97]. Both in mice and in humans, CCK can promote acinar cell enzyme secretion indirectly via stimulation of nerves that result in cholinergic input to the pancreas[98,99]. To what degree these direct and indirect signaling pathways govern the pro-tumorigenic effects of β cell CCK remains to be determined. Furthermore, whether CCK has differential impacts on PDAC that arises from transformation of acinar cells (via ADM) or ductal cells – which have different requirements and molecular pathways for progression to PDAC[100,101] – is not known. Future studies should define the precise mechanisms by which pancreatic endocrine-derived CCK drives exocrine tumorigenesis, which may reveal additional targetable nodes for PDAC prevention or interception.

## Methods

All research performed in this study complies with relevant ethical regulations including animal studies approved under the Yale University Institutional Animal Care and Use Committee (IACUC) protocol #20170 and Yale Environmental Health and Safety recombinant DNA protocol #17-35.

## Animal studies

Mice were housed in a specific-pathogen free (SPF) facility at room temperature and humidity with standard 12-hour day/night cycles. *Akita* (Stock #003548), *Kras^{LSL-G12D}* (Stock #008179), *Lep^{ob/ob}* (Stock #000632), *Ins1-Cre^{ERT}* (Stock #024709), *Gcg-Cre^{ERT2}* (Stock #042277), *Ngn3-Cre^{ERT2}* (Stock #028365), *Pdx1-Cre* (Stock #014647), *Rosa26^{LSL-TdTomato}* (Stock #007914), *Pgk1-FlpO* (Stock #011065), and *TrpS3^{LSL-R172H}* (Stock #008652) mice were obtained from the Jackson Laboratory (JAX). *Ins1-Cck* mice were a gift from D.B. Davis (University of Wisconsin). Cryorecovery of *Gcg-Cre^{ERT2}* and *Ngn3-Cre^{ERT2}* embryos was performed by the Yale Genome Editing Center (YGEC). CCK tm2a mice (EuCOMM) sperm was obtained from Riken, and mice were generated by in vitro fertilization (IVF) by the YGEC. *Cck^{flox}* mice were generated by crossing progeny with *Pgk1-FlpO* strain, which was later crossed out. *KPC* (± *Akita*), *KCO* (± *Ins1-Cck* or *Cck^{flox}*), and *Cre^{ERT};Lep^{ob/ob};Rosa26^{LSL-TdTomato}* mice were generated by breeding. Genotyping of the mice was done via PCR using genomic DNA from the tail or ear as a template. Tail DNA was isolated using Hotshot extraction, and PCR was run using GO Taq Green Mastermix (Promega). Primers and protocols used were obtained from JAX or previously described[18] with the exception of *Cck^{flox}* mice for which the following primers were used: forward primer: 5'-ATGGGCCCACACTTAAAACG-3', reverse primer: 5'-AGCCAAGCGAAGTTCCTAGA-3'. Primers were obtained from the W.M. Keck Biotechnology Resource Laboratory at Yale. For *Lep^{ob/ob}* breedings, *Lep^{ob/ob}* mice were administered AAV2/1-CAG-mLeptin via intramuscular injection once, which restores fertility[18]. For bulk RNA-seq experiments, *Lep^{ob/ob}* and congenic age-matched C57/B6 WT mice (Stock #000664) were purchased from JAX. *Lep^{ob/ob}* mice were treated with CCK receptor antagonists (proglumide (0.1 mg/mL; Sigma) or lorglumide sodium salt (0.05 mg/mL; Sigma)) or control drinking water continuously for 6 weeks starting at 6 weeks of age. Mice were euthanized at 12 weeks of age and isolated pancreata were snap frozen for RNA-seq analysis, as described below. *Cre^{ERT}* mice were induced at ~4 weeks of age with 8 mg/40 g (obese *Lep^{ob/ob}*) or 6 mg/40 g (non-obese *Lep^{ob/+}*) of tamoxifen (Sigma) in corn oil via oral gavage for 4 consecutive days. Mice were euthanized 1- or 12-weeks following start of tamoxifen administration for analysis. For STZ experiments, male mice were treated with 50 mg/kg per day of STZ or PBS (vehicle) by intra-peritoneal injection for 5 consecutive days and analyzed 7 weeks afterwards. For JNKi experiments, 17-week-old *Lep^{ob/ob}* mice were treated with either 20 mg/kg SP-600125 (MedChemExpress) or vehicle control (5% DMSO, 40% PEG-300, 5% Tween 80, 50% ddH$_2$O) by oral gavage once daily for five days prior to dissection. All mice were fed standard chow (Inotiv, 2018S) *ad libitum* except for mice fed high-fat diet (HFD), which received a 60% kcal lard-based HFD (Research Diets, D12492) for 10 weeks. Mouse weights were measured using a small animal scale, and blood glucose was measured using a OneTouch Ultra2 glucometer. Serum was collected at time of dissection under fed conditions unless otherwise noted. For fasted conditions, food was removed from the cage for 6 hours prior to sample acquisition. Serum C-peptide was measured by ELISA (Crystal Chem 90050) according to manufacturer's instructions. Serum insulin was measured by radioimmunoassay (RIA, Linco RI-I3K) by the Yale Diabetes Research Center. Mice for *KCO* tumor studies were kept in a hybrid C57/B6:129 background. Mice for *KPC* tumor studies were kept in a C57/B6 background. Specific endpoints are listed in the figure legends and were selected based on prior studies on tumor progression in these models[18,102]. Sibling littermates were used as controls in all tumor studies. The maximal tumor size permitted was 1 cm³, and this was not exceeded in the study. Mice of both sexes were used for all experiments except for the STZ and scRNA-seq experiments for which only male mice were used due to reduced β cell effects of STZ and obesogenic effects of HFD, respectively, in female mice.

## Cell culture

Min6 cells (at a passage of 31-50, obtained from Dr. G. Cline) were grown under high glucose DMEM (Corning) supplemented with 15% fetal bovine serum (FBS; ThermoFisher Scientific), penicillin-streptomycin (1%; ThermoFisher Scientific), L-glutamine (2 mM; ThermoFisher Scientific), sodium pyruvate (1 mM; ThermoFisher Scientific), and 2-mercaptoethanol (0.05 mM; Sigma). Cells tested negative for mycoplasma by PCR performed by the Yale Molecular and Serological Diagnostics Core.

## Immunoelectron microscopy

Cultured Min6 cells were fixed in 0.1% glutaraldehyde and 4% paraformaldehyde (PFA) in 0.1 M sodium cacodylate buffer (pH 7.4) for 30 minutes at room temperature, followed by 4% PFA in the same buffer for one hour. After buffer rinse, cells were scraped off the culture dish and gently centrifuged. Pelleted cells were resuspended in 10% gelatin and submerged in 2.3 M sucrose in PBS at 4 °C overnight prior to freezing in liquid nitrogen. 60-nm thick sections were cut using a Leica EM UC6/FC6 cryo-ultramicrotome and collected on nickel grids. After warming to room temperature, sections were immunolabeled with primary antibodies for 30 minutes, followed by 30-minute incubation with immunoglobulin G antibodies coupled to 5 nm and 10 nm protein A-gold particles (Utrecht University Medical Center, Netherlands). Specific antibodies used are listed in Supplementary Table 1. Images were acquired using a FEI Tecnai Biotwin transmission electron microscope (80 kV) and an AMT NanoSprint15 cMOS camera.

## Western Blot

Mouse pancreata were snap-frozen in liquid nitrogen and manually pulverized using the BioPulverizer (BioSpec Products 59012MS). Pancreatic tissue (~40 mg) and Min6 whole cell lysates were harvested in RIPA buffer (Pierce), EDTA (1:100) (ThermoFisher Scientific), and Halt protease and phosphatase inhibitor cocktails (1:100) (ThermoFisher Scientific). Tissue samples were vortexed and homogenized using a pellet mixer (VWR) and rotated at 4 °C for one hour prior to centrifugation at maximum speed for 20 minutes. Cells were scraped and placed on a rotator at 4 °C for 15 min and then spun as above. Protein concentrations were measured using the BCA Protein Assay Kit (Pierce) following the manufacturer's protocol. Equal amounts of protein (30 μg) were loaded to each well of mini-PROTEAN 4-20% TGX stain-free precast gels (Bio-Rad). Protein was transferred to nitrocellulose membranes using the TransBlot Turbo Transfer System (Bio-Rad). For pancreatic tissue lysates, membranes were stained with Ponceau S (0.5% (w/v) in 1% glacial acetic acid) to verify equal protein loading and then washed with PBS-T (PBS, Tween-20 (0.1%)). Membranes were blocked using Odyssey Blocking Buffer (Licor) in PBS (1:2) for one hour, then incubated with primary antibodies diluted in Odyssey Blocking Buffer and PBS-T (1:2) overnight at 4 °C. Membranes were washed for 10 minutes with PBS-T three times and then incubated in DyLight secondary antibodies (Cell Signaling Technologies) for one hour at room temperature. Membranes were washed with PBS-T for 10 minutes, three times each, prior to imaging. Specific antibodies used are listed in Supplementary Table 1. Membranes were imaged using the ChemiDoc Touch Imaging System (Bio-Rad). Band intensity was quantified using ImageLab software.

## RNA Extraction, qRT-PCR, and bulk RNA-sequencing

Mouse pancreata were snap-frozen in liquid nitrogen and manually pulverized using the BioPulverizer (BioSpec Products 59012MS). RNA was extracted from pulverized tissue (20-40 mg) and Min6 cells, homogenized via QIAshredder column (Qiagen), and purified using the Qiagen Mini RNeasy kit following the manufacturer's protocol with DNAse I treatment to remove any genomic DNA contamination. RNA quality and quantity was assessed using an Agilent Bioanalyzer, and only high-quality RNA samples (RIN > 7) were used in downstream analysis. 1 μg of RNA was used to synthesize cDNA using the Applied Biosystems™ High-Capacity cDNA Reverse Transcription Kit. cDNA was diluted in nuclease-free water (1:5), and qPCR was run using Taq-Man™ Universal PCR Master Mix and FAM probes (ThermoFisher Scientific) for *Gapdh* (Mm99999915_g1), *Ins1* (Mm01950294_s1), *Syn* (Mm00436850_m1), and *Cck* (Mm00446170_m1). qPCR assays were run on a CFX Opus 384 Real-Time PCR System (Bio-Rad). RNA-seq libraries were prepared with polyA selection (Illumina) and sequenced (>25 M 100-bp paired-end (2×100) reads on NovaSeq (Illumina)) with the Yale Center for Genome Analysis (YCGA). Low-quality bases and adapter sequences were trimmed using Trim Galore (v0.6.7) and CutAdapt (v3.5). Cleaned reads were aligned to the UCSC mouse genome (mm10) using the STAR aligner (v2.7.9). Differential gene expression analysis was performed using DESeq2 in R, and genes with an adjusted *p*-value < 0.05 were considered differentially expressed.

## CUT&RUN analysis

Min6 cells were plated (2.5 million) and grown for a total of three days. The day after plating, cells were treated with the JNK inhibitor SP600125 (20 μM, MedChemExpress) or DMSO (control) for 48 hours. 550,000 cells were harvested per reaction and condition. Cells were processed using the CUTANA ChIC/CUT&RUN Kit (14-1048), JUN/cJun CUTANA CUT&RUN antibody (SKU:13-2019), and positive (anti-H3K4Me3) and negative (IgG) control antibodies from the kit following the manufacturer's protocol. Library preparation was performed by the YCGA followed by sequencing to a depth of 30 million paired-end 150 bp reads (2×150) per sample on NovaSeq (Illumina). Adapter sequences from paired-end sequencing reads were trimmed with Trimmomatic and quality control was performed using FastQC. Bowtie2 (v2.4.2) was used to align reads to the GRCm38 genome using the −very-sensitive flag and allowing for one mismatch. SAMtools version 1.16 was used to remove unmapped reads, secondary alignment reads, and reads with a Phred score of <30. Macs2 (v2.2.7.1) was used to call narrow peaks with default peak-calling settings. H3K4me3 and cJun CUT&RUN peaks were called against IgG negative controls. IDR was used to determine significant peaks between the two biological replicates. Deeptools (v3.5.1) was used for data visualization. To visualize average CUT&RUN signal per sample, Deeptools bamCoverage was used to normalize bigWig signal based on *E. coli* spike-in control DNA. A scaling factor was calculated based on the percentage of reads mapped to the *E. coli* K12, MG1655 reference genome. The following additional bamCoverage flags were used: --extendReads --ignoreDuplicates --samFlagInclude 64. Differences in read coverage between samples were normalized based on CPM. In all plots, the mean normalized bigWig signal between two biological replicates is plotted. Peak intersections were performed using BEDTools (v2.30). Enriched motifs were identified using MEME v5.4.1 meme-chip. For all published and publicly available datasets, BED and/or bigWig files were converted to the mm10 genome or hg19 genome for mouse and human sequencing datasets, respectively. Peaks and bigWig signal were visualized with the Integrated Genomics Viewer (IGV). Hi-C data were visualized with the WashU Epigenome Browser[103–106].

## Tissue preparation for histology, immunohistochemistry, and immunofluorescence

Mice were euthanized by $CO_2$ asphyxiation. Pancreata was isolated, fixed in 10% neutral-buffered formalin overnight, dehydrated in 70% ethanol, and embedded in paraffin by Yale Tissue Pathology Services. Adjacent 5-μm sections were cut and stained with hematoxylin and eosin (H&E) or subject to immunohistochemistry (IHC) following citrate-buffer antigen retrieval using a ThermoFisher Scientific Autostainer 360 or manual staining (for CCK). Mach 2 HRP-labeled micropolymers (Biocare Medical) or HRP-conjugated secondary antibodies (Jackson ImmunoResearch or Vector Labs) were used for primary

antibody detection. Tissues were scanned using an Aperio whole slide scanner at 20X magnification. For immunofluorescence (IF) analyses, mice were fixed with 4% PFA (Electron Microscopy Sciences) via cardiac perfusion. The pancreas, small intestines, and liver were harvested and fixed in 4% PFA overnight at 4 °C, washed in PBS three times for 5 min, and cryoprotected in 30% sucrose in PBS at 4 °C. Tissue was embedded in O.C.T. (Tissue-TEK) and stored at −80 °C prior to cryosectioning. 10-µm cryosections were obtained using a Leica Cryostat (CM1860). Slides were thawed at room temperature for one hour and washed three times for 10 min with PBS. Tissue was permeabilized and blocked for one hour with PBT (PBS, TritonX-100 (0.3%)) with 10% donkey serum (Jackson ImmunoResearch). Slides were incubated at 4 °C with primary (Day 1) and secondary (Day 2) antibodies (Jackson ImmunoResearch) overnight in PBT and 5% donkey serum in a humidified chamber and washed three times for 10 min after each antibody incubation with PBT. On Day 3, slides were stained with DAPI (1:500; ThermoFisher Scientific) in PBS and washed once with PBS. Slides were mounted using Vectashield Hardset Antifade Mounting Medium (VectorLabs). Slides were imaged at the Yale West Campus Imaging Core using a Leica SP8 Spinning Disk Confocal Microscope and an Andor Benchtop Confocal Microscope BC43. Monochromatic red, green, and blue images were merged using ImageJ software (NIH). Specific antibodies used for IHC and IF analyses are listed in Supplementary Table 1.

## Whole-mount immunostaining, tissue clearing, and light sheet microscopy

Mouse pancreatic tissue was cleared using a modified iDISCO protocol[107,108]. Mice were transcardially perfused with 1× PBS followed by 4% PFA containing 10 µg/mL heparin. Pancreata were harvested and post-fixed overnight at 4 °C in 4% PFA. Samples were dehydrated through a graded methanol series (20%, 40%, 60%, 80%, and 100%) in B1N buffer (300 mM glycine, 0.1% Triton X-100, 0.01 N NaOH, 0.02% sodium azide in distilled water; 1 hr per step). Delipidation was performed in 100% dichloromethane (DCM), followed by three 30 min washes in 100% methanol. Tissues were bleached overnight at 4 °C in 5% hydrogen peroxide ($H_2O_2$) in methanol, then rehydrated through a reverse methanol gradient in B1N buffer. Permeabilization was achieved by overnight incubation in modified PTxwH buffer (0.5% Triton X-100, 0.1% Tween-20, 2 mg/mL heparin, 0.01% sodium azide in 1× PBS) supplemented with 5% dimethyl sulfoxide (DMSO) and 300 mM glycine, followed by three 30 min washes in PTxwH. Samples were blocked overnight at room temperature in blocking buffer (3% donkey serum in PTxwH), then incubated with anti-Glucagon antibody (Sigma-Aldrich, G2654; 1:2000 in blocking buffer) for 7 days at 37 °C. After washing in PTxwH, tissues were incubated with Alexa Fluor 568−conjugated anti-mouse secondary antibody under identical conditions. Post-immunostaining, samples were embedded in 0.75% agarose, dehydrated through a methanol gradient, incubated overnight in 100% methanol, delipidated in DCM, and refractive index–matched in dibenzyl ether. Cleared, agarose-embedded samples were imaged on an Ultramicroscope Blaze light-sheet microscope (Miltenyi Biotec) equipped with a scientific CMOS camera and an MI PLAN 4× objective lens (zoom 0.60). Samples were immersed in ethyl cinnamate (refractive index = 1.55) to minimize optical aberrations. Images were acquired in ImSpector Pro v7.7.2 (Miltenyi Biotec) in multicolor mosaic mode using 568 nm excitation (Glucagon) and 488 nm excitation (autofluorescence). The light sheet was generated by dual-sided illumination with a width of 50% and thickness of 5.00 µm; z-stacks were collected at 2.5 µm steps. Tiles were stitched using Imaris Stitcher (Oxford Instruments).

## Immunocytochemistry

Min6 cells were grown in a 6-well plate with a 20 mm round cover glass (Cell Treat, Part#: 229173) for 3 days, washed with PBS (2x, 5 min each), fixed with 4% PFA (15 min), and washed with PBS (2x, 5 min each) at room temperature. Cells were blocked using PBT with 10% donkey serum and incubated overnight at 4 °C in primary antibody in PBS-T with 2.5% donkey serum. On Day 2, cells were washed with PBS (3x, 5 min each) and incubated overnight at 4 °C in secondary antibody in PBS-T with 5% donkey serum. On Day 3, cells were washed with PBS-T (3x, 5 min each), incubated with DAPI (1:500) for 5 min, washed in PBS for 5 min, and mounted on slides using Vectashield Hardset Antifade Mounting Medium. Slides were imaged with a Leica SP8 Spinning Disk Confocal Microscope. Specific antibodies used for immunocytochemistry (ICC) are listed in Supplementary Table 1.

## Pancreatic islet isolation, perifusion, and single-cell RNA sequencing (scRNA-seq)

Male 16-week-old wild-type (WT) C57/B6 mice (Stock #000664) and male C57/B6 mice fed a 60% kcal fat diet (Research Diets 12492) for 10 weeks (Stock #380050) starting at 6 weeks of age were obtained from JAX. Islets were isolated, pooled for each group ($n = 4$ mice per group), and processed for perifusion for GSIS measurements or dispersed into single cells for sequencing, as previously described[18]. For scRNA-seq, dead cells were removed using a Dead Cell Removal Kit (Miltenyi Biotec, 130-090-101). Cell concentration and viability were determined using a Countess II Automated Cell Counter (Thermo-Fisher Scientific). 10X Genomics Chromium Single Cell 3′ Reagent Kit v3 library preparation and sequencing on Illumina HiSeq were performed by the YCGA.

## Quantification and statistical analysis of tumor development

All disease analyses were performed in a blinded manner on coded scanned H&E slides in QuPath (v0.6.0). Disease burden was quantified by measuring cross-sectional area of disease (ADM, PanIN, and PDAC including stroma) relative to total pancreas area. Histological analysis of tumor progression (presence of PanIN and/or PDAC) was performed by two independent blinded reviewers (C.F.R and M.D.M.) using previously established morphologic criteria[109].

## Quantification and statistical analysis of immunohistochemistry

Quantitative IHC analysis for synaptophysin, insulin, Ki67, CD45, and SMA was performed in QuPath (v0.6.0) using optimized marker-specific detection settings. Islet mass was quantified by measuring cross-sectional area of synaptophysin-positive cells relative to total pancreas area on tissue sections. Proportional β cell area was calculated by cross-sectional area of insulin+ cells relative to synaptophysin+ cells on tissue sections. For Ki67, CD45, and SMA quantification, representative annotations for each mouse were drawn for regions of interest (ROIs) containing PDAC or PanIN lesions (PanIN epithelial fields for Ki67 and SMA; PanIN epithelial and surrounding stromal fields for CD45). Annotation measurements were exported from QuPath and processed in R (v4.3) using the tidyverse package. For Ki67, positive cell density was calculated as the number of Ki67+ nuclei per mm² of annotated lesion area. For CD45, stromal infiltration in PanINs was estimated as CD45+ cells per mm² of stroma, determined by subtracting epithelial counts and areas from total field measurements; PDAC CD45 density was calculated directly from lesion ROIs. For SMA, fractional positive area was computed as SMA + DAB area divided by total annotation area. For each marker and classification, multiple ROIs per mouse were averaged to yield one value per mouse, and data were aggregated by genotype (e.g., KPC vs. KPC-Akita) to calculate mean ± SEM. Normality was confirmed by the Shapiro−Wilk test, and genotype comparisons were performed using unpaired two-sample $t$-tests ($α = 0.05$).

## Quantification and statistical analysis of tumor proximity to islets

Scanned H&E sections of pancreata from lean 13-week-old KC;Lep[ob/+] and obese 7-week-old KC;Lep[ob/ob] mice ($n = 4$-5 mice per group) were

digitally analyzed using QuPath (v0.6.0) in blinded fashion. Neoplastic lesions were largely low-grade PanINs in these models at the designated timepoints. All low-grade PanINs in the splenic lobe of the pancreas (where most islets are located in mice) of each mouse were quantified and classified as islet proximal or distant. Islets (diameter >100 μm) were defined as being proximal if they resided within a distance of 400 μm from the tumor epithelial cell edge to the center of an islet. This represents the approximate length of ten acinar clusters within a lobule, which is twice the expected diffusion limit but would allow hormone delivery via the interconnected islet-acinar capillary network[110,111]. Fisher's exact test was performed to compare lean and obese conditions. Despite harboring fewer overall PanINs ($n = 58$ (14.5 per mouse) vs. 146 (29.2 per mouse)) at this earlier timepoint, $KC;Lep^{ob/ob}$ mice exhibited a great proportion of PanINs close to islets. There was no significant difference ($p = 0.23$, Welch's $t$-test) in the average number of islets per mouse (14 for $KC;Lep^{ob/+}$ and 19.75 for $KC;Lep^{ob/ob}$). For 3D analyses, islets and neoplasia were reconstructed in Imaris v10.2.0 (Oxford Instruments). Islets were segmented based on glucagon immunofluorescence, and neoplastic lesions were identified by differences in autofluorescence intensity[112]. The "surface" tool in Imaris was used to quantify the shortest distances from islet to neoplasia and neoplasia to islet, as well as islet radii. Statistical comparisons of two groups were performed using the Mann–Whitney test. Cumulative distributions were compared with the Kolmogorov-Smirnov test.

### In vivo lineage tracing analysis

To analyze sufficient islets at single-cell resolution for each mouse, we obtained widefield fluorescence images using a Nikon Spinning Disk Confocal (CSU-W1) under 40x or 60x oil immersion with laser wavelengths of 405-nm, 488-nm, 561-nm, and 647-nm. Fluorescence channels were merged using ImageJ, and positive cells were counted manually. Briefly, 15-20 islets were analyzed for each 16-week-old experimental mouse per line, scored for presence or absence of TdTomato, insulin, or CCK expression, and each counted cell was marked in ImageJ to avoid double counting. The percentage of CCK+ or insulin+ cells also expressing TdTomato were calculated for each mouse for statistical comparisons between mice. For mouse line validation (5-week analysis timepoint), 10 islets were analyzed per mouse. For $Ins1$-$Cre^{ERT}$ and $Ngn3$-$Cre^{ERT2}$ mouse lines, the proportion of insulin+ cells harboring TdTomato fluorescence was quantified. For the $Gcg$-$Cre^{ERT2}$ line, we calculated the percentage of Gcg+/TdTomato+ cells and divided the result by the total Gcg+ cell population. Due to unequal variance, the Welch's $t$-test was used for statistical analysis when comparing groups.

### Preprocessing of scRNA-seq data

**Murine obesity models.** scRNA-seq data from 12 week-old C57/B6 $Lep^{ob/ob}$ and WT mice were previously published by our group[18], and raw data deposited in GEO (GSE137236) were reanalyzed in comparison to new HFD and WT data from 16-week-old mice, as described above. All sample data were processed with Cell Ranger (v3.0.2) using the mm10 mouse genome indices from 10X Genomics. We combined batches from all four scRNA-seq datasets and removed rare genes (expressed in fewer than 15 cells). We then filtered and L1 normalized cells by library size (maximum library size = 15000) and square-root transformed the data. We finally filtered cells with high mitochondrial expression (maximum expression = 12.5). All processed cells were visualized with PHATE (v1.0.11)[113], with $Ins1$, $Ins2$, $Sst$, $Ppy$, and $Gcg$ removed from PHATE calculation due to very high expression affecting cell-cell distances. All cells were then clustered with **k**-means clustering ($n\_clusters = 20$) and clusters were annotated based on published mouse and human islet marker genes[33] to distinguish endocrine cells and β cells for downstream analysis. Clusters 13 and 14 were annotated as low-quality clusters due to lack of enrichment for key markers of

pancreatic islet cell populations and distinct embedding visualization. Cluster 13 showed low overall counts, strong enrichment for hemoglobin genes, and low ribosomal counts, indicative of contaminating erythroid cells. Cluster 14 showed low ribosomal counts and enrichment primarily for lncRNAs (*e.g.*, *Malat1*, *Neat1*, and *Gm42418*) whose high expression reflects high nuclear fraction and cytoplasmic RNA loss due to membrane damage[114]. Cluster 11 was further split into clusters 11 and 20 due to enrichment of Schwann cell markers *S100b*, *Ngfr*, and *Pmp22* in cluster 20. This clustering was chosen due to stability over five runs (average pairwise ARI = 0.86) and balance between overclustering and merging different cell types. β cells were then subsetted and further filtered to remove outlier populations with large differences in the number of genes expressed per cell, indicating low quality of the cell populations. Endocrine cell clusters were retained for TrajectoryNet, re-embedded, and clustered with **k**-means clustering ($n\_clusters = 10$, grouped for annotation).

To verify transcriptional differences between models were driven by condition-specific effects (WT vs. HFD vs. $Lep^{ob/ob}$) rather than batch effects or age-based differences, we calculated pairwise Earth Mover's Distances (EMD) between the four samples, where a smaller distance reflects a higher similarity between cells in each sample. In endocrine cells measured from the principal component (PC) space used for trajectory inference and analysis (see below), we confirmed the two WT samples (12- and 16-week-old) were more similar to each other (EMD = 0.45) than to their respective age-matched samples from obese mice (EMD = 0.76 and 0.80). Furthermore, both samples were closer to the HFD sample than the $Lep^{ob/ob}$ sample (EMD = 0.63 vs. 0.76 for 12-week-old WT and 0.80 vs. 1.26 for 16-week-old WT). This trend held in β cells analyzed in the PC space, where the two WT samples (12- and 16-week-old) were more similar to each other (EMD = 0.59) than their respective age-matched samples (EMD = 0.94 and 0.66). Both samples were closer to the HFD sample than the $Lep^{ob/ob}$ sample (EMD = 0.59 vs. 0.94 for the 12-week-old and 0.66 vs. 0.84 for 16-week-old). Finally, we computed the modified average silhouette width (ASW) of batch, as previously implemented[115]. This metric measures the silhouette of a given batch for each cell type. For the scaled metric we used here, 0 indicates separation of batches, and 1 indicates perfect overlap of batches. In the combined β cell PC space, the two age-matched batches had a modified ASW of 0.71, and the two WT samples (subsetting to WT only) had a modified ASW of 0.84. Together, these results suggest higher concordance between the WT samples than their respective age-matched samples.

Given the similarity between WT samples, we merged cells from both samples for all downstream analyses. A total of 17,336 β ($Ins1+/Ins2+$) cells and 260 acinar ($Cpa1+/Prss2+$) cells were retained. For gene cluster analysis, GSEA of gene clusters, and Granger causality analysis, we used highly variable genes defined using default parameters in Scanpy (v1.9.3)[116]. Acinar cell embeddings were generated with PHATE. Endocrine β cell embeddings were generated by first denoising with MAGIC (v3.0.0)[117], then reducing dimensionality with PCA. This enabled visualization of heterogeneity similar to PHATE while maintaining invertibility to the gene space for trajectory and latent-space archetypal analysis.

**Published datasets.** Upregulated ($\log_2$FC expression > 1.5, FDR < 0.05) acinar-specific genes from RNA-seq analysis of islets isolated from $Lepr^{db/db}$ versus WT mice[71] were extracted from "Table S3" of Egozi et al.[72]. The gene signature associated with the acinar-REG+ population were derived from differentially expressed genes identified in Tosti et al.[72]. Differentially expressed genes for patients exposed to pharmacologic β cell stressors were extracted from the Gene Expression Omnibus (GEO, GSE237448)[58]. We subsetted to DEGs per stressor that were enriched versus control (CTRL) in the majority of patients (at least 3 of 5). We then computed the gene sets for DEGs per stressor. For BFA and TG, we took the intersection of gene sets before comparing

overlap with obesity. For CM, IFN, IL1B, IL1BIFNG, and TNFA, we took the intersection of gene sets before comparing overlap with obesity. All gene sets enriched per stressor and associated genes are provided in Supplementary Data 3.

Pre-processed and annotated scRNA-seq data from islets derived from mice treated with STZ (mSTZ) or control (vehicle) were obtained from GEO (GSE211799)[37] and subsetted to β cells. β1, β2, and β-mSTZ annotations were used from original work, except for six control cells labeled as β-mSTZ, which were relabeled as β2 for cell-of-origin analysis to differentiate between an immature pre-existing population and a differentiated cell state.

Human islet datasets were harmonized across measured clinical variables, subsetted to β cells from ND and T2D donors, and pre-processed for library size normalization and square-root transformation. The mouse atlas[33] was subsetted to only β cells from control/ND and type 2 diabetes (T2D) mouse models. Samples with fewer than 2 cells were excluded. Samples annotated "Embryonic" show embryo progression from E12.5 to E15.5. Samples annotated as P16, 2m, 4m, 6m, and aged were islet cells isolated from 16 day-old, 2-month-old, 4-month-old, 6-month-old, and ≥20-month-old mice, respectively, and sorted according to the Fltp lineage-tracing model. mSTZ samples are derived from healthy adult control mice, an STZ-treated model, and mSTZ models subject to different antidiabetic treatments. *db/db* samples are from healthy adult control *Lepr^{db/db}* mice and *Lepr^{db/db}* mice subject to different antidiabetic treatments. Chem samples are healthy young adult control samples (only control retained for this analysis).

## TrajectoryNet

**Overview.** Traditionally, characterizing progression axes across cells relies on "pseudotime" analysis, where each cell is assigned a given time value corresponding to its relative state on the main axis or axes of variation. Ordering of cells is determined by the choice of a starting cell or inferred by the method. However, due to the destructive nature of scRNA-seq, inferring the trajectory of each cell across timepoints remains experimentally challenging, and the inferred or chosen starting and ending timepoints may not reflect the true latent dynamics of the system. This motivated the development of TrajectoryNet, a manifold-aware neural ODE network that performs dynamic optimal transport between timepoints[23,28]. TrajectoryNet learns individual trajectories for each cell from the first timepoint (WT) to the last timepoint (*Lep^{ob/ob}*) by optimizing for distribution matching between the predicted and true timepoint, as well as energy efficiency, which enables dynamic optimal transport of cells. Additionally, TrajectoryNet is regularized for *unbalanced* dynamic optimal transport through modeling cellular proliferation and death with an auxiliary proliferation network. Together, this ensures TrajectoryNet can learn constrained continuous normalizing flows for each cell that are biologically plausible.

**TrajectoryNet-based cell-of-origin definition.** Each individual trajectory learned by TrajectoryNet defines the trajectory of a given cell in the last timepoint, where TrajectoryNet generates the most likely cells along the path to the first timepoint. Thus, the generated cells at the first timepoint represent the "cells-of-origin" for the learned trajectories. Notably, in addition to distribution matching, TrajectoryNet ensures cellular paths are energy efficient and models biological proliferation and death, meaning that the learned cells-of-origin may be enriched for a particular subset of cells within the first timepoint (in our dataset, cells in the WT population).

**Cflows for TrajectoryNet-based granger causality analysis.** Using the top 2,278 highly variable genes (defined in *Preprocessing of scRNA-seq data* above), we first computed the mean gene trends over time

with TrajectoryNet for two averaged runs. For each transcription factor (TF) in the TRRUST v2 database[63] and all highly variable genes, we computed the *p*-value of the Granger causality test, or the time-lagged correlation between the TF and the target with lag of 10 (10% of the trajectory), where the *p*-value is computed based on the chi square distribution. We then convert the *p*-value into a Granger causality score, as described in the *Cflows* framework[28], by taking the log and adding a sign based on the correlation between two gene trends.

For building a regulatory network of genes strongly associated with the obesity progression trajectory, we first identified the top 100 TFs with the most regulatory effect (*i.e.*, the highest Granger score across all highly variable genes) and subsetted all TF-target relationships from TRRUST to those TFs. Then, within this subset, we further filtered TF-target pairs to only those with high absolute Granger score (*i.e.*, -log (*p*-value) > 25). We considered this network the full regulatory network. For visualization, we extracted those pairs where both genes are considered "increasing" over the trajectory based on the gene cluster analysis. We visualized the largest connected component, which includes 28 out of 36 genes in the "increasing" subnetwork.

**Experimental details.** We learned cellular trajectories to all cells in the *Lep^{ob/ob}* condition for all endocrine cells and to cells assigned to *Lep^{ob/ob}* AT 2 for the β cells in our obesity models and in the mSTZ model. TrajectoryNet (v0.2.4) was run twice in each setting, where enrichment scores (see below) were calculated separately and plotted together, and the mean expression from both runs was computed for visualizing gene trends, calculating gene clusters, and downstream gene set enrichment analysis. One run was used for visualizing individual cellular trajectories. We ran TrajectoryNet on embeddings built from running MAGIC[117], then PCA (to ensure denoised trajectories as well as invertibility to the gene space), and with the proliferation regularization with alpha = 2 (other parameters default). TrajectoryNet proliferation rate was robust to choice of alpha within [0.01,10] (pairwise Spearman ρ between 0.81 and 1.0).

## scMMGAN

**Overview.** Mapping one dataset to another or generating a representation of one dataset in another modality, species, condition, or strain, allows us to build a broader understanding of the relationships between cells from different settings toward identifying common gene signatures. scMMGAN is a generative adversarial network architecture for domain adaptation, where the model is trained to preserve relationships between data points and ensure the original data can be reconstructed from the generated output[26]. Unlike batch correction methods, scMMGAN allows us to map β cells from different experimental setups onto the obesity progression manifold.

**Experimental details.** For all runs, we ran scMMGAN (v0.0) with 15,000 training steps, batch size of 128, learning rate of 0.0001, and $\lambda_{cycle}$ of 1. The correspondence loss for scMMGAN can be flexibly defined. For all mappings, we enforce the preservation of the Pearson correlation of a cell prior to mapping and after mapping to the new domain. This ensures maintenance of the cell's identity for accurate mappings. For mapping vehicle to WT cells and mSTZ to *Lep^{ob/ob}* cells, we ran scMMGAN after MAGIC, then PCA (to match the preprocessing of each condition for AAnet). We used $\lambda_{correspondence}$ of 10, and scMMGAN was run twice to show consistency of enrichment scores. For mapping the mouse atlas and the human samples, we ran scMMGAN after PCA (for the batch-integrated mouse atlas in 1 run and for the human datasets individually due to batch effect). Due to the larger differences between these datasets and our obesity progression, versus the mSTZ dataset, we used $\lambda_{correspondence}$ of 15 to increase preservation of biological identity. The mapping was then visualized using PHATE[113].

**Evaluation of integration.** To assess integration of batches, we assessed the tradeoff of integration and biological preservation. First, we computed the modified average silhouette width (ASW) of the batch, as previously implemented[115]. This metric measures the silhouette of a given batch for each cell type (where we have only one cell type). For the scaled metric we used here, 0 indicates separation of batches, and 1 indicates perfect overlap of batches. To ensure we did not enforce mixing where cells are transcriptionally dissimilar, we also evaluated our integration based on preservation of biological signatures. We thus computed the cosine similarity of each cell before and after mapping (where high cosine similarity is 1 and low cosine similarity is −1). We termed this measure global distortion based on a related metric using cell correlation[118]. A good mapping has both batch integration and cosine similarity near 1, though generally, a tradeoff is necessary due to large technical variation between datasets. In particular, we expected some transformation of cellular gene expression to ensure mapping. Thus, we aimed for mild to moderate global distortion (cosine similarity = 0.2-0.6) and high batch integration (integration > 0.6) in order to ensure the nearest cells in the obesity progression were truly near in the embedding space.

For vehicle to WT mapping, scMMGAN balanced batch integration (mean Batch ASW = 0.91) and biological preservation (mean cosine similarity before versus after correction = 0.69; Supplementary Data 4a). For mSTZ to $Lep^{ob/ob}$, scMMGAN also observed high integration performance (mean Batch ASW = 0.64, mean cosine similarity = 0.71; Supplementary Data 4b). For the mouse integrated atlas, after scMMGAN alignment, the datasets showed high batch integration (Batch ASW = 0.98) and mild global distortion across all cells (cosine similarity=0.58 before and after alignment), with no distortion specific to a particular archetype, developmental stage, disease state, or dataset (Supplementary Data 4c). For human samples, scMMGAN showed high batch integration (Batch ASW = 0.97) and moderate global distortion across all cells (cosine similarity=0.35 before and after alignment), common when mapping between datasets with substantial differences, including species-specific differences, with no distortion specific to a particular archetype, sex, or disease state (Supplementary Data 4d).

### AAnet

**Overview.** Defining cell states via cellular clustering remains a useful tool for single-cell analysis, but proves insufficient for characterizing heterogeneity within cell types. Indeed, β cell heterogeneity is well appreciated in the field, but cell states are highly plastic and require an alternative approach to model this plasticity. AAnet[24,25] is a latent-space archetypal analysis approach that learns cellular "archetypes", or extreme states, and defines each cell with respect to the extreme states. This allows us to identify cells highly specific to a given state.

**Archetypal commitment.** Archetypal commitment is captured in the AAnet latent space, where each cell is defined by its affinity to each archetype (a vector of size number of archetypes, where the entries sum to 1 and higher value implies higher commitment). Thus, we determine cells that have commitment over 0.5 for a given archetype as "committed" to that archetype[25].

**Archetypal assignment for mapped cells.** For the vehicle and mSTZ analysis[37], we learned mappings directly onto the embedding space used for training the WT AAnet and $Lep^{ob/ob}$ AAnet models, respectively. Thus, for these datasets, we used the trained AAnet models to encode archetypal assignments for cells. For other mappings, including with TrajectoryNet and with scMMGAN on the full trajectory of cells, we assigned each cell the archetype of its closest neighbor in the reference dataset.

**Experimental details.** We ran AAnet (v0.0) with default parameters on each condition embedding separately, defined by running MAGIC followed by PCA (again to enable denoising and invertibility to the gene space). To determine the number of archetypes for each condition[24], we first generated a training and test set, where the training test was generated by density subsampling 80% of the dataset, and the test set was the held-out group. We then trained AAnet on the training set 3 times for each number of archetypes between 2 and 10. For each trained model, we calculated the mean squared error (MSE) of the input versus reconstructed cells for the held-out test set. We plotted the MSE and chose the elbow point as the number of archetypes (Supplementary Fig. 6a).

### DiffusionEMD

**Overview.** Patient manifolds are multi-scale manifolds, where one level characterizes relationships between cells, and a higher level characterizes relationships between patients[119]. To compare single-cell datasets between patients, such as to compare cell type proportions, gene trends, or clinical variables across patient mappings, Diffusion Earth Mover's Distance (DiffusionEMD)[27] models each single-cell data set as a distribution supported on a common cell-cell graph enabling comparison via optimal transport.

**Experimental details.** We ran DiffusionEMD (v0.5.0) with default parameters to embed human patient samples after their mapping to the obesity progression with scMMGAN.

### Quantitative and statistical analyses of scRNA-seq data

**Enrichment score calculation.** To compute the enrichment of a given label in a mapped population of cells, we annotated each mapped cell using the label of the nearest "real" cell in the reference dataset. We then calculated the proportion of each class in the mapped population. This alone does not sufficiently characterize the mapping of a population due to large class imbalances between labels. Therefore, we calculated the ratio between the mapped label proportions and the label proportions in the reference dataset. We term this the "enrichment score". For each class, if the ratio is greater than 1, the class is "enriched" in the mapped population. If the ratio is less than 1, the class is enriched in the reference population. If the ratio is equal to 1, the class is equally represented in the reference and the mapped population. For all endocrine cells (Fig. 3), we calculated the enrichment of each cell type for cells-of-origin to $Cck+ Lep^{ob/ob}$ β cells and for cells-of-origin to $Cck+ Lep^{ob/ob}$ polyhormonal cells versus the distribution of cell types in the WT population. Within β cells (Fig. 3), we calculated the enrichment of each WT archetype for cells-of-origin versus the distribution of archetypes in the WT population. We also calculated the enrichment of each archetype for each timepoint versus the distribution of all archetypes to learn the trajectory of archetypes from WT AT 4 to $Lep^{ob/ob}$ AT 2. For the mSTZ dataset (Fig. 5), we calculated the enrichment of each WT archetype for the β2 population versus the distribution of archetypes in the reference WT population. We also calculated the enrichment of each $Lep^{ob/ob}$ archetype for the $Cck$-hi population versus the distribution of $Lep^{ob/ob}$ archetypes in the reference. Finally, we calculated the enrichment of β1 versus β2 for mapped cells-of-origin from TrajectoryNet to $Lep^{ob/ob}$ AT 2-annotated cells.

**Differential expression.** For differential gene expression of archetypes, we compared cells committed to each archetype versus all other cells and performed a two-sided Wilcoxon rank sum test, where differentially expressed genes have Benjamini-Hochberg (BH)-adjusted $p$-value ($q$-value) < 0.05 and log fold-change > 0.0.

**Gene clusters.** We clustered the 2,278 highly variable genes based on TrajectoryNet-inferred transcriptional changes from all WT to $Lep^{ob/ob}$ AT 2. We computed the mean expression over all trajectories from two TrajectoryNet runs and clustered each gene's association with continuous time. Genes were clustered using **k**-means clustering with the number of clusters set to two.

**Gene set enrichment analysis.** Gene set enrichment analysis was performed using GSEApy (v1.1.2)[56] and Enrichr[57], using the BioPlanet 2019, Panther 2016, KEGG 2019, Reactome 2022, and GO Biological Process 2023 databases. Gene sets with BH-adjusted $p$-value ($q$-value) $< 0.05$ were retained. The hypergeometric test was used for determining statistical significance of overlapping gene sets.

## Statistics and Reproducibility

Statistical tests used for each experiment, including definitions of center and precision measures, are described in the figure legends. A $p$-value threshold 0.05 was used to denote statistical significance. No statistical method was used to predetermine sample size. In single-cell analyses, outlier populations of β cells with large differences in the number of expressed genes per cell, indicating low cell quality, were filtered. No other data were excluded. For allocation into experimental treatment groups, mice were randomized following pre-stratification for sex and weight. Investigators were not blinded to allocation but were blinded to outcome assessment (*e.g.*, disease burden evaluation).

## Reporting summary

Further information on research design is available in the Nature Portfolio Reporting Summary linked to this article.

## Data availability

New scRNA-seq, bulk RNA-seq, and CUT&RUN sequencing data generated in this study were deposited into the Gene Expression Omnibus (GEO) under accession numbers GSE279485, GSE287290, and GSE280947, respectively. Accession numbers for reanalyzed data are listed below. All other data are included in the main text, supplementary information, or source data file. There are no restrictions on data availability. Our previously published mouse scRNA-seq data[18] were obtained from GEO (GSE137236: https://www.ncbi.nlm.nih.gov/geo/query/acc.cgi?acc=GSE137236). The following published datasets from ENCODE[66,67] were used: H3K27ac ChIP-seq from mouse cortex (ENCSR491RBV: https://www.ncbi.nlm.nih.gov/geo/query/acc.cgi?acc=GSE231221), H3K27ac ChIP-seq from mouse hippocampus (ENCSR527FRO: https://www.ncbi.nlm.nih.gov/geo/query/acc.cgi?acc=GSE231268), DNase-seq from mouse hippocampus (ENCSR544UYQ: https://www.ncbi.nlm.nih.gov/geo/query/acc.cgi?acc=GSE215676), and CTCF ChIP-seq from human adult pancreas tissue (ENCSR774PGN: https://www.ncbi.nlm.nih.gov/geo/query/acc.cgi?acc=GSE175121), H3K27ac ChIP-seq and ATAC-seq from human islets were obtained from Miguel-Escalada et al.[68] (https://ega-archive.org/studies/EGAS00001002917). Loop anchor and Hi-C data (https://cmdga.org/experiments/TSTSR043623/) from human islets were obtained from Greenwald et al.[69]. The following human islet scRNA-seq datasets were accessed from GEO: GSE101207, GSE81608, GSE86469, GSE154126, GSE83139, and GSE237448. Published mouse pancreatic islet scRNA-seq data were accessed from (GSE211799: https://cellxgene.cziscience.com/collections/296237e2-393d-4e31-b590-b03f74ac5070) and GSE211799. Source data are provided with this paper.

## Code availability

Code for TrajectoryNet v0.2.4, DiffusionEMD v0.5.0, MAGIC v3.0.0, PHATE v1.0.11, AANet v0.0, and scMMGAN v0.0 used in single-cell RNA-sequencing analyses has been deposited in Github[120] (https://doi.org/10.5281/zenodo.18316269).

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

## Acknowledgements

We thank the Muzumdar and Krishnaswamy lab members for helpful discussions and feedback; L. Lawres for technical assistance with mouse breeding and genotyping; Drs. F. Zhang and S. Chen for technical assistance with AAV-Leptin production; the Yale Center for Genome Analysis (YCGA) for library preparation and sequencing; the W.M. Keck Biotechnology Resource Laboratory at Yale University for primer generation; J. Nikolaus and the West Campus Imaging Core for assistance with confocal microscopy; the Center for Cellular and Molecular Imaging (CCMI) Electron Microscopy Facility for electron microscopy; X. Zhao for assistance with mouse islet isolations, islet dispersion, and intact islet perifusion studies at the Chemical Metabolism Core at Yale University; T. Nottoli and the Yale Genome Editing Center (YGEC) for cryorecovery of mouse embryos and in vitro fertilization (IVF); Dr. G. Cline for cell lines; and Drs. D. Davis, T. Jacks, K. Kaestner, A. Lowy, A. Leiter, L. Philipson, H. Zeng, the Allen Institute for Brain Science, and EuCOMM for mice. C.C.G. was a recipient of a National Cancer Institute (NCI) Ruth L. Kirchstein National Research Award (NRSA) for predoctoral students (F31-CA268845) and was funded by the Training Program in Genetics (TPG) training grant (5T32-GM148332). A.V. was funded by a Gruber Science Fellowship. D.C.M. and A.S. are supported by the Yale Medical Scientist Training Program (5T32-GM007205). S.S.A. was supported by a postdoctoral fellowship through the Yale Cancer Biology Training Program (T32-CA193200). C.F.R. was supported by a postdoctoral fellowship through the Yale Cancer Biology Training Program (T32-CA193200) and is supported by an NCI Research Supplement to Promote Diversity in Health-Related Research (R01-CA276108-03S1). J.B.J. was supported by a American Society of Clinical Oncology-Conquer Cancer Foundation Endowed Young Investigator Award (YIA) in Memory of John R. Durrant, MD, the Yale Cancer Center Advanced Training Program for Physician-Scientists (T32 CA233414), and is supported by the Yale Center for Clinical Investigation (YCCI) Scholar Award (NCATS KL2 TR001862) and a Department of Defense (DoD) PCARP Idea Development Award (HT9425-24-1-0954, to MDM). K.H.L. and B.M. acknowledge support from the NIDDK (R00-DK129712), the NIDA (P30-DA018343), and the American Cancer Society (IRG-21-132-60). R.G.K. is supported by the NIDDK (R01-DK127637). S.K. is supported by the National Science Foundation (NSF Career Grant 2047856, NSF DMS grant 2327211, and NSF CISE grant 2403317). M.D.M. acknowledges support from an NIH Director's New Innovator Award (DP2-CA248136), the Lustgarten Foundation Therapeutics Focused Research Program (TFRP), DoD (HT9425-24-1-0954), NCI (R01-CA276108, R01-CA292936), and in part, the Yale Comprehensive Cancer Center Support Grant (P30-CA016359). The content is solely the responsibility of the authors and does not necessarily represent the official views of the National Institutes of Health. This research was supported in part by a seed grant (to K.H.L. and M.D.M.) from the Cheryl and Allen Lipson Neuroendocrine Tumor Fund as part of the Center for Gastrointestinal Cancers at Smilow Cancer Hospital and Yale Cancer Center. This work was largely supported by Damon Runyon-Rachleff Innovation Awards (66-21/66S-21) to M.D.M.

## Author contributions

C.G., D.C.M., and M.D.M. conceived of and designed the study. C.G., D.C.M., S.S.A., C.F.R., C.Z., A.S., and J.B.J. performed mouse and cell line experiments and associated analyses. A.V. and A.T. performed computational experiments and analysis, which were supervised by S.K. B.M. performed tissue clearing and 3D pancreatic imaging experiments and analysis, which were supervised by K.H.L. R.C. performed islet isolations for scRNA-seq and perifusion experiments, which were supervised by R.G.K. R.S. performed analysis of CUT&RUN data. S.K. and M.D.M. supervised the overall study. C.G., A.V., D.C.M., and M.D.M. wrote the manuscript with input from all authors.

## Competing interests

M.D.M. and S.S.A. are inventors on a patent applied for by Yale University that is unrelated to this work. M.D.M. received research funding from a Genentech supported AACR grant and an honorarium from Nested Therapeutics, which is unrelated to this work. All other authors declare no competing interests.
