## [Transparent Peer Review file · Nature Communications]

Beta cell-derived cholecystokinin drives obesity-associated pancreatic adenocarcinoma development

Corresponding Author: Dr Mandar Muzumdar

Version 0:

Reviewer comments:

Reviewer #1

(Remarks to the Author)

This paper explores the role of pancreatic endocrine-exocrine crosstalk in tumorigenesis using a combination of experimental and sequencing methods. Key methods include β cell ablation experiments in mice to assess its impact on PDAC development, and the induction of obesity to study its effects on β cell hormone dysregulation and tumor formation. Single-cell RNA-sequencing was used to analyze transcriptional changes in peri-islet exocrine cells, coupled with in silico latent-space archetypal and trajectory analysis to map cellular states and transitions. Additionally, genetic lineage tracing in vivo was used to investigate β cell adaptation and pro-tumorigenic transformation, identifying the JNK/cJun signaling pathway as a putative mediator of these effects.

Major weaknesses:

Overall, the descriptions of the experiments in the paper are well written, but it is difficult to see the conceptual insights this paper contributes. Almost all panels are based on single-cell transcriptional data without functional or mechanistic follow-up. The conclusions are based on a few experiments measuring cellular expression without clear implications or next steps. The discussion does not address weaknesses or alternative interpretations, but rather lists the findings and references consistent with their own work.

Specific comments:

Fig 1d: The KPC-Akita group does not show any significant blood glucose levels changes relative the WT group which seems incorrect if they have no insulin-producing islets. The Akita group is hitting the roof at 600 mg/dl glucose and therefore this experiment is not reliable. This should be done under fasted conditions.

All heatmaps lack units in the scale bars. It is not possible for the reader to know the fold change based on the presented data in e.g. Fig 3f, 3d, 3h and many other figures. This is a major concern also because the variability is not shown either, and the values are based on a very low cell number extracted from the single-cell RNA seq analysis.

The impact of the proposed mechanism of cJun activation under beta cell stress leading to CCK expression is unclear without follow up experiments.

While a lot of transcriptional associations are shown, there are often no or little explanations of how PDAC develops or progresses based on these data. Overall, these concerns dampens my enthusiasm for this study.

(Remarks on code availability)

I have not reviewed the code

Reviewer #2

(Remarks to the Author)

Summary

In this manuscript, the authors use genetic and computational models to demonstrate that obesity drives a pro-tumorigenic

beta cell state, and that endocrine-exocrine crosstalk is important for the development of pancreatic ductal adenocarcinoma. Using the Akita mouse line crossed into the KPC PDAC model, the authors establish a tumor-supporting role for endocrine beta cells at baseline. They then leverage deep bioinformatic analysis to evaluate beta cell heterogeneity in obesity and show that pro-tumorigenic CCK+ beta cells are enriched in obesity and arise from the proliferation of immature Ngn3+ beta cells. Importantly, the authors use lineage tracing in vivo to support these findings, demonstrating that Ngn3-CreERT is sufficient to label CCK+ cells that are enriched over time in the context of obesity. Mechanistically, obesity was associated with beta cell transcriptional signatures associated with cellular and ER stress, and pharmacological induction of beta cell stress phenocopied obesity-driven transcriptional changes and increased CCK expression. CCK expression was regulated by cJun N-terminal kinases (JNK) in obese mice in vivo through direct regulation of a CCK enhancer by cJun. Together, the central implications of this work are that (1) beta cells are important for pancreatic tumor development at baseline and (2) that obesity triggers a general beta cell stress response via cJun/JNK, leading to proliferation of immature pro-tumorigenic CCK+ beta cells. These findings are each important in their own right, and are supported by a wealth of computational analysis backed by biological validation. However, one weakness of the manuscript is the failure to unify these two findings. We can infer that the mechanisms of obesity-driven beta cell remodeling identified here may be important for obesity-driven PDAC development, but this is not directly shown. Instead, the deeper investigation of beta cell biology in obesity is not extended to the context of PDAC or obesity-associated PDAC. The authors have the opportunity to strengthen this manuscript by drawing some more direct throughlines between these otherwise compelling and interesting findings. Further, the finding that tumorigenesis is reduced in KPC-Akita mice is not supported by in-depth characterization of the model; building on this work could also greatly strengthen the manuscript and the authors' claims.

Major comments

1. The finding that beta cell loss in the KPC-Akita model impairs tumor development (Fig.1c) is central to the manuscript and this claim would be better supported by additional characterization of the model. Is tumor mass or volume reduced in KPC-Akita mice? How is survival impacted? Cellular markers such as proliferation, cell death? Metastasis? Assessment of stromal content would also be useful. As it stands, tumor area is only assessed at one time point and it is difficult to parse whether these effects are driven by impacts on tumor latency or proliferation. Although assessing each of these points may not be necessary, any additional details would help the reader better assess the magnitude of the effect of beta cell loss on tumor trajectory in the commonly-used KPC model, and help the reader assess whether beta cells are really "essential" for PDAC development.

2. In Fig. 2j the authors quantify PanINs in proximity to islets in KCO mice. The text should be modified to reflect this; referring to PanIN as "early tumors" is somewhat misleading. This distinction matters in that low-grade PanIN will not all necessarily progress to PDAC and low-grade PanIN-like bystander lesions have even been detected in normal tissue adjacent to PDAC in mice. The increase in proximal PanIN in KCO mice is suggestive of beta cell promotion of early lesions but not necessarily the development of tumors. The authors could quantify tumors proximal to islets in KCO mice at later timepoints to support this point. Further, it seems possible that if obesity indeed drives up beta cell mass and PanIN content as would be expected, then the increase in peri-islet lesions could simply be driven by a greater density of islets or PanIN within the tissue in obese mice. It appears that the authors evaluated KC mice at 13 weeks and KCO at 7 weeks, possibly to account for this though it's not clear if this is why these timepoints were chosen. The authors should include quantification of islets/beta cells or PanIN to illustrate the magnitude of the difference in these two conditions so the reader could assess the possible relevance of islet/PanIN density. As an additional control, the authors could consider quantifying PanIN or tumors proximal to islets in KPC vs KPC-Akita. This study would have the same issue with different density of islet/PanIN by condition but would provide further support that beta cell function is (or is not) required for peri-islet PanIN or tumor development. Assessment of islet-proximal lesions in KCO mice +/- a CCK inhibitor or other beta cell signaling modulator would probably be most appropriate to truly answer this question but is certainly not necessary within the scope of this manuscript.

3. The authors set out to study both endocrine (beta cell) contributions to pancreatic tumorigenesis, and the underlying mechanisms driving pro-tumorigenic beta cell function in the context of obesity. However, the authors do not directly link the identified mechanisms of obesity-associated beta cell proliferation/phenotype to PDAC or obesity-associated PDAC. It would strengthen the manuscript to more directly demonstrate whether beta cell stress/proliferation and JNK-driven CCK expression contribute to baseline PDAC and obesity-associated PDAC. To extend these findings to the cancer context, the authors could:

- Show whether PDAC induces a similar beta cell stress response in vivo at baseline by comparing and quantifying beta cell number or insulin/CCK expression in wild-type, wild-type Akita, KPC and KPC-Akita mice, and an obesity KPC or KCO model. How does baseline PDAC development impact beta cells? It's unclear whether the obesity-driven changes to beta cell biology identified here are similarly present and important for baseline PDAC development. If changes to beta cell state/mass identified in obesity (for example increase in CCK+ beta cells) are not identified in KPC mice, what alternative mechanism explains the role for beta cells in baseline tumor development?
- The authors demonstrated an important role for beta cell CCK on obesity-associated tumorigenesis in prior work. However, additional studies in KPC/KPC-Akita mice would validate the role of CCK+ beta cells in baseline PDAC development. The authors could test if CCKR inhibitor or JNKi reduce tumor burden in KPC mice to verify the functional relevance of the new mechanism in vivo and the role of CCK+ beta cells outside of the obesity setting. KPC-Akita mice could be used to control for beta cell-independent effects of inhibitors on PDAC tumors.
- Evaluate the effect of obesity on tumorigenesis in the context of beta cell depletion (KPC-Akita). This experiment would delineate the extent to which obesity-driven cancer is dependent on beta cells. Is there a reason the authors did not pursue

these studies, or why the KPC-Akita model might not be suitable to answer this question?

Minor comments

1. Could the authors please provide any quantification of beta cell depletion in the KPC-Akita model (Figure 1b)? It would be helpful to know the extent to which insulin+ beta cells are depleted.
2. The authors should define "PP cells" in the section on "Dynamic changes in beta cell expression of insulin and CCK with obesity".
3. Reg expression was reduced by CCKRi treatment of Ob/ob mice; did this correspond to a loss of the Cluster 3 Reg+ cells? (Fig. 2i)
4. "Thus, Cck+ cells arise from pre-existing beta cells..." In statements regarding the use of bioinformatic tools such as TrajectoryNet it might be more appropriate to soften the language to indicate that these tools really generate a hypothesis as to the lineage or cell of origin instead of truly indicating which population arose from another as illustrated instead by the authors' later lineage tracing studies.

(Remarks on code availability)

Reviewer #3

(Remarks to the Author)

The article by Garcia et al. crosses the KPC model of PDAC with the Akita model of beta-cell ablation and shows decreased tumour formation at 12 weeks in the KPC-AKITA model even in the presence of elevated blood glucose, suggested that more available free glucose in the bloodstream isn't driving tumorigenesis but the beta-cells directly. Following this discovery, they switch focus to beta-cell transformation during obesity, which is a risk factor for PDAC development, and find that acinar cells in the peri-islet space have altered phenotype during obesity. They then characterise beta- heterogeneity during obesity, identifying an embryonic/immature beta-cell program in mice driven by JNK/cJun which leads to increased CCK expression, a known PDAC promoter.

Although the topic is highly relevant to better understand exocrine and endocrine pancreas interaction/communication there is need for a reinterpretation of the results and in some cases refine some of the stated conclusions:

Major Comments

- The author consistently jump between figures and the order of graphs within individual figures does not match the in-cite text. Re-ordering either the figures or the in-cite text to match each other would make it a lot easier for the reader to follow the data. This was especially frustrating with figures 3 and 4, and also the out-of-order graphs in extended Fig. 7. Furthermore, their reasoning and logic for using the various bioinformatic tools employed in the paper isn't stated very clearly until the discussion.
- Authors state, "Despite decreased tumorigenesis, KPC-Akita mice showed markedly elevated glucose levels (Fig. 1d), suggesting that hyperglycemia is not independently pro-tumorigenic in the absence of β cells". Surprisingly KPC mice have a glucose basal level of 400 mg/dl, these glucose levels are extremely high, could the authors comment on that. 400 vs 600 mg/dl will not allow us to draw any conclusion about the roles of hyperglycemia in PDAC.
- In extended figure 3. The authors show the single cell data of the 3 mouse models analyzed, but the number of cells per model is different. A proportion plot per model will be helpful in this regard as many conclusions are based on this single cell data.
- As the Akita model causes inflammatory cell death of beta-cells, can the authors be confident that there isn't an artificial recruitment of immune cells to the pancreas, which alters the tumour microenvironment of the pancreas, therefore priming it to remove early-stage tumours, and therefore the results they achieve is immune dependent and not beta-cell dependent?
- What would further strengthen the authors argument that glucose isn't driving PDAC tumour development is a time course of glucose levels over the disease progression of the two models KPC and KPC-AKITA. Moreover, further investigation of the role of insulin signalling pathway will also help elucidate the main driver of this conclusion: immature beta cell phenotype is activated by ER stress, glucose changes or insulin ones.
- The study has focused on acinar derived PDAC, but they frequently state PDAC as a whole, ignoring ductal derived PDAC. A more nuance discussion specifying acinar vs ductal origin is needed throughout.
- In the Peri-islet acinar cell adaptation section drawing any conclusion based on 260 acinar cells is an overstatement. Moreover, assuming that the acinar cells come from peri-islet region exclusively is not right as the islet preps contain also acinar contaminants not located in the peri-islet region. A detailed validation of this section is needed at list at the level of IF or similar. Tosto et al. already described 3 acinar populations in the human pancreas. The authors should look at Tosti or other scRNAseq data of acinar cells as well.
- In the section of "identification of the cell-of-origin for pro-tumorigenic β cells in obesity", the authors introduce the possibility of several cells of origin of new beta cells but then just analyze beta cell as possible pro-tumorigenic β of origin. Further elaborating this decision is important.
- In extended Fig 4a, the authors have quantified the amount of TdTom+ cells that are glucagon positive, the same should be performed for the Ins1-CreERT and Ngn3-CreERT models in Supp Fig 4b, given the incomplete labelling hypothesis

presented in the paper. I also think an explanation for the inefficient labelling should be stated in-text or hypothesised on. The interpretation of Ngn3-CreERT labelling results are unclear. Moreover, as far as it is known, the number of Neurog3 positive cells after weaning is extremely low, thus not sure about the labelling, are those cells in the islets or ducts? Several papers argue about it, a detailed characterization of the model and % of positive cells is needed.

- The reasoning for choosing Lepob/ob AT2 to examine in TrajectoryNet out of the different ATs in group 2 is not clear. By choosing that as the endpoint, are you then forcing its evolutionary trajectory? Explanation is needed as to why the authors have chosen this comparison over other archetypes.
- Extended Fig. 6 has not been expanded upon in the text. Please explain these results in more detail.
- In Fig.6b, WT AT 4 is only represented in 3/15 ND patients. This is inconsistent with the mouse data presented earlier, and questions whether your proposed cell of origin is conserved in humans.
- Does treatment with the JNKi inhibitor result in decreased tumour formation in obese PDAC models? While the embryonic program stimulated in B-cells during pharmacological (STZ) stress and obesity is interesting, the data isn't related back to the PDAC model in the first figure.

Minor Comments

- Overall, the authors concluding statements at the end of every results section are very vague, and a more informative summary of each section's results would make it easier for the reader to interrupt and follow their conclusions.
- In Fig.1b, the number of mice used to ascertain that insulin was decreased and ER stress induced in the KPC-Akita model relative to KPC is not stated in the figure legends or methods. Quantification of the insulin and BiP positive area within the islets would help support the conclusions the authors have made in the manuscript "We confirmed decreased β cell mass and induction of ER stress in islets of KPC-Akita mice "and therefore their conclusions that B-cell loss impedes PDAC tumorigenesis.
- In Figure 1C, it is not stated in methods how the authors quantified tumour burden in the KPC and KPC-Akita models.
- The author writes, "Having established that β cells are basally required for exocrine tumorigenesis, ...". This is a very strong conclusion, as the KPC-Akita mice still have 40% tumour burden. I think promote would be a more accurate interpretation of the author's results.
- In methods, it is stated that only male mice were used for STZ and scRNAseq experiments due to the reduced beta-cell effects of STD and a High Fed Diet. Please comment on whether these results are then relevant for human female patients.
- In Fig.3D and the accompanying text, as you heavily discuss the biological functions of the groups of archetypes, I would rearrange the heatmap on Fig. 3D to follow the evolutionary tree in ext. Fig.5 and then colour code them by WT, HFD and Lepob/ob to make it easier for the reader to follow your conclusions.
- In Fig.5J, you only show the mapping of WT AT 4 and OB/OB AT 2. What about the other archetypes, these should be included in the supplementary/extended figures.

(Remarks on code availability)

Reviewer #4

(Remarks to the Author)

The paper presents an interesting study with extensive analyses. However, the current structure makes it somewhat confusing. The experiments are not clearly presented in a hypothesis-driven manner, which makes it difficult to follow the logical flow of the study. Additionally, the bioinformatics methodology needs more detail, particularly regarding the parameters used and how they were selected.

Major Concerns

Tumorigenesis in KPC-Akita Mice (Figure 1)

Since tumors are still observed in KPC-Akita mice, can the authors definitively conclude that β -cell loss impedes pancreatic tumorigenesis? This claim should be carefully re-evaluated.

Single-Cell Data Generation and Experimental Design

The comparison between congenic obesity models (HFD-fed for 10 weeks, Lepob/ob) and age-matched WT controls raises concerns. The authors should clarify how they accounted for age-related differences to isolate the effects of the conditions studied.

Clustering and Data Quality (Figure 2 Extended Data)

Cluster 6 appears to express multiple markers from different cell types. How was cell-type annotation performed to ensure accuracy?

What were the nFeature thresholds and percentage of mitochondrial genes for clusters 13 and 14, which were considered low-quality?

Methods: Data Integration and Batch Effect Control

How was data integration performed, and how were batch effects controlled?

Which covariates were used to integrate the datasets and minimize technical noise?

How was $n = 20$ clusters determined?

What methods were used to verify that the clusters retained biological identity and were not artifacts of overclustering?

Trajectory Analysis Validity

Given the differences in age and conditions among the animals, could the trajectory analysis be artificially linking cell types that may not truly represent progenitor or intermediate states?

The authors hypothesize that acinar cells surround the islets (peri-islet localization). Can they compare their gene signature with spatial transcriptomics data from the literature to validate this claim?

Gene Expression Measurements (Figure 2)

what metric is used for gene expression measurement?

How are normalized counts defined in this analysis?

Validation of β -Cell Archetypes

The authors should validate β -cell archetypes using IHC or FACS.

How was the number of subpopulations determined?

In WT mice, how do the β -cell archetypes vary with age?

Batch Effect Concerns and Best Practices

The authors state that datasets were not directly comparable due to batch effects. However, best practices in single-cell data integration should be followed to improve comparability: <https://www.sc-best-practices.org/preamble.html>.

Methodological Reproducibility: Software, Databases, and Parameters

The methods section should explicitly state:

Software versions used (e.g., Seurat, Scanpy, Cell Ranger).

Reference databases used for annotation.

Flags and parameters for each function/tool and justification for their selection.

(Remarks on code availability)

Reviewer #5

(Remarks to the Author)

(Remarks on code availability)

Reviewer #6

(Remarks to the Author)

(Remarks on code availability)

Version 1:

Reviewer comments:

Reviewer #1

(Remarks to the Author)

The manuscript has been significantly improved by incorporating text changes and adding additional experiments. I have no further scientific comments.

The code has not been verified by any reviewer. This should be corrected.

(Remarks on code availability)

No reviewer has assessed the reproducibility of the code. This should be corrected by assigning a reviewer with this expertise.

Reviewer #2

(Remarks to the Author)

In this manuscript, the authors use genetic and computational models to demonstrate that obesity drives a pro-tumorigenic beta cell state, and that endocrine-exocrine crosstalk via CCK is important for obesity-associated pancreatic ductal adenocarcinoma. The authors use bioinformatic analysis of scRNA-seq followed with lineage tracing in vivo to show that obesity drives cellular stress via cJUN/JNK, leading to proliferation of immature (Ngn3+) CCK-expressing beta cells that could drive tumorigenesis. Through additional model characterization and modification/reorganization of the text the authors have sufficiently addressed our concerns. The authors have re-organized assessment of the KPC-Akita model which improves the clarity of the manuscript, highlighting the role of beta cells in tumor development in lean mice, and thus independent of obesity and CCK. Inclusion of a new CCKfl/fl allele crossed to the KCO mouse model provides compelling evidence that beta cell derived CCK is a driver of obesity-associated tumor development. The authors should be congratulated on this nice study that will expand our understanding regarding the relationship between obesity and PDAC.

My minor suggestions for improvements, are:

The authors should briefly note the timepoint at which each model was assessed for disease burden either in the text or legends. If CCKfl/fl KCO mice were assessed at the same timepoints as KC counterparts it would be helpful to present the disease burden in KCO +/- CCK side by side with KC counterparts to demonstrate the magnitude of the effect relative to lean mice.

Additionally, if the authors assessed ADM, PanIN, and PDAC across mouse models it would be more informative to report the burden of each rather than combined disease burden.

In Figure 8f, it is somewhat confusing that CD45 and SMA+ cells are quantified within PanIN, PDAC lesions; I'm not sure what that would mean. Were these cells quantified with FOV containing PanIN or PDAC lesions?

(Remarks on code availability)

Reviewer #3

(Remarks to the Author)

The reviewed manuscript submitted by Garcia et al. has addressed most of our previous concerns and figures have improved. The results support much better the conclusions of the manuscript.

Most strikingly, the addition of the CCK overexpression and CCK knockout models has really strengthened the conclusions of this manuscript and shows a much clearer role for CCK on PDAC progression than the first iteration of this manuscript. The serial glucose measurements between the two models strengthens the conclusions proposed in the original manuscript, that hyperglycaemia is not the main driver of PDAC progression in these models. And the correlation analyses showing CCK expression associates with disease burden instead of insulin strengthens these conclusions.

The revised manuscript is clearer, the figures are easier to interpret, and the story has a much better flow. They have corrected many of the statements that we felt were overstated or not clear in the original manuscript. The enrichment of the Acinar-REG+ signature identified by Tosti et al from the human pancreas in cluster 3 better supports that the acinar cells are of peri-islet origin. They have addressed the question of sex bias (where they previously only used male mice for the STZ and scRNAseq experiment) by performing all their new in-vivo experiments with both sexes and showing no significant differences between the sexes, suggesting the vast majority of their results will still be relevant for female patients.

Overall, we think the authors took great efforts to address many of the reviewer's previous concerns, and the manuscript is a lot stronger for it. I endorse the acceptance of this paper.

(Remarks on code availability)

Reviewer #4

(Remarks to the Author)

Overall, the revisions are appreciated. The revised organization and the new data linking β -cell CCK expression to obesity-associated PDAC progression improve the readability of the manuscript.

Please add minimal clarifications and diagnostics to strengthen rigor and reproducibility: specify the feature space and cell subset used for EMD. The added EMD analysis is a useful global similarity check, but it is not sufficient to support a claim of "lack of batch effect": sample-level EMD can be driven by differences in cell-state composition and QC/capture differences, may miss subtle batch/age shifts within β cells (or β substates), and does not directly validate the absence of batch effects in the same latent space used for downstream clustering/trajectory unless that is explicitly shown. Therefore, please temper the statement "verifying lack of batch effect" and include one within- β cell batch/age mixing diagnostic on the trajectory input/latent space (e.g., ASW_batch, iLISI, or kBET) plus a β -cell embedding colored by sample, and show the distribution of trajectory position/pseudotime by sample to demonstrate the inferred trajectory is not simply tracking sample identity. Finally, it may be more appropriate to soften trajectory/archetype language to reflect that these tools generate testable hypotheses

regarding lineage relationships or candidate cells of origin, whereas the definitive evidence for lineage directionality is provided by the subsequent in vivo lineage tracing experiments.

(Remarks on code availability)

Reviewer #5

(Remarks to the Author)

(Remarks on code availability)

Reviewer #6

(Remarks to the Author)

(Remarks on code availability)

RESPONSE TO REVIEWERS

Reviewer #1 (Remarks to the Author):

This paper explores the role of pancreatic endocrine-exocrine crosstalk in tumorigenesis using a combination of experimental and sequencing methods. Key methods include β cell ablation experiments in mice to assess its impact on PDAC development, and the induction of obesity to study its effects on β cell hormone dysregulation and tumor formation. Single-cell RNA-sequencing was used to analyze transcriptional changes in peri-islet exocrine cells, coupled with in silico latent-space archetypal and trajectory analysis to map cellular states and transitions. Additionally, genetic lineage tracing in vivo was used to investigate β cell adaptation and pro-tumorigenic transformation, identifying the JNK/cJun signaling pathway as a putative mediator of these effects.

Major weaknesses:

Overall, the descriptions of the experiments in the paper are well written, but it is difficult to see the conceptual insights this paper contributes. Almost all panels are based on single-cell transcriptional data without functional or mechanistic follow-up. The conclusions are based on a few experiments measuring cellular expression without clear implications or next steps. The discussion does not address weaknesses or alternative interpretations but rather lists the findings and references consistent with their own work.

We thank the reviewer for the comments. In the revised manuscript, we have provided additional experiments to establish the functional importance of β cell CCK expression in obesity-driven cancer. Specifically, we have used a new mouse model that enables conditional knockout of CCK ($CCK^{Flx/Flx}$) in the pancreas of *KCO* mice (**Fig. 2**). We find that pancreas-specific knockout of CCK reduces tumor formation (**Fig. 2**) without reducing body weight, decreasing insulin secretion, or compromising glucose homeostasis (**Extended Data Fig. 2**). Since our scRNA-seq and IF analyses do not identify significant CCK expression in any cells in the pancreas beyond β cells, our data argue that β cell-derived CCK is a *bona fide* driver of obesity-driven pancreatic cancer. We also confirmed that β cell CCK overexpression is sufficient to promote pancreatic tumorigenesis in lean *KC; Lep^{ob/+}* mice to a degree that was non-significantly different to obese *KCO* mice (**Fig. 1**). Finally, our results establish that CCK – rather than insulin – more strongly correlates with disease burden across our models (**Fig. 1l,m and Fig. 2g,h**), challenging the conceptual dogma in the field regarding insulin as the primary β cell driver of obesity-associated PDAC. Together, these data implicate targeting β cell CCK as potential means to intercept obesity-driven PDAC development.

Figure 1. β cell CCK expression promotes PDAC development.

- a) Schematic of alleles used to generate *KC;Lep^{ob/+}*, *KC;Lep^{ob/+};Ins1-CCK/+*, and *KCO* mice.
- b) Pancreatic CCK expression (mean \pm SEM) normalized to islet marker synaptophysin (Syn) demonstrates CCK upregulation in 3-month-old *KC;Lep^{ob/+};Ins1-CCK/+* and *KCO* mice relative to *KC;Lep^{ob/+}* littermates (n = 9-18 mice with sex denoted by symbols). p-values of one-way ANOVA with Kruskal-Wallis post-hoc test are shown.
- c) Representative IHC images of pancreata of mice in (b) showing elevated islet CCK expression in *KC;Lep^{ob/+};Ins1-CCK/+* and *KCO* mice. Scale bar, 100 μ m.
- d) Representative IHC images of ductal tumorigenesis (Ck19) and quantification of disease burden (mean \pm SEM) of mice in (b). p-values of one-way ANOVA with Kruskal-Wallis post-hoc test are shown. Scale bar, 100 μ m.
- e) Percentage of mice in (b) harboring PanINs and/or PDAC demonstrating enhanced progression in *KC;Lep^{ob/+};Ins1-CCK/+* and *KCO* mice relative to *KC;Lep^{ob/+}* littermates. p-values of Fisher's exact tests are shown.
- f) Terminal weight (mean \pm SEM) of mice in (b) shows weight is unchanged in 3-month-old *KC;Lep^{ob/+};Ins1-CCK/+* mice relative to *KC;Lep^{ob/+}* littermates but is significantly elevated in *KCO* mice relative to both groups. p-values of one-way ANOVA with Kruskal-Wallis post-hoc test are shown.
- g) Terminal glucose (mean \pm SEM) of mice in (b). p-values of one-way ANOVA with Kruskal-Wallis post-hoc test are shown.
- h) Terminal c-peptide (mean \pm SEM) of mice in (b). p-values of one-way ANOVA with Kruskal-Wallis post-hoc test are shown.
- i) No association between pancreatic CCK expression and terminal c-peptide in *KC;Lep^{ob/+};Ins1-CCK/+* mice. Each point represents one mouse (n=12). Pearson correlation coefficient (r) and p-value are shown.
- j) No association between terminal c-peptide and disease burden in *KC;Lep^{ob/+};Ins1-CCK/+* mice. Each point represents one mouse (n=12). Pearson correlation coefficient (r) and p-value are shown.
- k) Pancreatic CCK expression is inversely correlated with terminal C-peptide in *KCO* mice. Each point represents one mouse (n=18). Pearson correlation coefficient (r) and p-value are shown.
- l) Pancreatic CCK expression is positively correlated with disease burden in *KCO* mice. Each point represents one mouse (n=18). Pearson correlation coefficient (r) and p-value are shown.
- m) Terminal c-peptide levels are inversely correlated with disease burden in *KCO* mice. Each point represents one mouse (n=18). Pearson correlation coefficient (r) and p-value are shown.

Figure 2. Pancreatic CCK is required for obesity-associated PDAC progression.

- a) Schematic of alleles used to generate *KCO*, *KCO*;*CCK*^{Flox/+}, and *KCO*;*CCK*^{Flox/Flox} mice.
- b) Pancreatic CCK expression (mean ± SEM) normalized to synaptophysin validates CCK knockout in 3-month-old *KCO*;*CCK*^{Flox/Flox} mice relative to *KCO* and *KCO*;*CCK*^{Flox/+} littermates (n = 10-15 mice with sex denoted by symbols). p-values of one-way ANOVA with Kruskal-Wallis post-hoc test are shown.
- c) Representative IHC images of pancreata of mice in (b) showing decreased islet CCK expression in *KCO*;*CCK*^{Flox/Flox} mice. Scale bar, 100mm.
- d) Representative IHC images of ductal tumorigenesis (Ck19) and quantification of disease burden (mean ± SEM) of mice in (b). p-values of one-way ANOVA with Kruskal-Wallis post-hoc test are shown. Scale bar, 100mm.
- e) Percentage of mice in (b) lacking PanINs or harboring PanINs and/or PDAC demonstrating impaired progression in *KCO*;*CCK*^{Flox/Flox} and *KCO*;*CCK*^{Flox/+} mice compared to *KCO* littermates. p-values of Fisher's exact test with Freeman-Haltman extension are shown.
- f) Pancreatic CCK expression is inversely correlated with terminal c-peptide of mice in (b). Each point represents one mouse with *Cck* genotype denoted (n=35). Each point represents one mouse (n=35). Pearson correlation coefficient (r) and p-value are shown.

- g) Pancreatic CCK expression is positively correlated with disease burden of mice in **(b)**. Each point represents one mouse with *Cck* genotype denoted (n=35). Pearson correlation coefficient (r) and p-value are shown.
- h) Terminal c-peptide levels are inversely correlated with disease burden of mice in **(b)**. Each point represents one mouse with *Cck* genotype denoted (n=35). Pearson correlation coefficient (r) and p-value are shown.

These new results provide a strong rationale for the deep exploration of mechanisms of β cell CCK induction, which represented the majority of the original manuscript. We used complementary *in silico* and functional genetic lineage tracing mechanisms to establish how a subpopulation of β cells expand and adapt (through stress-induced JNK/cJun signaling) to enable CCK expression (**Fig. 3-4**). These data provide novel conceptual and mechanistic insights into β cell hormone dysregulation: CCK expression is not solely due to alterations in gene expression and signaling pathway activation as one might expect but also arises from the expansion of a specific β cell subpopulation (Ngn3+ cells), demonstrating how intrinsic β cell heterogeneity can contribute to disease states. Using both *in vitro* and *in vivo* methods, we have further validated a direct role for JNK/cJun signaling in CCK regulation (**Fig. 7**). Together, these experiments provide significant mechanistic insights into how β cells respond to obesity and produce hormones that are critical drivers of PDAC progression. We believe that these insights have important implications for understanding not only how endocrine-exocrine signaling drives pancreatic cancer development (an area of research that has previously lacked this deep mechanistic exploration) but also how obesity governs β cell fate and hormone expression. Therefore, our findings should have interest across a broad array of investigators studying cancer, endocrinology, obesity, and diabetes. We have highlighted these implications in our revised **Discussion**. However, as noted by the reviewer, there are some important limitations of our study, which we now addressed at the end of our revised **Discussion**.

Specific comments:

Fig 1d: The KPC-Akita group does not show any significant blood glucose levels changes relative the WT group which seems incorrect if they have no insulin-producing islets. The Akita group is hitting the roof at 600 mg/dl glucose and therefore this experiment is not reliable. This should be done under fasted conditions.

We thank the reviewer for the astute observation. *KPC* mice develop hyperglycemia over time likely due primary pancreatic disease invading and ablating functional islets and possible indirect effects on β cell function¹, referred to as Type 3c diabetes in humans. Although the *KPC-Akita* mice are universally hyperglycemic, this results in a non-significant increase in blood glucose levels compared to *KPC* mice at the endpoint. We have now included serial non-fasting glucose levels that demonstrate differences in the kinetics of hyperglycemia between *KPC* and *KPC-Akita* mice (**Fig. 8e**). Hyperglycemia occurs much earlier in *KPC-Akita* mice. We have also provided insulin levels that demonstrate that Akita mice exhibit reduced β cell mass and significantly decreased insulin secretion relative to *KPC* mice (**Fig. 8c,d**). Fasting glucose and insulin represent only the basal levels of each and are not a suitable surrogate for the actual exposure of pancreatic cells at any given time, as these are modulated by feeding. Therefore, we

believe (and have shown previously²) that non-fasting measurements are more appropriate to test whether glucose and/or insulin are linked to tumorigenesis.

Fig. 8. Functional depletion of β cells suppresses pancreatic tumorigenesis

- c) Quantification of proportion β cells area (insulin-positive) of total islet area (synaptophysin-positive) in *KPC* and *KPC-Akita* mice. Mean \pm SEM (n=4-6 mice per group with sex denoted by symbols) and p-value of Welch's t-test are shown.
- d) Final non-fasting insulin and C-peptide levels (mean \pm SEM, n=6-10 mice per group with sex denoted by symbols). p-values of Welch's t-test are shown.
- e) Serial non-fasting blood glucose measurements (mean \pm SEM, n=4-9 mice per timepoint per group with sex denoted by symbols) of *KPC* and *KPC-Akita* mice. Maximum glucometer measurement is 600 mg/dL). P-value of mixed-effects model with Geisser-Greenhouse correction and Tukey's post-hoc test are shown.

All heatmaps lack units in the scale bars. It is not possible for the reader to know the fold change based on the presented data in e.g. Fig 3f, 3d, 3h and many other figures. This is a major concern also because the variability is not shown either, and the values are based on a very low cell number extracted from the single-cell RNA seq analysis.

We thank the reviewer for the suggestion and acknowledge that, while heatmaps present a straightforward overview of the archetypes for a broad audience, more detail on the archetypal differences and variability of expression within archetypes is needed. We have added **Extended Data Fig. 6** for visualization of expression of each gene for each condition and cited **Table S1** for Wilcoxon rank sum differential expression testing of all genes for each archetype versus rest within each condition. We have also provided violin plots for β cell subclusters from vehicle/mSTZ settings (**Extended Data Fig. 8**) and acinar cell clusters (**Extended Data Fig. 12**).

Extended Data Fig. 6. Marker gene expression and variability across archetypes.

- a) WT archetype marker gene violin plots showing density estimates of the distribution of expression and individual cells overlaid.
- b) HFD archetype marker gene violin plots showing density estimates of the distribution of expression and individual cells overlaid.
- c) *Lep^{ob/ob}* archetype marker gene violin plots showing density estimates of the distribution of expression and individual cells overlaid.

Extended Data Fig. 8. Marker gene expression and variability across STZ and control β cell subclusters.

Violin plots depicting density estimates of the distribution of expression with individual cells overlaid belonging to STZ and control β cell subclusters.

Extended Data Fig. 12. Marker gene expression and variability across acinar cell subclusters.

Violin plots depicting density estimates of the distribution of expression with individual cells overlaid belonging to each acinar subcluster.

Additionally, for figure panels where we represent expression of a single marker (*e.g.* **Fig. 9g** *Tff2* expression in acinar cells), we have now added units to the scale bar. For scaled heatmaps, which convey relative expression across multiple clusters/archetypes for each gene, we have added additional visualization to indicate the variability of expression within populations and the normalized expression measurement for each gene.

The impact of the proposed mechanism of cJun activation under beta cell stress leading to CCK expression is unclear without follow up experiments.

We thank the reviewer for the comment. Consistent with data in Min6 cells *in vitro*, we showed in *Lep^{ob/ob}* mice, that JNK inhibition (JNKi) decreases phospho- and total cJun levels and these levels linearly correlates with CCK expression (**Fig. 7c,d**), arguing that the JNK/cJun signaling pathway regulates obesity-mediated CCK expression *in vivo*. While it would be ideal to show that JNKi alters tumor progression in *KCO* mice, this experiment cannot be done cleanly, as we would not be able to exclude the likely possibility that JNKi would have direct effects on tumor cells themselves leading to the phenotype (rather than due to a reduction in β cell CCK expression). To circumvent this concern, we performed gain- and loss-of-function experiments of pancreas-specific CCK to demonstrate that β cell CCK expression is necessary for obesity-driven PDAC development and sufficient to promote tumor formation in lean *KC;Lep^{ob/+}* mice (comparable to obese *KCO* mice) (**Fig. 1-2**).

Fig. 7. JNK/cJun signaling governs β cell CCK expression.

- c) Relative pancreatic *Cck* expression (mean \pm SEM, n=4-5 mice/group) by qRT-PCR (normalized to synaptophysin a marker of islet cells) of *Lep^{ob/ob}* mice treated with JNKi (20 mg/kg SP-600125 x5 days) or vehicle control. p-value of Welch's test is shown.
- d) Correlation of pancreatic *Cck* expression with cJun and phospho-cJun S73 levels (normalized to total protein (Ponceau)) in *Lep^{ob/ob}* mice in (c). Pearson correlation coefficients and *p*-values are shown.

While a lot of transcriptional associations are shown, there are often no or little explanations of how PDAC develops or progresses based on these data. Overall, these concerns dampens my enthusiasm for this study.

We appreciate that our initial manuscript lacked clear experiments that would link the obesity-associated transcriptional associations to functional impacts on tumorigenesis. As described above, we have now established that β cell-derived CCK is a *bona fide* driver of obesity-driven pancreatic cancer (**Fig. 1-2**). We hope that these experiments and the additional analyses and explanations now raise the reviewer's enthusiasm for the study.

Reviewer #2 (Remarks to the Author):

Summary

In this manuscript, the authors use genetic and computational models to demonstrate that obesity drives a pro-tumorigenic beta cell state, and that endocrine-exocrine crosstalk is important for the development of pancreatic ductal adenocarcinoma. Using the Akita mouse line crossed into the KPC PDAC model, the authors establish a tumor-supporting role for endocrine beta cells at baseline. They then leverage deep bioinformatic analysis to evaluate beta cell heterogeneity in obesity and show that pro-tumorigenic CCK+ beta cells are enriched in obesity and arise from the proliferation of immature Ngn3+ beta cells. Importantly, the authors use lineage tracing in vivo to support these findings, demonstrating that Ngn3-CreERT is sufficient to label CCK+ cells that are enriched over time in the context of obesity. Mechanistically, obesity was associated with beta cell transcriptional signatures associated with cellular and ER stress, and pharmacological induction of beta cell stress phenocopied obesity-driven transcriptional changes and increased CCK expression. CCK expression was regulated by cJun N-terminal kinases (JNK) in obese mice in vivo through direct regulation of a CCK enhancer by cJun. Together, the central implications of this work are that (1) beta cells are important for pancreatic tumor development at baseline and (2) that obesity triggers a general beta cell stress response via cJun/JNK, leading to proliferation of immature pro-tumorigenic CCK+ beta cells. These findings are each important in their own right, and are supported by a wealth of computational analysis backed by biological validation. However, one weakness of the manuscript is the failure to unify these two findings. We can infer that the mechanisms of obesity-driven beta cell remodeling identified here may be important for obesity-driven PDAC development, but this is not directly shown. Instead, the deeper investigation of beta cell biology in obesity is not extended to the context of PDAC or obesity-associated PDAC. The authors have the opportunity to strengthen this manuscript by drawing some more direct throughlines between these otherwise compelling and interesting findings.

We thank the reviewer for recognizing the importance and compelling nature of our findings and our use of both computational methods with biological validation. In the revised manuscript, we have provided additional experiments to establish the functional importance of β cell CCK expression in obesity-driven cancer. Specifically, we have used a new mouse model that enables conditional knockout of CCK ($CCK^{Fllox/Fllox}$) in the pancreas of *KCO* mice (**Fig. 2**). We find that pancreas-specific knockout of CCK reduces tumor formation (**Fig. 2**) without altering body weight or glucose homeostasis (**Extended Data Fig. 2**). Since our scRNA-seq and IF analyses do not identify significant CCK expression in any cells in the pancreas beyond β cells, our data argue that β cell-derived CCK is a *bona fide* driver of obesity-driven pancreatic cancer. We further confirmed that β cell CCK overexpression is sufficient to promote pancreatic tumorigenesis (**Fig. 1**). Finally, our results establish that CCK – rather than insulin – more strongly correlates with disease burden across our models (**Fig. 1l,m and Fig. 2g,h**), challenging the conceptual dogma in the field regarding insulin as the primary β cell driver of obesity-associated PDAC. Together, these data directly link obesity-induced β cell transcriptional changes to tumorigenesis and implicate targeting β cell CCK as potential means to intercept obesity-driven PDAC development.

Figure 1. β cell CCK expression promotes PDAC development.

- a) Schematic of alleles used to generate *KC;Lep^{ob/+}*, *KC;Lep^{ob/+};Ins1-CCK/+*, and *KCO* mice.
- b) Pancreatic CCK expression (mean \pm SEM) normalized to islet marker synaptophysin (Syn) demonstrates CCK upregulation in 3-month-old *KC;Lep^{ob/+};Ins1-CCK/+* and *KCO* mice relative to *KC;Lep^{ob/+}* littermates (n = 9-18 mice with sex denoted by symbols). p-values of one-way ANOVA with Kruskal-Wallis post-hoc test are shown.
- c) Representative IHC images of pancreata of mice in (b) showing elevated islet CCK expression in *KC;Lep^{ob/+};Ins1-CCK/+* and *KCO* mice. Scale bar, 100 μ m.
- d) Representative IHC images of ductal tumorigenesis (Ck19) and quantification of disease burden (mean \pm SEM) of mice in (b). p-values of one-way ANOVA with Kruskal-Wallis post-hoc test are shown. Scale bar, 100 μ m.
- e) Percentage of mice in (b) harboring PanINs and/or PDAC demonstrating enhanced progression in *KC;Lep^{ob/+};Ins1-CCK/+* and *KCO* mice relative to *KC;Lep^{ob/+}* littermates. p-values of Fisher's exact tests are shown.
- f) Terminal weight (mean \pm SEM) of mice in (b) shows weight is unchanged in 3-month-old *KC;Lep^{ob/+};Ins1-CCK/+* mice relative to *KC;Lep^{ob/+}* littermates but is significantly elevated in *KCO* mice relative to both groups. p-values of one-way ANOVA with Kruskal-Wallis post-hoc test are shown.
- g) Terminal glucose (mean \pm SEM) of mice in (b). p-values of one-way ANOVA with Kruskal-Wallis post-hoc test are shown.
- h) Terminal c-peptide (mean \pm SEM) of mice in (b). p-values of one-way ANOVA with Kruskal-Wallis post-hoc test are shown.
- i) No association between pancreatic CCK expression and terminal c-peptide in *KC;Lep^{ob/+};Ins1-CCK/+* mice. Each point represents one mouse (n=12). Pearson correlation coefficient (r) and p-value are shown.
- j) No association between terminal c-peptide and disease burden in *KC;Lep^{ob/+};Ins1-CCK/+* mice. Each point represents one mouse (n=12). Pearson correlation coefficient (r) and p-value are shown.
- k) Pancreatic CCK expression is inversely correlated with terminal C-peptide in *KCO* mice. Each point represents one mouse (n=18). Pearson correlation coefficient (r) and p-value are shown.
- l) Pancreatic CCK expression is positively correlated with disease burden in *KCO* mice. Each point represents one mouse (n=18). Pearson correlation coefficient (r) and p-value are shown.
- m) Terminal c-peptide levels are inversely correlated with disease burden in *KCO* mice. Each point represents one mouse (n=18). Pearson correlation coefficient (r) and p-value are shown.

Figure 2. Pancreatic CCK is required for obesity-associated PDAC progression.

- Schematic of alleles used to generate *KCO*, *KCO;CCK^{Flox/+}*, and *KCO;CCK^{Flox/Flox}* mice.
- Pancreatic CCK expression (mean \pm SEM) normalized to synaptophysin validates CCK knockout in 3-month-old *KCO;CCK^{Flox/Flox}* mice relative to *KCO* and *KCO;CCK^{Flox/+}* littermates (n = 10-15 mice with sex denoted by symbols). p-values of one-way ANOVA with Kruskal-Wallis post-hoc test are shown.
- Representative IHC images of pancreata of mice in (b) showing decreased islet CCK expression in *KCO;CCK^{Flox/Flox}* mice. Scale bar, 100mm.
- Representative IHC images of ductal tumorigenesis (Ck19) and quantification of disease burden (mean \pm SEM) of mice in (b). p-values of one-way ANOVA with Kruskal-Wallis post-hoc test are shown. Scale bar, 100mm.
- Percentage of mice in (b) lacking PanINs or harboring PanINs and/or PDAC demonstrating impaired progression in *KCO;CCK^{Flox/Flox}* and *KCO;CCK^{Flox/+}* mice compared to *KCO* littermates. p-values of Fisher's exact test with Freeman-Haltman extension are shown.
- Pancreatic CCK expression is inversely correlated with terminal c-peptide of mice in (b). Each point represents one mouse with *Cck* genotype denoted (n=35). Each point represents one mouse (n=35). Pearson correlation coefficient (r) and p-value are shown.

- g) Pancreatic CCK expression is positively correlated with disease burden of mice in (b). Each point represents one mouse with *Cck* genotype denoted (n=35). Pearson correlation coefficient (r) and p-value are shown.
- h) Terminal c-peptide levels are inversely correlated with disease burden of mice in (b). Each point represents one mouse with *Cck* genotype denoted (n=35). Pearson correlation coefficient (r) and p-value are shown.

Extended Data Figure 2. β cell CCK expression is dispensable for islet homeostasis in obese mice.

- a) Schematic of alleles used to generate *Pdx1-Cre;Lep^{ob/ob}*, *Pdx1-Cre;Lep^{ob/ob};CCK^{Flox/+}*, and *Pdx1-Cre;Lep^{ob/ob};CCK^{Flox/Flox}* mice.
- b) Terminal weight (mean ± SEM) is unchanged between a 4-month-old *Pdx1-Cre;Lep^{ob/ob}*, *Pdx1-Cre;Lep^{ob/ob};CCK^{Flox/+}*, and *Pdx1-Cre;Lep^{ob/ob};CCK^{Flox/Flox}* littermates (n=3-9 mice, sex is denoted by symbols). p-values of one-way ANOVA with Kruskal-Wallis post-hoc test are shown.
- c) Pancreatic CCK expression (mean ± SEM) normalized to synaptophysin validates CCK knockout in *Pdx1-Cre;Lep^{ob/ob};CCK^{Flox/Flox}* mice in (b). p-values of one-way ANOVA with Kruskal-Wallis post-hoc test are shown.
- d) Representative IHC images (10x magnification) of pancreata from mice in (b) shows decreased islet CCK expression in *Pdx1-Cre;Lep^{ob/ob};CCK^{Flox/Flox}* mice. Scale bar, 100mm.
- e) Duodenal CCK expression (mean ± SEM) normalized to GAPDH is unchanged in *Pdx1-Cre;Lep^{ob/ob};CCK^{Flox/Flox}* mice in (b). p-values of one-way ANOVA with Kruskal-Wallis post-hoc test are shown.
- f) Terminal glucose (mean ± SEM) is unchanged in mice (b). p-values of one-way ANOVA with Kruskal-Wallis post-hoc test are shown.
- g) Islet mass (mean ± SEM) is non-significantly different between mice in (a). p-values of one-way ANOVA with Kruskal-Wallis post-hoc test are shown.
- h) Terminal c-peptide (mean ± SEM) under both 6-hour fasted and fed conditions is non-significantly different between mice in (a). p-values of one-way ANOVA with Kruskal-Wallis post-hoc test are shown.
- i) Terminal weight (mean ± SEM) is significantly elevated in 3-month-old *KCO;CCK^{Flox/Flox}* mice relative to *KCO* littermates and is unchanged in *KCO;CCK^{Flox/+}* mice (n=10-15 mice, sex is denoted by symbols). p-values of one-way ANOVA with Kruskal-Wallis post-hoc test are shown.
- j) Terminal glucose (mean ± SEM) is unchanged in mice in (i). p-values of one-way ANOVA with Kruskal-Wallis post-hoc test are shown.
- k) Terminal c-peptide (mean ± SEM) in mice in (i) showing significant elevation in 3-month-old *KCO;CCK^{Flox/Flox}* mice relative to *KCO* littermates and is unchanged in *KCO;CCK^{Flox/+}* mice. p-values of one-way ANOVA with Kruskal-Wallis post-hoc test are shown.

Further, the finding that tumorigenesis is reduced in KPC-Akita mice is not supported by in-depth characterization of the model; building on this work could also greatly strengthen the manuscript and the authors' claims.

We thank the reviewer for pointing this out. We have now added additional data in the *KPC-Akita* experiments (Fig. 8), as described below.

Major comments

1. The finding that beta cell loss in the KPC-Akita model impairs tumor development (Fig.1c) is central to the manuscript and this claim would be better supported by additional characterization of the model. Is tumor mass or volume reduced in KPC-Akita mice? How is survival impacted? Cellular markers such as proliferation, cell death? Metastasis? Assessment of stromal content would also be useful. As it stands, tumor area is only

assessed at one time point and it is difficult to parse whether these effects are driven by impacts on tumor latency or proliferation. Although assessing each of these points may not be necessary, any additional details would help the reader better assess the magnitude of the effect of beta cell loss on tumor trajectory in the commonly-used KPC model, and help the reader assess whether beta cells are really “essential” for PDAC development.

We thank the reviewer for recognizing the importance of the findings in *KPC-Akita* mice and the need for additional characterization. We have now included new information describing that there are no differences in progression to advanced disease (all mice develop PDAC), metastases (no mice exhibited liver metastases at the timepoint analyzed), proliferation (Ki67), immune cells (CD45), or myofibroblasts (SMA) in the tumor microenvironment comparing *KPC* and *KPC-Akita* mice at the endpoint (**Fig. 8f**). We also provide more detailed characterization on the level of functional depletion of β cells in the *KPC-Akita* model by analyses of β cell mass and insulin/C-peptide levels (**Fig. 8b-d**). These data demonstrate that loss of β cells results in decreased overall disease burden. However, lesions that form are capable of progressing to more advanced disease, which correlates with expected changes in the tumor microenvironment (increased immune cell and myofibroblast infiltration with more advanced tumors). These data argue that tumor initiation is partially compromised with functional β cell depletion, showcasing an essential basal role for functional β cells in PDAC development, even in lean mice. We posit that this could be due to reduced insulin levels that play a basal role in tumorigenesis (see **Discussion**).

Fig. 8. Functional depletion of β cells suppresses pancreatic tumorigenesis

- Schematic of the alleles used to generate *KPC* and *KPC-Akita* mice (generated with Biorender).
- H&E and IHC stains show a reduction in the proportion of insulin-expressing islet cells and increased ER stress (BiP) without alterations in CD45+ immune cell infiltration of islets (labeled by pan-neuroendocrine marker synaptophysin) of *KPC-Akita* mice. Scale bar is 50 μ m.
- Quantification of proportion β cells area (insulin-positive) of total islet area (synaptophysin-positive) in *KPC* and *KPC-Akita* mice. Mean \pm SEM (n=4-6 mice per group with sex denoted by symbols) and p-value of Welch's t-test are shown.
- Final non-fasting insulin and C-peptide levels (mean \pm SEM, n=6-10 mice per group with sex denoted by symbols). p-values of Welch's t-test are shown.
- Serial non-fasting blood glucose measurements (mean \pm SEM, n=4-9 mice per timepoint per group with sex denoted by symbols) of *KPC* and *KPC-Akita* mice. Maximum glucometer measurement is 600 mg/dL). P-value of mixed-effects model with Geisser-Greenhouse correction and Tukey's post-hoc test are shown.

- f) Quantification of Ki67-positive tumor cells, CD45-positive immune cells, and smooth-muscle actin (SMA)-positive fibroblast area in PanIN and PDAC lesions of designated mouse genotypes. No statistically significant differences were observed between *KPC* and *KPC-Akita* mice for any marker in either PanIN or PDAC lesions, unpaired two-sample t-tests.
- g) Representative H&E images of *KPC* and *KPC-Akita* mice and quantification of tumor burden (mean \pm SEM, n=7-10 mice per group) at 12 weeks of age. p-value of Welch's t-test is shown. Scale bar is 250 μ m.

2. In Fig. 2j the authors quantify PanINs in proximity to islets in KCO mice. The text should be modified to reflect this; referring to PanIN as “early tumors” is somewhat misleading. This distinction matters in that low-grade PanIN will not all necessarily progress to PDAC and low-grade PanIN-like bystander lesions have even been detected in normal tissue adjacent to PDAC in mice. The increase in proximal PanIN in KCO mice is suggestive of beta cell promotion of early lesions but not necessarily the development of tumors. The authors could quantify tumors proximal to islets in KCO mice at later timepoints to support this point. Further, it seems possible that if obesity indeed drives up beta cell mass and PanIN content as would be expected, then the increase in peri-islet lesions could simply be driven by a greater density of islets or PanIN within the tissue in obese mice. It appears that the authors evaluated KC mice at 13 weeks and KCO at 7 weeks, possibly to account for this though it's not clear if this is why these timepoints were chosen. The authors should include quantification of islets/beta cells or PanIN to illustrate the magnitude of the difference in these two conditions so the reader could assess the possible relevance of islet/PanIN density. As an additional control, the authors could consider quantifying PanIN or tumors proximal to islets in *KPC* vs *KPC-Akita*. This study would have the same issue with different density of islet/PanIN by condition but would provide further support that beta cell function is (or is not) required for peri-islet PanIN or tumor development. Assessment of islet-proximal lesions in KCO mice +/- a CCK inhibitor or other beta cell signaling modulator would probably be most appropriate to truly answer this question but is certainly not necessary within the scope of this manuscript.

We thank the reviewer for the comments and suggestions. We have now changed the text to refer to lesions forming near islets as PanINs (or neoplastic lesions) and not early tumors. The time points chosen for this analysis were indeed to avoid excess PanIN formation in *KCO* mice, which would confound our interpretation of PanINs forming near islets as opposed to growing towards them. This makes analyzing differences between *KPC* vs. *KPC-Akita* and *KCO* mice subject to pancreas-specific CCK knockout challenging, given the vast differences in baseline disease burden. Given the increased number of PanINs in a similar pancreatic area with comparable islet number in *KC;Lep^{ob/+}* mice, we would have expected that PanINs would be more likely by chance to be proximal to islets in the lean model, but that was not the case. We have described this in the **Methods** section as follows:

*“Despite harboring fewer overall PanINs (n=58 (14.5 per mouse) vs. 146 (29.2 per mouse)) at this earlier timepoint, *KC;Lep^{ob/ob}* mice exhibited a great proportion of PanINs close to islets. There was no significant difference (p=0.23, Welch's t-test) in the average number of islets per mouse (14 for *KC;Lep^{ob/+}* and 19.75 for *KC;Lep^{ob/ob}*).”*

We have further solidified these spatial relationships in three dimensions using intact pancreata subject to tissue clearing, immunofluorescence, and light sheet microscopy. Again, we observed an increased proportion of neoplastic lesions in proximity to islets in obese *KCO* mice relative to lean *KC; Lep^{ob/+}* mice with comparable disease volume between groups (Fig. 10b-g).

Fig. 10. Enhanced islet proximal neoplasia in obese mice.

- b) Light sheet microscopy 2D projection of a pancreatic neoplastic lesion (outlined in green) in an 8-week-old *KCO* mouse proximal to an islet (red).
 c) 3D reconstruction of image neoplastic lesion and islet in (b).

- d) Relative neoplastic disease volume normalized to total pancreas volume (mean \pm SEM) of lean 12-week-old *KC; Lep^{ob/+}* and obese 8-week-old *KC; Lep^{ob/ob}* (KCO) mice (n=3 per group).
- e) Violin distribution plots (min/max with 25th, 50th, and 75th percentiles delineated by lines) of the radius of islets from *KC; Lep^{ob/+}* (n=444 islets from n=3 mice) and *KCO* (n=271 islets from n=3) mice. p-value of Wilcoxon rank sum test is shown.
- f) Violin distribution plots (min/max with 25th, 50th, and 75th percentiles delineated by lines) of the shortest islet-to-neoplasia (n=444 islets from n=3 *KC; Lep^{ob/+}* mice and n=271 islets from n=3 *KCO* mice) and neoplasia-to-islet (n=329 neoplastic lesions from n=3 *KC; Lep^{ob/+}* mice and n=271 neoplastic lesions from n=3 *KCO* mice) distances. p-values of Wilcoxon rank sum test are shown.
- g) Cumulative distribution frequencies of the shortest neoplasia-to-islet distances of lesions in (f). The cumulative distributions are significantly different between groups (p<0.0001, Kolmogorov-Smirnov test).

3. The authors set out to study both endocrine (beta cell) contributions to pancreatic tumorigenesis, and the underlying mechanisms driving pro-tumorigenic beta cell function in the context of obesity. However, the authors do not directly link the identified mechanisms of obesity-associated beta cell proliferation/phenotype to PDAC or obesity-associated PDAC. It would strengthen the manuscript to more directly demonstrate whether beta cell stress/proliferation and JNK-driven CCK expression contribute to baseline PDAC and obesity-associated PDAC. To extend these findings to the cancer context, the authors could:

- **Show whether PDAC induces a similar beta cell stress response in vivo at baseline by comparing and quantifying beta cell number or insulin/CCK expression in wild-type, wild-type Akita, KPC and KPC-Akita mice, and an obesity KPC or KCO model. How does baseline PDAC development impact beta cells? It's unclear whether the obesity-driven changes to beta cell biology identified here are similarly present and important for baseline PDAC development. If changes to beta cell state/mass identified in obesity (for example increase in CCK+ beta cells) are not identified in KPC mice, what alternative mechanism explains the role for beta cells in baseline tumor development?**

We thank the reviewer for this suggestion. We have previously reported that islet CCK is not significantly induced in lean *KC* or *KPC* mice². In contrast, CCK is induced in obese *KCO* and *KC;Lepr^{db/db}* mice. CCK expression is abolished by weight loss in *KCO* mice². Thus, β cell CCK expression is a consequence of obesity and not tumor formation. To directly link the altered β cell phenotype in obese mice to obesity-driven PDAC, we have used a new mouse model that enables conditional knockout of CCK (*CCK^{Flox/Flox}*) in the pancreas of *KCO* mice., as described above. We find that pancreas-specific knockout of CCK reduces tumor formation (**Fig. 2**) without reducing body weight, decreasing insulin secretion, or compromising glucose homeostasis (**Extended Data Fig. 2**). Since our scRNA-seq and IF analyses do not identify significant CCK expression in any cells in the pancreas beyond β cells, our data argue that β cell-derived CCK is a *bona fide* driver of obesity-driven pancreatic cancer. The absence of CCK in *KPC* mice argue that decreased baseline tumor development in *KPC-Akita* mice is likely due to

loss of insulin or another β cell hormone in lean conditions. Indeed, *KPC-Akita* mice exhibited reduced β cell mass and insulin secretion relative to *KPC* mice (**Fig. 8b-d**), which aligns with prior observation suggesting that basal insulin is important for tumorigenesis^{3,4}.

• The authors demonstrated an important role for beta cell CCK on obesity-associated tumorigenesis in prior work. However, additional studies in KPC/KPC-Akita mice would validate the role of CCK+ beta cells in baseline PDAC development. The authors could test if CCKR inhibitor or JNKi reduce tumor burden in KPC mice to verify the functional relevance of the new mechanism in vivo and the role of CCK+ beta cells outside of the obesity setting. KPC-Akita mice could be used to control for beta cell-independent effects of inhibitors on PDAC tumors.

These are excellent experimental suggestions. However, as noted above, we do not argue that CCK is promoting tumor development in baseline lean conditions. Instead, our data support a role for basal insulin in tumor formation, consistent with prior studies^{3,4}. In contrast, experiments with conditional genetic loss of *Cck* in the pancreas (**Fig. 2**) demonstrates that β cell CCK is a driver of obesity-associated PDAC. Importantly, CCK loss did not reduce body weight, decrease insulin secretion, or compromise glucose homeostasis (**Extended Data Fig. 2**), arguing the impact on tumorigenesis is not due to an effect on insulin, as we observe in the *KPC-Akita* model. Indeed, in *KCO* mice, CCK expression strongly correlates with disease burden while C-peptide is negatively associated with disease burden. Together, these data establish that β cells play a role in both in lean and obese conditions, though the mechanisms by which they drive tumorigenesis are context-dependent. While it would be ideal to show that JNKi decreases tumor development in *KCO* or *KPC* mice, this experiment cannot be done cleanly. We would not be able to exclude the likely possibility that JNKi would have direct effects on tumor cells themselves leading to a change in tumor phenotype rather than an indirect effect via impacts on β cells.

• Evaluate the effect of obesity on tumorigenesis in the context of beta cell depletion (KPC-Akita). This experiment would delineate the extent to which obesity-driven cancer is dependent on beta cells. Is there a reason the authors did not pursue these studies, or why the KPC-Akita model might not be suitable to answer this question?

We thank the reviewer for this interesting experimental suggestion. Unfortunately, combining *KPC-Akita* with *Lep^{ob/ob}* is not feasible, as this would lead to rapid demise of the animals independent of tumorigenesis due to intractable effects on glucose homeostasis. At minimum, this impact on glucose homeostasis would significantly confound interpretation of the results. Instead, we performed experiments to conditionally knockout CCK in the pancreas in *KCO* mice (**Fig. 2**), as described above, which we believe provides a cleaner method to establish the functional importance of β cell CCK in obesity-driven PDAC development.

Minor comments

1. Could the authors please provide any quantification of beta cell depletion in the KPC-

Akita model (Figure 1b)? It would be helpful to know the extent to which insulin+ beta cells are depleted.

We thank the reviewer for this suggestion. We have now provided quantification of relative β cell mass (based on IHC staining for insulin+ cells on tissue sections) and circulating insulin and C-peptide levels (**Fig. 8b-d**), which demonstrate significant functional depletion of β cells in *KPC-Akita* vs. *KPC* mice.

2. The authors should define “PP cells” in the section on “Dynamic changes in beta cell expression of insulin and CCK with obesity”.

We apologize for this omission. We have now defined pancreatic polypeptide (PP) cells in the text.

3. Reg expression was reduced by CCKRi treatment of Ob/ob mice; did this correspond to a loss of the Cluster 3 Reg+ cells? (Fig. 2i)

As we did not perform single-cell RNA sequencing analyses on these mice, we cannot know for sure. As an orthologous approach, we performed IHC for Reg2 on *KCO* and *KC;Lep^{ob/+};Ins1-CCK/+* mice, consistent with β cell CCK expression inducing peri-islet Reg+ cells (**Fig. 9k**).

Fig. 9. Peri-islet acinar cell transcriptional alterations in obesity.

k) Representative IHC images demonstrate increased peri-islet (synaptophysin-positive) acinar cell expression of Reg2 in β cell CCK-expressing tumor models (*KC;Lep^{ob/+};Ins1-CCK/+* and *KCO*) compared to controls (*KC;Lep^{ob/+}*). Scale bar is 100 μ m.

4. “Thus, Cck+ cells arise from pre-existing beta cells....” In statements regarding the use of bioinformatic tools such as TrajectoryNet it might be more appropriate to soften the language to indicate that these tools really generate a hypothesis as to the lineage or cell of origin instead of truly indicating which population arose from another as illustrated instead by the authors’ later lineage tracing studies.

We thank the reviewer for this suggestion. We have adjusted the text to note that our *in silico* analyses were hypothesis-generating and thereafter confirmed by our genetic lineage tracing experiments, as follows:

“These data raise the possibility that CCK+ cells arise from pre-existing β cells rather than by transdifferentiation of other cell types.”

“Collectively, these findings suggest that a subpopulation of immature β cells with proliferative potential (WT AT 4) give rise to CCK-expressing β cells in obesity.”

We have changed the **Results** text and **Figure** order accordingly to describe *in silico* trajectory analysis in full first and then discuss genetic lineage tracing afterwards as validation.

Reviewer #3 (Remarks to the Author):

The article by Garcia et al. crosses the KPC model of PDAC with the Akita model of beta-cell ablation and shows decreased tumour formation at 12 weeks in the KPC-AKITA model even in the presence of elevated blood glucose, suggested that more available free glucose in the bloodstream isn't driving tumorigenesis but the beta-cells directly. Following this discovery, they switch focus to beta-cell transformation during obesity, which is a risk factor for PDAC development, and find that acinar cells in the peri-islet space have altered phenotype during obesity. They then characterise beta- heterogeneity during obesity, identifying an embryonic/immature beta-cell program in mice driven by JNK/cJun which leads to increased CCK expression, a known PDAC promoter.

Although the topic is highly relevant to better understand exocrine and endocrine pancreas interaction/communication there is need for a reinterpretation of the results and in some cases refine some of the stated conclusions:

Major Comments

- The author consistently jump between figures and the order of graphs within individual figures does not match the in-cite text. Re-ordering either the figures or the in-cite text to match each other would make it a lot easier for the reader to follow the data. This was especially frustrating with figures 3 and 4, and also the out-of-order graphs in extended Fig. 7. Furthermore, their reasoning and logic for using the various bioinformatic tools employed in the paper isn't stated very clearly until the discussion.

We thank the reviewer for this helpful suggestion regarding presentation of the data. We have now reorganized the **Results** text and **Figure** panel order to better align. For example, we describe the results of the *in silico* trajectory analysis in full first (citing **Fig. 3**) and subsequently delineate the genetic lineage tracing experiments as validation (citing **Fig. 4**). We have also split **Fig. 7** by moving some of the data panels into an **Extended Data Fig. 11** to make it more readable. Finally, we have included more details on the choice of bioinformatics tools in the **Results**, while maintaining a paragraph in the **Discussion** to highlight the advances these tools provide over existing methods.

- Authors state, “Despite decreased tumorigenesis, KPC-Akita mice showed markedly elevated glucose levels (Fig. 1d), suggesting that hyperglycemia is not independently pro-tumorigenic in the absence of β cells”. Surprisingly KPC mice have a glucose basal level of 400 mg/dl, these glucose levels are extremely high, could the authors comment on that. 400 vs 600 mg/dl will not allow us to draw any conclusion about the roles of hyperglycemia in PDAC.

We thank the reviewer for the astute observation. *KPC* mice develop hyperglycemia over time likely due primary pancreatic disease invading and ablating functional islets and possible indirect effects on β cell function¹, so called Type 3c diabetes in humans. Although the *KPC-Akita* mice are universally hyperglycemic, this results in a non-significant increase in blood glucose levels compared to *KPC* mice at the endpoint. We have now included serial non-fasting glucose levels

that demonstrate differences in the kinetics of hyperglycemia between *KPC* and *KPC-Akita* mice (**Fig. 8e**). Hyperglycemia occurs much earlier in *KPC-Akita* mice. Overall, the *KPC-Akita* mice have higher and more prolonged hyperglycemia than *KPC* mice but have reduced tumor burden, which agrees with the conclusion that hyperglycemia *per se* is not the primary driver of PDAC.

Fig. 8. Functional depletion of β cells suppresses pancreatic tumorigenesis

e) Serial non-fasting blood glucose measurements (mean \pm SEM, n=4-9 mice per timepoint per group with sex denoted by symbols) of *KPC* and *KPC-Akita* mice. Maximum glucometer measurement is 600 mg/dL). P-value of mixed-effects model with Geisser-Greenhouse correction and Tukey's post-hoc test are shown.

• In extended figure 3. The authors show the single cell data of the 3 mouse models analyzed, but the number of cells per model is different. A proportion plot per model will be helpful in this regard as many conclusions are based on this single cell data.

We thank the reviewer for this comment. We have added two new subpanels reflecting the number of cells per mouse model. First, we have added the number of endocrine cells per condition, subdivided by endocrine cell types to **Extended Data Fig. 4a** (right). Further, we have replaced **Extended Data Fig. 5b** with the number of β cells per condition, subdivided by archetypes.

Extended Data Fig. 4. Dynamic changes in β cell expression of *insulin* and *Cck* with obesity.
 a) Left: Combined embedding of endocrine cells from three conditions (WT, HFD, *Lep^{ob/ob}*) and annotated clusters based on marker genes. Right: endocrine cell type counts per condition.

Extended Data Fig. 5. Analysis of β cell heterogeneity within WT, HFD, and *Lep^{ob/ob}* embeddings.

b) Count of cells for each condition and archetype.

• As the Akita model causes inflammatory cell death of beta-cells, can the authors be confident that there isn't an artificial recruitment of immune cells to the pancreas, which alters the tumour microenvironment of the pancreas, therefore priming it to remove early-stage tumours, and therefore the results they achieve is immune dependent and not beta-cell dependent?

We apologize if we were not clear in our explanation of the *Akita* model. β cells are lost through non-inflammatory β cell death in *Akita* mice, which motivated the selection of this model for this experiment. This contrasts with immune-mediate β cell destruction in other models (e.g., NOD mouse). We confirmed the lack of islet inflammation by staining for Cd45+ immune cells in *KPC* and *KPC-Akita* mice and observed no differences in intact islets (Fig. 8b). More broadly,

the *Akita* model does not itself exhibit exocrine pancreatitis. Finally, it is not clear what antigens T cells would be responding to for clearance of PanIN lesions, if there were to be T cell-dependent clearance. Prior studies in *KPC* mice have showed few single-nucleotide variants even in advanced tumors in mice⁵. While both mutant *Kras* and *p53* may generate neopeptides for immune recognition^{6,7}, antigen-specific T cells would be predicted to be deleted by central tolerance despite their expression being restricted by a LSL cassette, as has been observed previously⁸. Therefore, it is unlikely that the reduced tumor phenotype in the *Akita* model is due to immune-mediated clearance of PanINs.

Fig. 8. Functional depletion of β cells suppresses pancreatic tumorigenesis

b) H&E and IHC stains show a reduction in the proportion of insulin-expressing islet cells and increased ER stress (BiP) without alterations in CD45+ immune cell infiltration of islets (labeled by pan-neuroendocrine marker synaptophysin) of *KPC-Akita* mice. Scale bar is 50 μ m.

• **What would further strengthen the authors argument that glucose isn't driving PDAC tumour development is a time course of glucose levels over the disease progression of the two models *KPC* and *KPC-AKITA*. Moreover, further investigation of the role of insulin signalling pathway will also help elucidate the main driver of this conclusion: immature beta cell phenotype is activated by ER stress, glucose changes or insulin ones.**

We thank the reviewer for this excellent suggestion. As noted above, we performed a time course analysis of glucose showing early hyperglycemia in *KPC-Akita* mice (**Fig. 8e**). We now also

show that *KPC-Akita* mice exhibit significantly reduced insulin secretion (circulating insulin and C-peptide) relative to *KPC* mice concordant with reduced β cell mass (**Fig. 8b-d**). Our data argue that decreased baseline tumor development in lean *KPC-Akita* mice is likely due to loss of insulin, which aligns with prior observation suggesting that basal insulin is important for tumorigenesis^{3,4}. In contrast, in obese conditions, we find that CCK has an insulin-independent role in driving obesity-associated PDAC progression. Specifically, we have used a new mouse model that enables conditional knockout of CCK (*CCK^{Flox/Flox}*) in the pancreas of *KCO* mice (**Fig. 2**). We find that pancreas-specific knockout of CCK reduces tumor formation (**Fig. 2**) without altering body weight or glucose homeostasis (**Extended Data Fig. 2**). Since our scRNA-seq and IF analyses do not identify significant CCK expression in any cells in the pancreas beyond β cells, our data argue that β cell-derived CCK is a *bona fide* driver of obesity-driven pancreatic cancer. We further confirmed that β cell CCK overexpression is sufficient to promote pancreatic tumorigenesis (**Fig. 1**). Finally, our results establish that CCK – rather than insulin – more strongly correlates with disease burden across our models (**Fig. 1l,m and Fig. 2g,h**).

Fig. 8. Functional depletion of β cells suppresses pancreatic tumorigenesis

- c) Quantification of proportion β cells area (insulin-positive) of total islet area (synaptophysin-positive) in *KPC* and *KPC-Akita* mice. Mean \pm SEM (n=4-6 mice per group with sex denoted by symbols) and p-value of Welch's t-test are shown.
- d) Final non-fasting insulin and C-peptide levels (mean \pm SEM, n=6-10 mice per group with sex denoted by symbols). p-values of Welch's t-test are shown.

Figure 1. β cell CCK expression promotes PDAC development.

- l) Pancreatic CCK expression is positively correlated with disease burden in *KCO* mice. Each point represents one mouse (n=18). Pearson correlation coefficient (r) and p-value are shown.
- m) Terminal c-peptide levels are inversely correlated with disease burden in *KCO* mice. Each point represents one mouse (n=18). Pearson correlation coefficient (r) and p-value are shown.

Figure 2. Pancreatic CCK is required for obesity-associated PDAC progression.

- g) Pancreatic CCK expression is positively correlated with disease burden of mice in **(b)**. Each point represents one mouse with *Cck* genotype denoted ($n=35$). Pearson correlation coefficient (r) and p -value are shown.
- h) Terminal c-peptide levels are inversely correlated with disease burden of mice in **(b)**. Each point represents one mouse with *Cck* genotype denoted ($n=35$). Pearson correlation coefficient (r) and p -value are shown.

• **The study has focused on acinar derived PDAC, but they frequently state PDAC as a whole, ignoring ductal derived PDAC. A more nuance discussion specifying acinar vs ductal origin is needed throughout.**

We thank the reviewer for bringing up this important point. The cell-of-origin for PDAC in humans remains controversial, and definitive evidence for an acinar vs. ductal origin is lacking. We focus on acinar cells because the preponderance of mouse studies supports an acinar cell-of-origin for PDAC^{9,10}. We have clarified this point in the manuscript by describing ductal-derived PDAC in a new limitations paragraph at the end of the **Discussion**.

• **In the Peri-islet acinar cell adaptation section drawing any conclusion based on 260 acinar cells is an overstatement. Moreover, assuming that the acinar cells come from peri-islet region exclusively is not right as the islet preps contain also acinar contaminants not located in the peri-islet region. A detailed validation of this section is needed at list at the level of IF or similar. Tosto et al. already described 3 acinar populations in the human pancreas. The authors should look at Tosti or other scRNAseq data of acinar cells as well.**

We thank the reviewer for bringing up this point. Although the small number of acinar cells recovered suggests the islet preparations were very pure and thus recovered acinar cells are more likely to be peri-islet, we agree this cannot be determined with certainty. We have changed the **Results** text to reflect this:

“we suspected that these cells were enriched in those found in the peri-islet area, as they were obtained from highly pure islet isolations and represented ~1% of sequenced cells.”

However, we describe several additional lines of evidence to support a peri-islet origin of these acinar cells, their similarity to human peri-islet acinar cells, and their association with β cell CCK expression. We have included in the revised **Results** and **Figures**:

- 1) Transcriptional signatures in obese (*Lep^{ob/ob}*, HFD) single acinar cells are enriched in previously published murine peri-islet signatures (**Fig. 9b**)
- 2) As noted by the reviewer, Tosti et al. described a Reg⁺ population by snRNA-seq that was found in the peri-islet region of the human pancreas¹¹. We find significant enrichment for the Reg⁺ acinar gene signature from Tosti et al. in our Reg⁺ cluster (cluster 3) (**Fig. 9i**)
- 3) We performed IHC for Reg2 on *KCO* and *KC;Lep^{ob/+};Ins1-CCK* mice, consistent with β cell CCK expression inducing peri-islet Reg⁺ cells (**Fig. 9k**).

Fig. 9. Peri-islet acinar cell transcriptional alterations in obesity.

- a) PHATE visualization of *Cpa1*⁺/*Prss2*⁺ acinar cells coded by sample condition.
- b) Enrichment of gene signatures observed in bulk RNA-seq of peri-islet acinar cells from *Lep^{db/db}* mice (\log_2FC expression > 1.5, FDR < 0.05)¹² in obesity models. Box plots display 25th, 50th, and 75th percentile enrichment scores +/- 1.5 interquartile range (IQR) for each cell in each condition. *p*-value is derived from one-sided Wilcoxon rank sum test.
- c) Proteases are upregulated in HFD/*Lep^{ob/ob}* (n=143 cells) versus WT (n=117 cells) acinar cells. Violin plots represent gene expression distribution (min/max with 25th, 50th, and 75th percentiles delineated by lines). *q*-values of Wilcoxon rank sum test adjusted for multiple comparisons with two-stage Benjamini, Krieger, Yekutieli step-up are shown. Notably, *Cela3b*, *Cpb1*, and *Ctrb1* are induced by CCK in AR42J acinar cell carcinoma cells and *Try4*, *Try5*, and *Try10* were shown to be enriched in peri-islet acinar cells in *Lep^{db/db}* mice¹².
- d) PHATE visualization of *k*-means clustering of acinar cells in (a) derived from lean (WT) vs. obese (HFD/*Lep^{ob/ob}*) conditions.
- e) Differential proportions of acinar cell clusters in (d) *p* < 0.0001, chi-square test.
- f) Heatmap of row normalized mean expression of *Tff2*, proteases, and *Reg* genes across cells within each cluster in (d).
- g) PHATE visualization and violin plots of *Tff2* expression in acinar cell clusters. *p*-values of Kruskal-Wallis test with Dunn's post-hoc test are shown.
- h) Violin plots of *Reg* gene expression distribution (min/max with 25th, 50th, and 75th percentiles delineated by lines). *q*-values of Wilcoxon rank sum test adjusted for multiple comparisons with two-stage Benjamini, Krieger, Yekutieli step-up are shown.
- i) Enrichment of the human peri-islet acinar-REG⁺ differential gene expression signature derived from Tosti et al¹² for each murine acinar cell cluster. *p*-value of Wilcoxon rank sum test is shown.
- j) Violin plots of pancreatic *Reg* gene expression by RNA-seq of 12-week-old C57/B6 WT (n=5), *Lep^{ob/ob}* (n=7), and *Lep^{ob/ob}* mice treated with CCK receptor antagonists (proglumide, lorglumide; n=2 mice/drug) for 6 weeks. *q*-values of Wilcoxon rank sum test adjusted for multiple comparisons with two-stage Benjamini, Krieger, Yekutieli step-up are shown.
- k) Representative IHC images demonstrate increased peri-islet (synaptophysin-positive) acinar cell expression of *Reg2* in β cell CCK-expressing tumor models (*KC;Lep^{ob/+};Ins1-CCK/+* and *KCO*) compared to controls (*KC;Lep^{ob/+}*). Scale bar is 100 μ m.

• In the section of “identification of the cell-of-origin for pro-tumorigenic β cells in obesity”, the authors introduce the possibility of several cells of origin of new beta cells but then just analyze beta cell as possible pro-tumorigenic β of origin. Further elaborating this decision is important.

We thank the reviewer for the comment. In addition to β cells, we also experimentally assessed α cells with the *Gcg-Cre^{ERT2}* tracing experiments because they were the most plausible cells (beyond β cells) to be the cell-of-origin for CCK⁺ β cells. This is based on multiple prior studies demonstrating a potential α cell origin^{13,14}, the proximity of α cells to β cells (compared to duct cells) in gene expression space (**Extended Fig. 4a**), and the availability of specific *Cre^{ERT}* drivers. Our goal was not to do an exhaustive lineage tracing experiment, which falls beyond the scope of the current manuscript, but instead validate our *in silico* findings demonstrating a β cell

(specifically and Ngn3+ subpopulation) origin. We have reorganized our **Results** and **Figure** panel order to reflect this relationship between the *in silico* and experimental lineage tracing studies.

• In extended Fig 4a, the authors have quantified the amount of TdTom+ cells that are glucagon positive, the same should be performed for the *Ins1-CreERT* and *Ngn3-CreERT* models in Supp Fig 4b, given the incomplete labelling hypothesis presented in the paper. I also think an explanation for the inefficient labelling should be stated in-text or hypothesised on. The interpretation of *Ngn3-CreERT* labelling results are unclear. Moreover, as far as it is known, the number of Neurog3 positive cells after weaning is extremely low, thus not sure about the labelling, are those cells in the islets or ducts? Several papers argue about it, a detailed characterization of the model and % of positive cells is needed.

We thank the reviewer for the comments and suggestions. Quantification of co-immunofluorescence for **Extended Data Fig. 7b** (*Ins1-CreERT* and *Ngn3-CreERT*; previously **Extended Data Fig. 4b**) was incorporated into the graphs in **Fig. 4b** (5 weeks: columns 1, 3, and 5). Incomplete labeling for *Ins1-CreERT* has been well-described and is likely due to the intrinsic inefficiency of the Cre driver¹⁵. We now describe this in the text as follows:

*“The incomplete labeling of CCK+ cells in *Ins1-Cre^{ERT}* mice was likely due to inefficient labeling of β cells with this *Cre^{ERT}* line in obese mice (**Extended Data Fig. 7b**), which has been previously observed in lean mice¹⁶. In support of this hypothesis, we discovered a positive linear correlation between TdTomato+/insulin+ cells (a marker of labeling efficiency) and TdTomato+/CCK+ cells (**Extended Data Fig. 7c,d**).”*

In the *Ngn3-Cre^{ERT2}* line, tdTomato+ cells 1 week after tamoxifen administration are located in the islet and also express insulin, consistent with β cell labeling. Our data on the degree of Ngn3 labeling is consistent with a prior study using the same *Cre^{ERT}* line with tamoxifen given at the same timepoint¹⁷. We cannot exclude that the *Ngn3-Cre^{ERT2}* line has some leakiness, as noted by the reviewer. However, the much sparser labeling observed with this line compared to *Ins1-Cre^{ERT}* and the expansion of the tdTomato+ cells specifically in the obese model validate our *in silico* findings that the *Ngn3-Cre^{ERT2}* line is labeling a regenerative population concordant with our Ngn3-expressing WT AT4 population. We have included a discussion of this in our new limitations paragraph at the end of the **Discussion**:

*“...experimental lineage tracing studies may not be conclusive due to the lack of specific cell type markers and the potential for unforeseen leakiness of *Cre^{ERT}* lines. For example, to label WT AT 4 cells, we performed lineage tracing with the most readily available line with the greatest specificity for this archetype, *Ngn3-Cre^{ERT2}*. Prior studies in older lean adult mice that suggest that *Ngn3+* β cells are rare and do not contribute significantly to β cell neogenesis during homeostasis¹⁸. In contrast, we observed that obesity is associated with expansion and adaptation of *Ngn3+* β cells to express CCK (**Fig. 4**). However, we cannot exclude the possibility that we are labelling a larger islet population than WT AT 4 with this *Cre^{ERT}* line. Nonetheless, our findings are supported by orthogonal *in silico* analyses (**Fig. 3**) and are*

concordant with a recent lineage tracing study which showed the expansion of *Ngn3*⁺ cells in response to a high-fat fast mimicking diet to comparable levels as we observed in *Lep*^{ob/ob} mice at similar timepoints¹⁹. These data argue that obesity results in the expansion of *Ngn3*⁺ cells independent of model or diet.”

Extended Data Fig. 7. Validation of *Cre*^{ERT} lines for *in vivo* lineage tracing.

- a) Representative images of Gcg immunofluorescence and lineage-traced TdTomato⁺ cells in 5-week-old *Gcg-Cre*^{ERT2}; *Lep*^{ob/ob}; *Rosa26*^{LSL-TdTomato} mice administered tamoxifen at 4 weeks of age. DAPI labels nuclei blue. Average percentage (mean \pm SEM, n=3 mice, sex designated by symbols) of Gcg⁺ cells labeled with TdTomato in mice is shown showing near complete overlap.

- b) Representative images of insulin immunofluorescence and lineage-traced TdTomato⁺ cells in 5-week-old *Ins1-Cre^{ERT}; Lep^{ob/ob}; Rosa26^{LSL-TdTomato}* and *Ngn3-Cre^{ERT2}; Lep^{ob/ob}; Rosa26^{LSL-TdTomato}* mice administered tamoxifen at 4 weeks of age. DAPI labels nuclei blue.
- c) Strong positive correlation between total labeled insulin⁺/TdTomato⁺ and CCK⁺/TdTomato⁺ cells in *Ins1-Cre^{ERT}; Lep^{ob/ob}; Rosa26^{LSL-TdTomato}* mice in (c). Pearson correlation coefficient (r) and p-value are shown.
- d) Representative images of insulin immunofluorescence and lineage-traced TdTomato⁺ cells in 5-week-old and 16-week-old lean *Ngn3-Cre^{ERT2}; Lep^{ob/+}; Rosa26^{LSL-TdTomato}* mice administered tamoxifen at 4 weeks of age. DAPI labels nuclei blue.

Fig. 4. Ngn3⁺ immature β cells expand to give rise to CCK-expressing β cells.

- b) Average percentage (mean \pm SEM., n=2-10 mice per group, sex is designated by symbols) of insulin⁺ cells labelled with TdTomato in obese *Ins1-Cre^{ERT}; Lep^{ob/ob}; Rosa26^{LSL-TdTomato}*, obese *Ngn3-Cre^{ERT2}; Lep^{ob/ob}; Rosa26^{LSL-TdTomato}*, and lean *Ngn3-Cre^{ERT2}; Lep^{ob/+}; Rosa26^{LSL-TdTomato}* mice analyzed at 5 (1 week post-tamoxifen) or 16 weeks (12 weeks post-tamoxifen). p-values of Welch's t-test are shown.

• The reasoning for choosing *Lep^{ob/ob}* AT2 to examine in TrajectoryNet out of the different ATs in group 2 is not clear. By choosing that as the endpoint, are you then forcing its evolutionary trajectory? Explanation is needed as to why the authors have chosen this comparison over other archetypes.

We apologize for not making the reason for this choice clearer. *Lep^{ob/ob}* AT2 was chosen as the endpoint because it had the highest CCK expression. We have clarified this in the text as follows:

*“We next used learned cellular dynamics from TrajectoryNet to decipher which β cell archetypes are most likely to be the cell-of-origin for the highest Cck-expressing (Cck-hi) *Lep^{ob/ob}* cells (*Lep^{ob/ob}* AT 2; Fig. 3b)”*

• Extended Fig. 6 has not been expanded upon in the text. Please explain these results in more detail.

Thank you for pointing out this omission. We have abbreviated **Extended Data Fig. 6** into a single panel (now **Extended Data Fig. 7e**) and described this more clearly in the results as follows:

“Neither *TdTomato*⁺ nor *CCK*⁺ β cells in the *Ngn3-Cre^{ERT2}* or *Ins1-Cre^{ERT}* lines at 16 weeks of age were *Ki67*⁺ (**Extended Data Fig. 7e**), suggesting that islets cells were largely post-mitotic at this time point.”

Extended Data Fig. 7. Validation of *Cre^{ERT}* lines for *in vivo* lineage tracing.

e) Co-immunofluorescence for insulin and *Ki67* of islets from 16-week-old *Ins1-Cre^{ERT}; Lep^{ob/ob}; Rosa26^{LSL-TdTomato}* and *Ngn3-Cre^{ERT2}; Lep^{ob/ob}; Rosa26^{LSL-TdTomato}* mice shows no *Ki67*⁺ cells within the islet (DAPI: blue, insulin: green, TdTomato: red, *Ki67*: magenta). Small intestine from 16-week-old *Ins1-Cre^{ERT}; Lep^{ob/ob}; Rosa26^{LSL-TdTomato}* mice demonstrates *Ki67*⁺ cells in the crypts as a positive control. DAPI labels nuclei blue.

• In Fig.6b, WT AT 4 is only represented in 3/15 ND patients. This is inconsistent with the mouse data presented earlier, and questions whether your proposed cell of origin is conserved in humans.

We apologize that the proportion of patients that harbor specific archetypes is not clear in Fig. 6b. We have now provided the percentage of patients harboring cells mapped to each archetype broken down by sex or diabetes status in **Extended Data Fig. 10**. WT AT 4 is represented in 7/15 ND patients (46.67%) and 2/9 T2D patients (22.22%).

Extended Data Fig. 10. Percentage of patients harboring cells assigned to each archetype.

- Percentage of male and female patients harboring each archetype (AT).
- Percentage of ND (non-diabetic) and T2D (type II diabetic) patients harboring each AT.

• **Does treatment with the JNKi inhibitor result in decreased tumour formation in obese PDAC models? While the embryonic program stimulated in B-cells during pharmacological (STZ) stress and obesity is interesting, the data isn't related back to the PDAC model in the first figure.**

While it would be ideal to show that JNKi alters tumor progression in *KCO* mice, this experiment cannot be done cleanly, as we would not be able to exclude the likely possibility that JNKi would have direct effects on tumor cells themselves leading to the phenotype (rather than due to a reduction in β cell CCK expression). The embryonic program stimulated in β cells with STZ and its concordance with obesity demonstrates that both physiologic (obesity) and pharmacologic (STZ) β cell stress induce similar transcriptional profiles including CCK expression. To directly link β cell CCK expression to obesity-associated tumorigenesis, we have used a new mouse model that enables conditional knockout of CCK (*CCK^{Flox/Flox}*) in the pancreas of *KCO* mice (Fig. 2). We find that pancreas-specific knockout of CCK reduces tumor formation (Fig. 2) without altering body weight or glucose homeostasis (Extended Data Fig. 2). Since our scRNA-seq and IF analyses do not identify significant CCK expression in any cells in the pancreas beyond β cells, our data argue that β cell-derived CCK is a *bona fide* driver of obesity-driven pancreatic cancer.

Minor Comments

- Overall, the authors concluding statements at the end of every results section are very vague, and a more informative summary of each section's results would make it easier for the reader to interrupt and follow their conclusions.

We thank the reviewer for this constructive suggestion. We have edited the concluding statements at the end of each results section to more directly state the main conclusions.

- In Fig.1b, the number of mice used to ascertain that insulin was decreased and ER stress induced in the KPC-Akita model relative to KPC is not stated in the figure legends or methods. Quantification of the insulin and BiP positive area within the islets would help support the conclusions the authors have made in the manuscript “We confirmed decreased β cell mass and induction of ER stress in islets of KPC-Akita mice “and therefore their conclusions that B-cell loss impedes PDAC tumorigenesis.

We thank the reviewer for this suggestion. We observed BiP expression in most islets in *KPC-Akita* but not *KPC* mice (n=4-6 mice/group), demonstrating that ER stress does occur specifically in *Akita* mice. We did not quantify the area of BiP expression, because it is not clear it directly leads to the tumor phenotype, as opposed to insulin and C-peptide, which are more appropriate functional readouts of β cells, as the reviewer suggests. Therefore, we have now provided quantification of relative β cell mass (based on area of IHC staining for insulin+ cells on tissue sections) and circulating insulin and C-peptide levels (**Fig. 8b-d**), which demonstrate significant functional depletion of β cells in *KPC-Akita* vs. *KPC* mice. We therefore now conclude:

“These data showcase basal role for functional β cells in PDAC development, even in lean mice.”

Fig. 8. Functional depletion of β cells suppresses pancreatic tumorigenesis

- Schematic of the alleles used to generate *KPC* and *KPC-Akita* mice (generated with Biorender).
- H&E and IHC stains show a reduction in the proportion of insulin-expressing islet cells and increased ER stress (BiP) without alterations in CD45+ immune cell infiltration of islets (labeled by pan-neuroendocrine marker synaptophysin) of *KPC-Akita* mice. Scale bar is 50 μ m.
- Quantification of proportion β cells area (insulin-positive) of total islet area (synaptophysin-positive) in *KPC* and *KPC-Akita* mice. Mean \pm SEM (n=4-6 mice per group with sex denoted by symbols) and p-value of Welch's t-test are shown.
- Final non-fasting insulin and C-peptide levels (mean \pm SEM, n=6-10 mice per group with sex denoted by symbols). p-values of Welch's t-test are shown. Sex of each mouse is denoted by symbols.

- **In Figure 1C, it is not stated in methods how the authors quantified tumour burden in the KPC and KPC-Akita models.**

We apologize for this omission. We have more carefully delineated how disease burden is calculated across all models in the **Methods** as follows:

“Quantification and statistical analysis of tumor development

All disease analyses were performed in a blinded manner on coded scanned slides in QuPathv0.6.0. Disease burden was quantified by measuring cross-sectional area of disease (ADM, PanINs, or PDAC including stroma) relative to total pancreas area. Histological analysis of tumor staging was performed and confirmed by two independent reviewers (C.F.R and M.D.M.).”

- **The author writes, “Having established that β cells are basally required for exocrine tumorigenesis, ...”. This is a very strong conclusion, as the KPC-Akita mice still have 40% tumour burden. I think promote would be a more accurate interpretation of the author’s results.**

We thank the reviewer for this comment. We have removed this phrase. Instead, we repositioned the *KPC-Akita* data later in the manuscript to better align with the story (including new gain- and loss-of-function CCK studies *in vivo*) and concluded the section with:

“These data showcase a basal role for functional β cells in PDAC development, even in lean mice.”

- **In methods, it is stated that only male mice were used for STZ and scRNAseq experiments due to the reduced beta-cell effects of STZ and a High Fed Diet. Please comment on whether these results are then relevant for human female patients.**

This is an important point raised by the reviewer. The induction of β cell CCK by obesity occurs in both male and female mice in both tumor-bearing (**Fig. 1b,c** and **2b,c**) and non-tumor-bearing (**Fig. 4f**) conditions. Furthermore, all functional studies in *KCO* mice and new *in vivo* experiments gain- and loss-of-function of β cell CCK expression include mice of both sexes (**Fig. 1-2**). No sex-dependent differences were observed, and sex of each mouse is denoted by corresponding symbols. Finally, as previously shown in **Figure 6e**, the mapped obesity trajectory shows no association with sex (Pearson $R=-0.17$), and all archetypes observed in scRNA-seq data in mice are found in both human males and females (new **Extended Data Fig. 10a**). Therefore, we believe that the vast majority of our results in mice would be relevant for female patients.

- **In Fig.3D and the accompanying text, as you heavily discuss the biological functions of the groups of archetypes, I would rearrange the heatmap on Fig. 3D to follow the evolutionary tree in ext. Fig.5 and then colour code them by WT, HFD and Lepob/ob to make it easier for the reader to follow your conclusions.**

We thank the reviewer for this suggestion. We have rearranged the heatmap in **Fig. 3b** based on the similarity tree in **Extended Data Fig. 5**, maintaining the expression scaled per condition in order to easily identify how each archetype differs from other archetypes in a similar setting.

Fig. 3. *In silico* identification of the cell-of-origin for CCK-expressing β cells in obesity
 b) Heatmap (row-normalized within each condition) for marker genes expression for each archetype.

• In Fig.5J, you only show the mapping of WT AT 4 and OB/OB AT 2. What about the other archetypes, these should be included in the supplementary/extended figures.

We thank the reviewer for this comment. We have now adjusted **Fig. 5j** to include all archetypal mappings.

Fig. 5: Obesity is associated with transcriptional signatures of increased oxidative and endoplasmic reticulum stress in β cells

j) Visualization of vehicle and mSTZ cells mapped to all archetypes.

Reviewer #4 (Remarks to the Author):

The paper presents an interesting study with extensive analyses. However, the current structure makes it somewhat confusing. The experiments are not clearly presented in a hypothesis-driven manner, which makes it difficult to follow the logical flow of the study. Additionally, the bioinformatics methodology needs more detail, particularly regarding the parameters used and how they were selected.

We thank the reviewer for recognizing the merits of our study and for the constructive suggestions. We have included new data that directly links β cell CCK expression to obesity-driven pancreatic tumorigenesis (**Fig. 1-2**). We have now readjusted the structure of the story to begin by addressing the hypothesis that β cell CCK is a *bona fide* driver of obesity-associated PDAC progression (**Fig. 1-2**) followed by defining the cellular and molecular mechanisms that lead to β cell CCK expression (**Fig. 3-7**). We close by testing for a basal role for β cells, even in lean mice (**Fig. 8**) and establishing transcriptional and function of β cell alterations on the exocrine pancreas that govern local peri-islet tumorigenesis (**Fig. 9-10**). We have addressed the concerns regarding more details on the bioinformatics methodology below.

Major Concerns

Tumorigenesis in KPC-Akita Mice (Figure 1)

Since tumors are still observed in KPC-Akita mice, can the authors definitively conclude that β -cell loss impedes pancreatic tumorigenesis? This claim should be carefully re-evaluated.

We thank the reviewer for this comment. Given additional data showing functional depletion of β cells in *KPC-Akita* mice (**Fig. 8b-e**), we have reworded our conclusion to state:

“These data showcase a basal role for functional β cells in PDAC development, even in lean mice.”

Fig. 8. Functional depletion of β cells suppresses pancreatic tumorigenesis

- Schematic of the alleles used to generate *KPC* and *KPC-Akita* mice (generated with Biorender).
- H&E and IHC stains show a reduction in the proportion of insulin-expressing islet cells and increased ER stress (BiP) without alterations in CD45+ immune cell infiltration of islets (labeled by pan-neuroendocrine marker synaptophysin) of *KPC-Akita* mice. Scale bar is 50 μ m.
- Quantification of proportion β cells area (insulin-positive) of total islet area (synaptophysin-positive) in *KPC* and *KPC-Akita* mice. Mean \pm SEM (n=4-6 mice per group with sex denoted by symbols) and p-value of Welch's t-test are shown.
- Final non-fasting insulin and C-peptide levels (mean \pm SEM, n=6-10 mice per group with sex denoted by symbols). p-values of Welch's t-test are shown.
- Serial non-fasting blood glucose measurements (mean \pm SEM, n=4-9 mice per timepoint per group with sex denoted by symbols) of *KPC* and *KPC-Akita* mice. Maximum glucometer

measurement is 600 mg/dL). P-value of mixed-effects model with Geisser-Greenhouse correction and Tukey's post-hoc test are shown.

Single-Cell Data Generation and Experimental Design

The comparison between congenic obesity models (HFD-fed for 10 weeks, Lepob/ob) and age-matched WT controls raises concerns. The authors should clarify how they accounted for age-related differences to isolate the effects of the conditions studied.

We thank the reviewer for this critique. The differences between models were driven by the effects of the conditions studied. By visualization, β cells from the two WT samples embed near each other and within the same part of the obesity trajectory, whereas the HFD and *Lep^{ob/ob}* samples embed in distinct regions of the obesity trajectory. To quantify this observation, we calculated the pairwise Earth Mover's Distance (EMD) between the four samples where shorter distance is associated with greater similarity. We found that that the two WT samples (16-week and 12-week) were more similar to each other than to their respective age-matched samples (16-week HFD-fed and 12-week *Lep^{ob/ob}* sample, respectively). Furthermore, Both WT samples were closer to the HFD sample than to the *Lep^{ob/ob}* sample, reflecting the transcriptional dynamics described in our work. We have included added this analysis to the **Methods** as part of our preprocessing steps for scRNA-seq analysis:

*“To verify transcriptional differences between models were driven by condition-specific effects (WT vs. HFD-fed vs. *Lep^{ob/ob}*) rather than batch effect or age-based differences, we calculated pairwise Earth Mover's Distances (EMD) between the four samples, where a smaller distance reflects a higher similarity between cells in each sample. We confirmed the two WT samples (12- and 16-week-old) were more similar to each other (EMD=0.45) than to their respective age-matched samples from obese (EMD=0.80 and 0.76). Furthermore, both samples were closer to the HFD-fed sample than the *Lep^{ob/ob}* sample (EMD=0.63 vs. 0.76 for 12-week-old and 0.80 vs. 1.26 for 16-week-old), also verifying lack of batch effect. Given the similarity between WT samples, we merged cells from both samples for all downstream analyses.”*

Clustering and Data Quality (Figure 2 Extended Data)

Cluster 6 appears to express multiple markers from different cell types. How was cell-type annotation performed to ensure accuracy?

Our dotplot was constructed using marker genes derived from the Mouse Islet Atlas²⁰, and annotation of clusters was performed based on strong enrichment of marker genes. As cluster 6 was enriched for stellate markers and Schwann cell marker *Cryab*, we extended the list of Schwann cell markers from a human and mouse pancreatic islet map²¹, adding Schwann cell markers *SI100b*, *Ngfr*, and *Pmp22*. We identified a subpopulation of cluster 11 as strongly enriched for these markers, so we divided cluster 11 into two clusters and annotated the clusters based on the final marker list. This new dot plot confirms cluster 6 is enriched for stellate cell marker genes, and cluster 20 is enriched for Schwann cell marker genes (revised **Extended Data Fig. 3f,g**). Importantly, due to the focus of this study on β cell adaptation, this reannotation does not change the findings of this work. We have added these details to the **Methods** section:

“All cells were then clustered with *k*-means clustering ($n_clusters=20$) and clusters were annotated based on published mouse and human islet marker genes²⁰ to distinguish endocrine cells and β cells for downstream analysis... Cluster 11 was further split into clusters 11 and 20 due to enrichment of Schwann cell markers *S100b*, *Ngfr*, and *Pmp22* in cluster 20. This clustering was chosen due to stability over five runs (average pairwise ARI=0.86) and balance between overclustering and merging different cell types.”

Extended Data Fig. 3. TrajectoryNet enables the identification of the cell-of-origin for aberrant β cell expression of cholecystokinin.

- f) PHATE embedding of exocrine and endocrine islet cells, colored by cluster annotation.
- g) Dot plot of cluster expression for marker genes pertaining to islet populations. Clusters 0, 5, 8, 15, 16, 18, and 19, were annotated as β cells; 2 as polyhormonal (insulin and another hormone); 9 as δ and PP cells; 3 as α cells; 12 as duct cells; 7 and 11 as endothelial cells; 4 as acinar cells; 1, 10, 17 as immune cells; 6 as stellate-activated cells; 20 as Schwann cells; and 13, 14 as low-quality clusters.

What were the nFeature thresholds and percentage of mitochondrial genes for clusters 13 and 14, which were considered low-quality?

We annotated clusters 13 and 14 as low quality due to several markers of quality control. Cluster 13 showed low overall counts (**Reviewer Fig. 1a**), strong enrichment for hemoglobin genes (**Reviewer Fig. 1b**), and low ribosomal counts (**Reviewer Fig. 1c**), indicative of contaminating erythroid cells. Cluster 14 showed low ribosomal counts (**Reviewer Fig. 1c**) and enrichment primarily for markers such as *Malat1* (**Reviewer Fig. 1d**), *Neat1*, and *Gm42418*, lncRNAs whose high expression reflects high nuclear fraction and cytoplasmic RNA loss due to plasma membrane damage²². Neither of these clusters were enriched for key markers of pancreatic islet cell populations, and embedded uniquely in the visualization space, suggesting distinct gene

expression signatures compared to the cells that passed quality control. We have added these details to the **Methods** section:

*“All cells were then clustered with k-means clustering ($n_clusters=20$) and clusters were annotated based on published mouse and human islet marker genes²⁰ to distinguish endocrine cells and β cells for downstream analysis. Clusters 13 and 14 were annotated as low-quality clusters due to lack of enrichment for key markers of pancreatic islet cell populations and distinct embedding visualization. Cluster 13 showed low overall counts, strong enrichment for hemoglobin genes, and low ribosomal counts, indicative of contaminating erythroid cells, and cluster 14 showed low ribosomal counts and enrichment primarily for marks such as *Malat1*, *Neat1*, and *Gm42418*, lncRNAs whose high expression reflects high nuclear fraction and cytoplasmic RNA loss due to membrane damage²².”*

Reviewer Fig. 1. Quality-control metrics of clusters in scRNA-seq analysis.

- a) Log-normalized total counts for each cell per cluster
- b) Total counts for hemoglobin genes for each cell per cluster
- c) Total counts for ribosomal genes for each cell per cluster
- d) Total *Malat1* counts for each cell per cluster

Methods: Data Integration and Batch Effect Control

How was data integration performed, and how were batch effects controlled?

Which covariates were used to integrate the datasets and minimize technical noise?

Data integration was not necessary for the three mouse models as the samples were sequenced in parallel. We show the samples do not show strong batch effect based on the visualization and Earth Mover’s Distance data provided above, which confirms that the two WT samples embed closer together than to their age-matched counterparts. We note this in the **Methods**. For

comparison with other published datasets, including vehicle and mSTZ reanalysis, the mouse islet atlas, and the human ND and T2D scRNA-seq datasets, we used scMMGAN to map the published datasets onto our samples. We evaluated the performance of scMMGAN in preserving biological signatures by cosine similarity before and after mapping, and performance in mixing batches with the modified average silhouette width (ASW) of batch after mapping. See the scMMGAN section in the **Methods** for more details.

How was n = 20 clusters determined?

What methods were used to verify that the clusters retained biological identity and were not artifacts of overclustering?

Clustering was performed first to distinguish cellular populations before merging populations with similar expression of key marker genes into unique cell types. Thus, we chose the number of clusters in order to balance overclustering with merging different cell types or merging cell types with low quality clusters. For example, clustering with $n = 10$ merges acinar cells with immune cells into one coarse cluster 5 (**Reviewer Fig. 4a,b**). It also merges low quality cluster 13 with the stellate population, and low quality cluster 14 with the ductal population. Clustering with 30 clusters further subdivides existing clusters but does not reveal new populations (**Reviewer Fig. 4c,d**). Furthermore, our clustering provided a stable clustering over multiple runs based on adjusted rand index (ARI, where $ARI=1.0$ means clustering is identical, 0.0 for random labelings) (**Reviewer Fig. 4e**). We annotated and merged clusters into a representative cell type based on strong expression of marker genes from prior literature. Importantly, as the goal is not to identify new subpopulations of pancreatic islet cells but coarsely identify endocrine cells and β cells for confirmation with *in vivo* genetic lineage tracing, this procedure was sufficient. We have added results on the mean ARI and justification for number of clusters to the **Methods** section:

*“All cells were then clustered with k-means clustering ($n_clusters=20$) and clusters were annotated based on published mouse and human islet marker genes²⁰ to distinguish endocrine cells and β cells for downstream analysis. Clusters 13 and 14 were annotated as low-quality clusters due to lack of enrichment for key markers of pancreatic islet cell populations and distinct embedding visualization. Cluster 13 showed low overall counts, strong enrichment for hemoglobin genes, and low ribosomal counts, indicative of contaminating erythroid cells, and cluster 14 showed low ribosomal counts and enrichment primarily for marks such as *Malat1*, *Neat1*, and *Gm42418*, lncRNAs whose high expression reflects high nuclear fraction and cytoplasmic RNA loss due to membrane damage²². Cluster 11 was further split into clusters 11 and 20 due to enrichment of Schwann cell markers *S100b*, *Ngfr*, and *Pmp22* in cluster 20. This clustering was chosen due to stability over five runs (average pairwise $ARI=0.86$) and balance between overclustering and merging different cell types.”*

Reviewer Fig. 2. Clustering.

- PHATE plot of exocrine and endocrine islet cells, colored by cluster annotation with 10 clusters.
- Dot plots of cluster expression for marker genes pertaining to 10 clusters.
- PHATE plot of exocrine and endocrine islet cells, colored by cluster annotation with 30 clusters.
- Dot plots of cluster expression for marker genes pertaining to 30 clusters.
- Pairwise adjusted rand index (ARI) across five clustering runs.

Trajectory Analysis Validity

Given the differences in age and conditions among the animals, could the trajectory analysis be artificially linking cell types that may not truly represent progenitor or

intermediate states?

We thank the reviewer for this comment. As shown in **Fig. 6** and **Extended Fig. 9**, our obesity trajectory is associated strongly with diabetes but not age in both humans and mouse, arguing that differences in age are not a major confounding factor in our analyses. More importantly, we validated the predicted *in silico* trajectories with *in vivo* genetic lineage tracing, confirming that Ngn3+ cells (which correspond to WT AT 4) expand and adapt to generate CCK+ β cells (**Fig. 4**), meaning that the predicted trajectories are likely not artificially linking cell states. To the reviewer's point, traditional pseudotime approaches, including diffusion pseudotime²³ and Slingshot²⁴ derive trajectories by inferring the ordering of the cells according to expression similarity, and these approaches have the potential to artificially draw trajectories in datasets where there is no trajectory structure. By contrast, TrajectoryNet is guided by the additional information that all trajectories should start from cells from WT samples, then trace through cells from HFD samples, and end at cells from *Lep^{ob/ob}* samples. Thus, the trajectory analysis identifies the optimal continuous path for a given *Lep^{ob/ob}* cell back to the WT cell population. We note this distinction in the revised **Results** section:

“TrajectoryNet is designed to interpolate continuous dynamics for every single cell across distinct timepoints, where the dynamics learned are biologically plausible through ensuring energy efficiency and modeling cellular proliferation. By learning individual cellular trajectories and leveraging real timepoint dynamics to guide trajectories, this approach represents an advancement over traditional trajectory-based inference methods, which infer a single pseudotemporal ordering over the entire population of cells based on expression similarity, which may not correspond to true latent dynamics²⁵.”

The authors hypothesize that acinar cells surround the islets (peri-islet localization). Can they compare their gene signature with spatial transcriptomics data from the literature to validate this claim?

We thank the reviewer for this helpful suggestion. Tosti et al.¹¹ previously identified three acinar cell subtypes in humans, including a Reg+ subtype that they spatially confirmed was enriched in the peri-islet area. We showed significant enrichment for the Reg+ acinar gene signature from Tosti et al. in our Reg+ cluster (**Fig 9i**), arguing that acinar cells from our islet cell isolations harbor peri-islet acinar cell features found in humans.

Fig. 9. Peri-islet acinar cell transcriptional alterations in obesity.

- i) Enrichment of the human peri-islet acinar-REG⁺ differential gene expression signature derived from Tosti et al¹² for each murine acinar cell cluster. p-value of Wilcoxon rank sum test is shown.

Gene Expression Measurements (Figure 2)

what metric is used for gene expression measurement?

How are normalized counts defined in this analysis?

All scRNA-seq sample data were processed using Cell Ranger v3.0.2 with mm10 mouse genome indices from 10X Genomics into UMI counts. which were then further filtered and L1 normalized for library size, then transformed by square root transformation. This was provided in the **Methods** and reannotated in **Fig. 3** to reflect the expression colorbar title used for all other figures.

Validation of β -Cell Archetypes

The authors should validate β -cell archetypes using IHC or FACS.

Given the lack of unique markers for each archetype, archetypes cannot be reliably distinguished by IHC or FACS. The goal of archetypal analyses was not to firmly define novel archetypes, thus a careful validation of every archetype using orthologous methods is beyond the scope of this paper. Instead, we used this approach to determine whether a subpopulation of cells could be the cell-of-origin for β cells that express CCK⁺ in obesity. Indeed, we validated the *in silico* findings using *in vivo* genetic lineage tracing to show that Ngn3⁺ cells (which are enriched in WT AT 4) are a subpopulation of β cells that expand and adapt to generate CCK⁺ β cells (**Fig. 4**). We note that we provide *in silico* validation for our archetypes through mapping cells from distinct conditions, developmental stages, and species to our archetypes (**Fig. 6** and **Extended Data Fig. 9**). The high confidence of this mapping based on the entire transcriptional profile of the cells is strong evidence that our archetypes reflect true and shared signatures of β cell adaptation to diverse stressors. We note this as a key point of validation in our **Discussion**:

“These approaches, used in conjunction, represent a significant advance in single-cell analysis to attain novel biological insights, especially in the context of β cell biology. Standard practices in single-cell analysis characterize cellular states based on clustering and pseudotime inference²⁵. Indeed, the majority of published single-cell analyses of mouse and human pancreatic islets in normal physiology and diabetes involved unsupervised clustering to identify endocrine and exocrine cell types, then subclustering or identifying factors of variation within cell types of interest, including β cells^{21,26-29}. However, these analyses are limited in their ability to infer cells-of-origin, gene regulatory relationships, and transcriptional dynamics within and across cell types, analyses enabled by TrajectoryNet through calculation of cell-specific trajectories. Furthermore, unlike subclustering, latent-space archetypal analysis (AAnet) avoids arbitrarily discretizing the continuous state space and losing gene signals and behavior that result from high plasticity and transformation between states³⁰. By mapping published datasets directly to our obesity progression through batch integration, scMMGAN established that β cell

archetypes in the context of obesity are highly related to β cell states emerging in the context of diverse stressors across the entire transcriptome, beyond marker genes or β cell-specific signatures. The high confidence of scMMGAN mappings across distinct conditions, developmental stages, and species provides in silico validation of our identified archetypes.”

How was the number of subpopulations determined?

The chosen number of archetypes is the one where the “held-out” cells are best represented in the simplicial latent space. The details on this procedure are outlined in the **Methods** section and associated **Extended Data Fig. 5a**:

*“To determine the number of archetypes for each condition³¹, we first generated a training and test set, where the training test was generated by density subsampling 80% of the dataset, and the test set was the held-out group. We then trained AAnet on the training set 3 times for each number of archetypes between 2 and 10. For each trained model, we calculated the mean squared error (MSE) of the input versus reconstructed cells for the held-out test set. We plotted the MSE and chose the elbow point as the number of archetypes (**Extended Data Fig. 5a**).”*

In WT mice, how do the β -cell archetypes vary with age?

Based on our scMMGAN mapping of non-diabetic (ND) mice from different developmental stages onto our trajectory, we see some differences in archetypal commitment between mice of different ages. We have revised the **Results** and **Extended Data Fig. 9** to specifically highlight these features, which confirm that aging alone does not mimic the effects of obesity”

*“ND mice from different developmental stages showed differences in archetypal commitment. Embryonic ND samples had a significantly higher proportion of $Lep^{ob/ob}$ AT 2 versus other samples (**Extended Data Fig. 9e-g**), suggesting Cck-hi cells result from dedifferentiation towards an embryonic state. In contrast, postnatal P16 mice (when β cells begin to mature) had a significantly higher proportion of WT AT 4 (**Extended Data Fig. 9e-g**), an immature state. 2-month-old mice were enriched for archetypal states representing an intermediate maturation phenotype WT AT 3 and HFD AT 2; 4-6-month-old mice were enriched for WT AT 1,5,7 and HFD AT 1, representing a mature phenotype, and 20-month-old mice were enriched for intermediate maturation archetypes WT AT 2 and 6 (**Extended Data Fig. 9g**). These data argue that aging alone does not mimic the effects of obesity.”*

Extended Data Fig. 9. Comparison of obesity progression to β cells from non-diabetic (ND) and type II diabetes (T2D) mouse models.

- e) Archetypal proportion for Embryonic and P16 samples versus other samples. Box plots display 25th, 50th, and 75th percentiles +/- 1.5 interquartile range (IQR). p-value of Wilcoxon rank sum test is shown.
- f) Archetypal proportion for WT AT 4, *Lep^{ob/ob}* AT 1, *Lep^{ob/ob}* AT 2, and *Lep^{ob/ob}* AT 3 for ND datasets over age (embryonic to aged mice). Box plots display 25th, 50th, and 75th percentiles +/- 1.5 interquartile range (IQR). p-value of Wilcoxon rank sum test is shown.
- g) Archetypal proportion for ND datasets from each age (embryonic to aged mice). Box plots display 25th, 50th, and 75th percentiles +/- 1.5 interquartile range (IQR).

Batch Effect Concerns and Best Practices

The authors state that datasets were not directly comparable due to batch effects. However, best practices in single-cell data integration should be followed to improve comparability: <https://www.sc-best-practices.org/preamble.html>.

We thank the reviewer for this comment. We discussed batch effects in the context of comparing external studies to our dataset. In order to align cells from these studies, we used scMMGAN³², a batch integration method which aligns datasets based on the underlying data geometry. We followed best practices for single-cell data integration by showing scMMGAN results in both removal of batch effects and preservation of biology. We described this in detail in the **Methods** scMMGAN *Evaluation of integration* section. In particular, we ensured batch removal by evaluating the modified average silhouette width (ASW) of batch³³, and we ensured preservation of biological signatures by evaluating the cosine similarity of each cell before and after mapping³⁴. We found all mappings showed high batch integration and moderate to high

preservation of biology, with no biological distortion specific to a particular archetype, developmental stage, disease state, sex, or dataset.

Methodological Reproducibility: Software, Databases, and Parameters

The methods section should explicitly state:

Software versions used (e.g., Seurat, Scanpy, Cell Ranger).

Reference databases used for annotation.

Flags and parameters for each function/tool and justification for their selection.

We thank the reviewer for pointing this out. The **Methods** section now explicitly states software versions for the following tools: Cell Ranger (3.0.2), TrajectoryNet (v0.2.4), AAnet (v0.0), scMMGAN (v0.0), DiffusionEMD (v0.5.0), Scanpy (v1.9.3), PHATE (v1.0.11), GSEAPy (v1.1.2), and MAGIC (v3.0.0).

All reference databases have been cited: the mouse islet atlas (used to annotate endocrine and exocrine cell types), TRRUST v2 (used for gene regulatory inference), and Enrichr (used for gene set enrichment analysis).

All parameters are included in the **Methods** section. All methods were run with default hyperparameters. The following key parameter choices were also listed and justified:

1. TrajectoryNet was run with proliferation regularization hyperparameter $\alpha = 2$. The predicted proliferation rate was robust to the choice of α , tested within the range $[0.01, 10.0]$ (pairwise Spearman ρ between 0.81-1.0).
2. Granger causality computed with time-lag of 10 (10% of the total trajectory length).

Reviewer #5 (Remarks to the Author): co-review with Reviewer #2

We thank the reviewer for their thoughtful comments and appreciate the value of co-review to augment training for early career researchers.

Reviewer #6 (Remarks to the Author): co-review with Reviewer #3

We thank the reviewer for their thoughtful comments and appreciate the value of co-review to augment training for early career researchers.

REFERENCES

- 1 Parajuli, P. *et al.* Pancreatic cancer triggers diabetes through TGF- β -mediated selective depletion of islet β -cells. *Life Science Alliance* **3**, e201900573 (2020). <https://doi.org/10.26508/lsa.201900573>
- 2 Chung, K. M. *et al.* Endocrine-Exocrine Signaling Drives Obesity-Associated Pancreatic Ductal Adenocarcinoma. *Cell* **181**, 832-847 e818 (2020). <https://doi.org/10.1016/j.cell.2020.03.062>
- 3 Zhang, A. M. Y. *et al.* Endogenous Hyperinsulinemia Contributes to Pancreatic Cancer Development. *Cell Metab* **30**, 403-404 (2019). <https://doi.org/10.1016/j.cmet.2019.07.003>
- 4 Zhang, A. M. Y. *et al.* Hyperinsulinemia acts via acinar insulin receptors to initiate pancreatic cancer by increasing digestive enzyme production and inflammation. *Cell Metab* **35**, 2119-2135.e2115 (2023). <https://doi.org/10.1016/j.cmet.2023.10.003>
- 5 Chung, W. J. *et al.* Kras mutant genetically engineered mouse models of human cancers are genomically heterogeneous. *Proc Natl Acad Sci U S A* **114**, E10947-E10955 (2017). <https://doi.org/10.1073/pnas.1708391114>
- 6 Leidner, R. *et al.* Neoantigen T-Cell Receptor Gene Therapy in Pancreatic Cancer. *N Engl J Med* **386**, 2112-2119 (2022). <https://doi.org/10.1056/NEJMoa2119662>
- 7 Hsiue, E. H.-C. *et al.* Targeting a neoantigen derived from a common TP53 mutation. *Science* **371**, eabc8697 (2021). <https://doi.org/doi:10.1126/science.abc8697>
- 8 Cheung, A. F., Dupage, M. J., Dong, H. K., Chen, J. & Jacks, T. Regulated expression of a tumor-associated antigen reveals multiple levels of T-cell tolerance in a mouse model of lung cancer. *Cancer Res* **68**, 9459-9468 (2008). <https://doi.org/10.1158/0008-5472.Can-08-2634>
- 9 Storz, P. Acinar cell plasticity and development of pancreatic ductal adenocarcinoma. *Nat Rev Gastroenterol Hepatol* **14**, 296-304 (2017). <https://doi.org/10.1038/nrgastro.2017.12>
- 10 Storz, P. & Crawford, H. C. Carcinogenesis of Pancreatic Ductal Adenocarcinoma. *Gastroenterology* **158**, 2072-2081 (2020). <https://doi.org/10.1053/j.gastro.2020.02.059>
- 11 Tosti, L. *et al.* Single-Nucleus and In Situ RNA-Sequencing Reveal Cell Topographies in the Human Pancreas. *Gastroenterology* **160**, 1330-1344.e1311 (2021). <https://doi.org/10.1053/j.gastro.2020.11.010>
- 12 Egozi, A., Bahar Halpern, K., Farack, L., Rotem, H. & Itzkovitz, S. Zonation of Pancreatic Acinar Cells in Diabetic Mice. *Cell Rep* **32**, 108043 (2020). <https://doi.org/10.1016/j.celrep.2020.108043>
- 13 Thorel, F. *et al.* Normal glucagon signaling and beta-cell function after near-total alpha-cell ablation in adult mice. *Diabetes* **60**, 2872-2882 (2011). <https://doi.org/10.2337/db11-0876>
- 14 Thorel, F. *et al.* Conversion of adult pancreatic alpha-cells to beta-cells after extreme beta-cell loss. *Nature* **464**, 1149-1154 (2010). <https://doi.org/10.1038/nature08894>
- 15 Tamarina, N. A., Roe, M. W. & Philipson, L. H. Characterization of mice expressing Ins1 gene promoter driven CreERT recombinase for conditional gene deletion in pancreatic β -cells. *Islets* **6**, e27685 (2014). <https://doi.org/10.4161/isl.27685>
- 16 Wicksteed, B. *et al.* Conditional gene targeting in mouse pancreatic β -Cells: analysis of ectopic Cre transgene expression in the brain. *Diabetes* **59**, 3090-3098 (2010). <https://doi.org/10.2337/db10-0624>
- 17 Cheng, C. W. *et al.* Fasting-Mimicking Diet Promotes Ngn3-Driven β -Cell Regeneration to Reverse Diabetes. *Cell* **168**, 775-788.e712 (2017). <https://doi.org/10.1016/j.cell.2017.01.040>

- 18 Huang, X. *et al.* Ductal or Ngn3(+) cells do not contribute to adult pancreatic islet beta-cell neogenesis in homeostasis. *Embo j* **44**, 2856-2881 (2025). <https://doi.org/10.1038/s44318-025-00434-z>
- 19 Cheng, C. W. *et al.* Fasting-Mimicking Diet Promotes Ngn3-Driven beta-Cell Regeneration to Reverse Diabetes. *Cell* **168**, 775-788 e712 (2017). <https://doi.org/10.1016/j.cell.2017.01.040>
- 20 Hrovatin, K. *et al.* Delineating mouse beta-cell identity during lifetime and in diabetes with a single cell atlas. *Nat Metab* **5**, 1615-1637 (2023). <https://doi.org/10.1038/s42255-023-00876-x>
- 21 Baron, M. *et al.* A Single-Cell Transcriptomic Map of the Human and Mouse Pancreas Reveals Inter- and Intra-cell Population Structure. *Cell Syst* **3**, 346-360 e344 (2016). <https://doi.org/10.1016/j.cels.2016.08.011>
- 22 Clarke, Z. A. & Bader, G. D. MALAT1 expression indicates cell quality in single-cell RNA sequencing data. *bioRxiv*, 2024.2007.2014.603469 (2024). <https://doi.org/10.1101/2024.07.14.603469>
- 23 Haghverdi, L., Büttner, M., Wolf, F. A., Buettner, F. & Theis, F. J. Diffusion pseudotime robustly reconstructs lineage branching. *Nature Methods* **13**, 845-848 (2016). <https://doi.org/10.1038/nmeth.3971>
- 24 Street, K. *et al.* Slingshot: cell lineage and pseudotime inference for single-cell transcriptomics. *BMC Genomics* **19**, 477 (2018). <https://doi.org/10.1186/s12864-018-4772-0>
- 25 Luecken, M. D. & Theis, F. J. Current best practices in single-cell RNA-seq analysis: a tutorial. *Mol Syst Biol* **15**, e8746 (2019). <https://doi.org/10.15252/msb.20188746>
- 26 Segerstolpe, A. *et al.* Single-Cell Transcriptome Profiling of Human Pancreatic Islets in Health and Type 2 Diabetes. *Cell Metab* **24**, 593-607 (2016). <https://doi.org/10.1016/j.cmet.2016.08.020>
- 27 Lawlor, N. *et al.* Single-cell transcriptomes identify human islet cell signatures and reveal cell-type-specific expression changes in type 2 diabetes. *Genome Res* **27**, 208-222 (2017). <https://doi.org/10.1101/gr.212720.116>
- 28 Wang, Y. J. *et al.* Single-Cell Transcriptomics of the Human Endocrine Pancreas. *Diabetes* **65**, 3028-3038 (2016). <https://doi.org/10.2337/db16-0405>
- 29 Muraro, M. J. *et al.* A Single-Cell Transcriptome Atlas of the Human Pancreas. *Cell Syst* **3**, 385-394 e383 (2016). <https://doi.org/10.1016/j.cels.2016.09.002>
- 30 Venkat, A. *et al.* AAnet resolves a continuum of spatially-localized cell states to unveil tumor complexity. *bioRxiv*, 2024.2005.2011.593705 (2024). <https://doi.org/10.1101/2024.05.11.593705>
- 31 van Dijk, D. *et al.* Finding archetypal spaces using neural networks. *arXiv* **1901.09078v2** (2019). <https://doi.org/https://arxiv.org/abs/1901.09078v2>
- 32 Amodio, M. *et al.* Single-cell multi-modal GAN reveals spatial patterns in single-cell data from triple-negative breast cancer. *Patterns (N Y)* **3**, 100577 (2022). <https://doi.org/10.1016/j.patter.2022.100577>
- 33 Luecken, M. D. *et al.* Benchmarking atlas-level data integration in single-cell genomics. *Nature Methods* **19**, 41-50 (2022). <https://doi.org/10.1038/s41592-021-01336-8>
- 34 Zhang, Z. *et al.* Signal recovery in single cell batch integration. *bioRxiv* (2023). <https://doi.org/10.1101/2023.05.05.539614>

RESPONSE TO REVIEWERS

Reviewer #1 (Remarks to the Author):

The manuscript has been significantly improved by incorporating text changes and adding additional experiments. I have no further scientific comments.

We thank the reviewer for their positive feedback on our work.

Reviewer #2 (Remarks to the Author):

In this manuscript, the authors use genetic and computational models to demonstrate that obesity drives a pro-tumorigenic beta cell state, and that endocrine-exocrine crosstalk via CCK is important for obesity-associated pancreatic ductal adenocarcinoma. The authors use bioinformatic analysis of scRNA-seq followed with lineage tracing in vivo to show that obesity drives cellular stress via cJUN/JNK, leading to proliferation of immature (Ngn3+) CCK-expressing beta cells that could drive tumorigenesis. Through additional model characterization and modification/reorganization of the text the authors have sufficiently addressed our concerns. The authors have re-organized assessment of the KPC-Akita model which improves the clarity of the manuscript, highlighting the role of beta cells in tumor development in lean mice, and thus independent of obesity and CCK. Inclusion of a new CCKfl/fl allele crossed to the KCO mouse model provides compelling evidence that beta cell derived CCK is a driver of obesity-associated tumor development. The authors should be congratulated on this nice study that will expand our understanding regarding the relationship between obesity and PDAC.

My minor suggestions for improvements, are:

The authors should briefly note the timepoint at which each model was assessed for disease burden either in the text or legends. If CCKfl/fl KCO mice were assessed at the same timepoints as KC counterparts it would be helpful to present the disease burden in KCO +/- CCK side by side with KC counterparts to demonstrate the magnitude of the effect relative to lean mice.

In both the main text and all associated figure legends, we have clarified that disease burden was assessed in 3-month-old littermates for all KCO mouse studies. While this does mean that disease burden for KCO;CCK^{Flox/Flox} mice (Fig. 2) was assessed at the same timepoint as KC;Lep^{ob/+} mice (Fig. 1), we have chosen to present these data in separate figures as these mice were generated from separate crosses and are thus not littermates, which confounds attempts to directly compare disease burden across groups.

Additionally, if the authors assessed ADM, PanIN, and PDAC across mouse models it would be more informative to report the burden of each rather than combined disease burden.

We thank the reviewer for this comment. We did not subcategorize ADM, PanIN, and PDAC in our disease burden and tumor staging analyses. Given differences in proliferation rates between and within these morphologically distinct lesions, it is not clear that subclassifying these lesions would provide additional insight on the propensity to undergo morphologic progression from ADM to PanIN and ultimately to PDAC. Consequently, we believe that the presence or absence of PanIN and PDAC (as currently described in the text) is a more meaningful description of tumor progression in our mouse models.

In Figure 8f, it is somewhat confusing that CD45 and SMA+ cells are quantified within PanIN, PDAC lesions; I'm not sure what that would mean. Were these cells quantified with FOV containing PanIN or PDAC lesions?

Yes, CD45+ and SMA+ cells were quantified within fields of view containing PanIN and PDAC lesions. This has been clarified in the figure legend for Figure 8f and described in more detail in the Methods section.

Reviewer #3 (Remarks to the Author):

The reviewed manuscript submitted by Garcia et al. has addressed most of our previous concerns and figures have improved. The results support much better the conclusions of the manuscript.

Most strikingly, the addition of the CCK overexpression and CCK knockout models has really strengthened the conclusions of this manuscript and shows a much clearer role for CCK on PDAC progression than the first iteration of this manuscript. The serial glucose measurements between the two models strengthens the conclusions proposed in the original manuscript, that hyperglycaemia is not the main driver of PDAC progression in these models. And the correlation analyses showing CCK expression associates with disease burden instead of insulin strengthens these conclusions.

The revised manuscript is clearer, the figures are easier to interpret, and the story has a much better flow. They have corrected many of the statements that we felt were overstated or not clear in the original manuscript. The enrichment of the Acinar-REG+ signature identified by Tosti et al from the human pancreas in cluster 3 better supports that the acinar cells are of peri-islet origin. They have addressed the question of sex bias (where they previously only used male mice for the STZ and scRNAseq experiment) by performing all their new in-vivo experiments with both sexes and showing no significant differences between the sexes, suggesting the vast majority of their results will still be relevant for female patients.

Overall, we think the authors took great efforts to address many of the reviewer's previous concerns, and the manuscript is a lot stronger for it. I endorse the acceptance of this paper.

We thank the reviewer for their acknowledgment of the improvement of the manuscript and overall enthusiasm for our work.

Reviewer #4 (Remarks to the Author):

Overall, the revisions are appreciated. The revised organization and the new data linking β -cell CCK expression to obesity-associated PDAC progression improve the readability of the manuscript.

We thank the reviewer for the acknowledgment of the improvement of the manuscript.

Please add minimal clarifications and diagnostics to strengthen rigor and reproducibility: specify the feature space and cell subset used for EMD.

The EMD analysis provided was done on all endocrine cells, measured from the principal component (PC) latent space used for visualization, trajectory inference, and trajectory analysis. We repeated the analysis in the PC latent space for β cells only, showing that the same trend holds. Specifically, the two WT samples (12- and 16-week-old) were more similar to each other (EMD=0.59) than their respective age-matched samples (EMD=0.94 and 0.66). Both samples were closer to the HFD-fed sample than the *Lep^{ob/ob}* sample (EMD=0.59 vs. 0.94 for the 12-week-old and 0.66 vs. 0.84 for 16-week-old). This has been added to the **Methods** section under “Preprocessing of scRNA-seq data”.

The added EMD analysis is a useful global similarity check, but it is not sufficient to support a claim of “lack of batch effect”: sample-level EMD can be driven by differences in cell-state composition and QC/capture differences, may miss subtle batch/age shifts within β cells (or β substates), and does not directly validate the absence of batch effects in the same latent space used for downstream clustering/trajectory unless that is explicitly shown. Therefore, please temper the statement “verifying lack of batch effect” and include one within- β cell batch/age mixing diagnostic on the trajectory input/latent space (e.g., ASW_batch, iLISI, or kBET) plus a β -cell embedding colored by sample, and show the distribution of trajectory position/pseudotime by sample to demonstrate the inferred trajectory is not simply tracking sample identity.

In addition to the EMD analysis, we have computed the modified average silhouette width (ASW) to evaluate batch/age mixing within β cells. This metric measures the silhouette of a given batch for each cell type. For the scaled metric we used here, 0 indicates separation of batches, and 1 indicates perfect overlap of batches. In the combined β cell PC embedding space used for visualization, trajectory inference, and trajectory analysis, the two age-matched batches had a modified ASW of 0.71, and the two WT samples (subsetting to WT only) had a modified ASW of 0.84. We have changed the statement “verifying lack of batch effect” to state that “these results suggest higher concordance between the WT samples than their respective age-matched samples”. This has been added to the **Methods** section under “Preprocessing of scRNA-seq data”.

As requested, we have provided a β cell embedding colored by sample in updated **Supplementary Figure 4**.

Supplementary Fig. 4. Single-cell RNA-sequencing reveals β cell adaptation to obesity. **a** Combined embedding of endocrine cells from three conditions (wild-type (WT), high-fat diet (HFD), and *Lep^{ob/ob}*) and annotated clusters based on marker gene expression (color scales represent min to max of normalized UMI, except *Cck* for which max is the 99th percentile; gray denotes 0 normalized UMI). **b** Endocrine cell type counts per condition. **c** Composite β cell embedding colored by sample. **d** Composite β cell embedding with individual samples plotted separately. Age of harvest is listed. Source data are provided as a Source Data file.

Finally, we computed the enrichment score of each sample at each timepoint across two TrajectoryNet runs in β cells and plotted the mean score across both runs, included in updated **Supplementary Figure 5**. This shows that the two WT samples peak first, followed by the HFD sample, then the *Lep^{ob/ob}* sample along the obesity trajectory.

Supplementary Fig. 5. Dynamic changes in β cell hormone expression with obesity.
a β cell embedding colored by condition (wild-type (WT), high-fat diet (HFD), and *Lep^{ob/ob}*). Visualized trajectories to high *Cck*-expressing cells on obesity progression axis are shown. **b** Enrichment scores (mean over $n=2$ TrajectoryNet runs) for each sample within cells on the trajectory from WT to high *Cck*-expressing calculated at each timepoint, demonstrating trajectories are guided by obesity progression and not age-matched batch. **c** β cells colored by

insulin (*Ins1*, *Ins2*) and *Cck* scaled expression (color scale represents min to the 99th percentile of normalized UMI; gray denotes 0 normalized UMI). **d** Gene expression (color scale represents min to max of normalized UMI, scaled per gene) across all cells along the obesity progression axis in **(a)** for key marker genes. **e** Representative co-immunofluorescence images of mouse insulinoma cells (Min6) and *KCO* mice displaying nuclei (DAPI, blue), CCK (green), and insulin (red). **f** Representative co-immunoelectron microscopy images of Min6 cells labeled with insulin (5 nm dots) and CCK (10 nm dots). Arrows denote insulin and CCK at the plasma membrane, indicating possible active co-secretion. Scale bar, 100 nm. Source data are provided as a Source Data file.

Finally, it may be more appropriate to soften trajectory/archetype language to reflect that these tools generate testable hypotheses regarding lineage relationships or candidate cells of origin, whereas the definitive evidence for lineage directionality is provided by the subsequent in vivo lineage tracing experiments.

We thank the reviewer for this comment. In order to reflect the hypothesis-generating nature of the trajectory analyses, we have revised the text to state that these analyses “*raise the possibility* that CCK+ cells arise from pre-existing β cells,” and that “WT AT4 *may* give rise to CCK-expressing β cells in obesity.”